# High-Probability Convergence Guarantees of Decentralized SGD

**Aleksandar Armacki** [1]   **Ali H. Sayed** [1]

## Abstract

Convergence in high-probability (HP) has attracted increasing interest, due to implying exponentially decaying tail bounds and strong guarantees for individual runs of an algorithm. While many works study HP guarantees in centralized settings, much less is understood in the decentralized setup, where existing works require strong assumptions, like uniformly bounded gradients, or asymptotically vanishing noise. This results in a significant gap between the assumptions used to establish convergence in the HP and the mean-squared error (MSE) sense, and is also contrary to centralized settings, where it is known that **SGD** converges in HP under the same conditions on the cost function as needed for MSE convergence. Motivated by these observations, we study the HP convergence of Decentralized **SGD** (**DSGD**) in the presence of light-tailed noise, providing several strong results. First, we show that **DSGD** converges in HP under the same conditions on the cost as in the MSE sense, removing the restrictive assumptions used in prior works. Second, our sharp analysis yields order-optimal rates for both non-convex and strongly convex costs. Third, we establish a linear speed-up in the number of users, leading to matching or strictly better transient times than those obtained from MSE results, further underlining the tightness of our analysis. To the best of our knowledge, this is the first work that shows **DSGD** achieves a linear speed-up in the HP sense. Our relaxed assumptions and sharp rates stem from several technical results of independent interest, including a result on the variance-reduction effect of decentralized methods in the HP sense, as well as a novel bound on the moment-generating function of strongly convex costs, of interest even in centralized settings. Numerical experiments validate our theory.

[1]STI, EPFL, Lausanne, Switzerland. Correspondence to: Aleksandar Armacki <aleksandar.armacki@epfl.ch>.

*Proceedings of the 43rd International Conference on Machine Learning*, Seoul, South Korea. PMLR 306, 2026. Copyright 2026 by the author(s).

## 1. Introduction

Modern large-scale machine learning applications and the abundance of data necessitate alternatives to the centralized computation framework, giving rise to distributed learning, a paradigm where multiple users collaborate to jointly train a model, e.g., (Sayed, 2014b; McMahan et al., 2017; Vlaski et al., 2023). The features of distributed learning, such as storing the data locally and only exchanging smaller updates, like (quantized) model parameters or gradients, and the lack of a single point of failure, further make it an attractive paradigm from a privacy and security perspective (Bonawitz et al., 2017; Yu & Kar, 2023). Many applications, such as federated training of models on mobile devices (Konečný et al., 2016), controlling and coordinating robot swarms (Bullo et al., 2009), or distributed control and power grids (Chang et al., 2020), all rely on distributed computation. From a communication perspective, distributed frameworks can be client-server (i.e., federated) or decentralized (i.e., networked), with the main difference being that in the client-server setup users communicate with a central server, while in decentralized settings users communicate directly with each other.[1] Noting that, from the model update perspective, the client-server setup is equivalent to the decentralized setup with a fully connected communication network, we focus on the more general, decentralized setup.

The study of convergence guarantees of decentralized optimization algorithms has a long history, e.g., (Nedić & Ozdaglar, 2009; Jakovetić et al., 2014; Shi et al., 2015; Lorenzo & Scutari, 2016; Yuan et al., 2016; Jakovetić, 2019; Swenson et al., 2022), with most works focusing on MSE convergence, e.g., (Jakovetić et al., 2018; Pu & Nedić, 2021; Vlaski & Sayed, 2021a;b; Koloskova et al., 2020; Xin et al., 2020; 2021; Wang & Joshi, 2021), see also (Sayed, 2014a) for an extensive treatment of the topic. Another type of convergence guarantees, namely convergence in HP, has garnered increasing attention recently. In particular, for a

---

[1]While the client-server setup is often characterized by performing local updates and periodic communication (Stich, 2019), as well as partial user participation (Cho et al., 2022), we note that these features can also be incorporated in the decentralized setup, by modifying the methods to perform multiple local updates before communicating (Koloskova et al., 2020) and considering a dynamic network with users that are occasionally idle (Cannelli et al., 2020). However, this is not the focus of our current work.

non-negative stochastic process $\{X^t\}_{t\in\mathbb{N}}$, the goal of HP convergence is to establish that, for all $t \in \mathbb{N}$ and any $\epsilon > 0$

$$\mathbb{P}(X^t > \epsilon) \leq \exp(-Ct^{\gamma_1}\epsilon^{\gamma_2}), \tag{1}$$

where $\gamma_1, \gamma_2, C > 0$ are constants. If $\{X^t\}_{t\in\mathbb{N}}$ is a measure of performance, e.g., $X^t = \frac{1}{t}\sum_{k\in[t]} \|\nabla f(x^k)\|^2$, where $\{x^t\}_{t\in\mathbb{N}}$ is a sequence generated by some algorithm and $f$ is a non-convex cost, the relation (1) provides strong guarantees with respect to a single run of that algorithm. This is particularly important in modern applications like LLMs, where it is often intractable to perform multiple runs. Numerous works study HP guarantees of **SGD**-type methods in centralized settings, under both light-tailed (Nemirovskiĭ et al., 2009; Ghadimi & Lan, 2013; Li & Orabona, 2020; Harvey et al., 2019; Bajović et al., 2023; Liu et al., 2023a) and heavy-tailed noise (Nguyen et al., 2023; Liu et al., 2023b; Hübler et al., 2025; Kornilov et al., 2025; Armacki et al., 2025). Comparatively, there have been very few studies of HP convergence of decentralized methods.

## 1.1. Literature Review

We now review the literature, focusing on centralized HP results and decentralized MSE and HP convergence results.

**Centralized HP Convergence.** Nemirovskiĭ et al. (2009) show optimal convergence rates of **SGD** for convex costs under light-tailed stochastic gradients (see assumption **(A4)** ahead for a formal definition of light-tailed noise). Ghadimi & Lan (2013) establish, among other, the optimal HP convergence rate of **SGD** for non-convex costs under light-tailed noise, while Li & Orabona (2020) show the same for momentum **SGD**. Harvey et al. (2019) and Bajović et al. (2023) respectively provide optimal HP convergence rates for the last iterate of **SGD** for non-smooth and smooth strongly convex costs, while Liu & Zhou (2024) establish unified HP convergence guarantees for smooth and non-smooth convex and strongly convex costs. Liu et al. (2023a) generalize the previous works on non-convex and convex costs, providing unified guarantees for several algorithms, including **SGD** and AdaGrad for smooth and non-smooth costs. Madden et al. (2024) study HP convergence of **SGD** under sub-Weibull noise. Another line of work studies HP convergence under heavy-tailed noise,[2] e.g., (Gorbunov et al., 2020; Sadiev et al., 2023; Nguyen et al., 2023; Liu et al., 2023b; Puchkin et al., 2024; Hübler et al., 2025; Kornilov et al., 2025; Armacki et al., 2025; 2026a;b), where it is necessary to introduce algorithmic modifications, e.g., clipping, normalization, or sign, to ensure concentration of the form in (1). Crucially, convergence in the HP sense is achieved under the same conditions on the cost function as in the

MSE sense, for both light-tailed and heavy-tailed noise.

**Decentralized MSE Convergence.** MSE guarantees are typically studied under a variety of bounds on the second noise moment.[3] Following one of these settings, Jakovetić et al. (2018) show that **DSGD** converges at an optimal rate for strongly convex costs, while Pu & Nedić (2021) show that **DSGD** with a fixed step-size and the gradient tracking (GT) mechanism, e.g., (Lorenzo & Scutari, 2016; Nedić et al., 2017; Qu & Li, 2018), converges to a neighbourhood of the optimal solution. Vlaski & Sayed (2021a;b) study MSE guarantees of **DSGD** for non-convex costs and show that it escapes saddle points with high probability. Wang & Joshi (2021) propose a general framework dubbed cooperative **SGD**, showing optimal rates for non-convex costs. Koloskova et al. (2023) provide unified guarantees for **DSGD** with local updates and changing network topology, with optimal rates and linear speed-up in the number of users for non-convex and (strongly) convex costs. Xin et al. (2020) show **DSGD** with GT and variance reduction converges at a linear rate for strongly convex costs, while Xin et al. (2021) establish optimal rates of **DSGD** with GT and linear speed-up in the number of users for non-convex costs and costs satisfying the Polyak-Łojasiewicz (PL) condition, e.g., (Karimi et al., 2016). Yu et al. (2026) show that **DSGD** with GT, momentum and normalization achieves optimal rates under heavy-tailed noise. It is worth mentioning a rich line of works studying MSE guarantees for decentralized problems such as estimation, detection, multi-objective and multitask optimization, see, e.g., (Kar et al., 2012; Chen & Sayed, 2013; Nassif et al., 2020) and references therein.

**Decentralized HP Convergence.** Compared to MSE guarantees, there is a significantly smaller body of work on HP convergence guarantees of decentralized algorithms. In particular, Lu et al. (2024) study HP convergence of **DSGD** for general non-convex and PL costs under light-tailed noise, requiring uniformly bounded gradients, as well as asymptotically vanishing noise for PL costs (see the discussion after Theorem 3.7 for details), while showing optimal rates in both cases. Lu (2024); Xu et al. (2024) study HP convergence of a decentralized mirror descent algorithm under light-tailed noise, for online noncooperative games and dynamic regrets, respectively, requiring bounded gradients and compact domains. Qin et al. (2025); Yang et al. (2026) study HP convergence of decentralized algorithms with clipping, for convex costs under heavy-tailed noise, while (Yang et al., 2025) also consider general non-convex costs. Finally, Gorbunov et al. (2024) study clipped SGD under heavy-tailed noise, in the distributed client-server (i.e., *fully connected network*) setting. It is worth mentioning works like (Bajović et al., 2011; 2012; Matta et al., 2016a;b; Bajović, 2024),

---

[2]Since heavy-tailed noise is not the focus of our work, we point the reader to (Nair et al., 2022; Nguyen et al., 2023; Armacki et al., 2025) for various conditions used to study heavy-tailed noise.

[3]See, e.g., (Khaled & Richtárik, 2023) for a good overview of the various conditions used in the literature.

where the authors study asymptotic large deviation guarantees for decentralized problems such as detection and inference. A common thread for all these works is the need for uniformly bounded gradients, or algorithmic modifications like gradient clipping, which ensure that the gradients stay bounded, as well as the fact that none of the existing HP studies achieve linear speed-up in the number of users, which is a known feature of decentralized algorithms in the MSE sense in, e.g., (Koloskova et al., 2020; Xin et al., 2021; Alghunaim & Yuan, 2022). These observations raise the following important questions: (i) Can decentralized algorithms converge in the HP sense under the same assumptions on the cost as used for MSE results, where bounded gradients are not required? (ii) Is it possible to achieve linear speed-up in the number of users in the HP sense, which is a well-known feature of decentralized algorithms? This is further contrasted by the centralized setting, where **SGD**-type algorithms converge in both HP and MSE sense under the same conditions on the cost function, e.g., (Ghadimi & Lan, 2013; Liu & Zhou, 2024).

## 1.2. Contributions

Motivated by the observed gaps between the existing literature on HP and MSE convergence in decentralized settings, we revisit the HP convergence guarantees in decentralized optimization, by studying a variant of vanilla **DSGD** in the presence of light-tailed noise, establishing several strong results in the process. In particular, we show that **DSGD** converges in HP under the same conditions on the cost as required in the MSE sense, for both non-convex and strongly convex costs, removing strong conditions like uniformly bounded gradients and (asymptotically) vanishing noise. Next, we show that **DSGD** achieves optimal rates and linear speed-up in the number of users, for both non-convex and strongly convex costs, further improving on existing works and closing the gap between HP and MSE guarantees in decentralized settings. Our results are established by carefully bounding the moment-generating function (MGF) of the quantity of interest (i.e., average norm-squared of the gradient for non-convex and optimality gap of the last iterate for strongly convex costs) and the MGF of the consensus gap. Compared to (Lu et al., 2024), the work closest to ours, we provide several improvements. In particular, we remove the uniformly bounded gradient requirement, as well as the vanishing noise requirement imposed for PŁ costs (see the discussion after Theorem 3.7 ahead), while achieving linear speed-up in the number of users. Compared to works studying MSE guarantees for **DSGD**, e.g., (Koloskova et al., 2020; Jakovetić et al., 2018), our results require directly working with the MGF and carefully balancing between the MGFs of the quantity of interest and of the consensus gap. This is particularly challenging for strongly convex costs, where we need to show an "almost decreasing" property of the MGF

to get improved rates on the last iterate (see Lemma 3.6 and the related discussion ahead). Compared to centralized **SGD**, e.g., (Harvey et al., 2019; Bajović et al., 2023; Liu et al., 2023a; Liu & Zhou, 2024), the main challenge lies in the additional consensus gap and simultaneously dealing with its MGF and that of the quantity of interest. These challenges are resolved by introducing several novelties, outlined next.

**Novelty.** Toward establishing our improved HP guarantees, we faced several challenges requiring novel results. In particular, to remove uniformly bounded gradients, we provide Lemma 3.3 and use the "offset trick" for non-convex costs (see the proof sketch of Theorem 3.4 in Section 4), while for strongly convex costs we establish a novel bound on the MGF of the consensus gap, removing the need for both bounded gradients and heterogeneity, in the form of Lemma 3.5. To achieve linear speed-up, we show that the variance reduction benefit of decentralized learning is maintained in the HP sense, in the form of Lemma 3.2, which is of independent interest when studying HP guarantees of decentralized algorithms. Further, we establish a novel result on the MGF of the optimality gap for strongly convex costs (more broadly, "almost decreasing" processes, see Section 3.3 ahead), in the form of Lemma 3.6, which is essential in ensuring linear speed-up, by providing a more fine-grained bound on the MGF compared to existing centralized results (Harvey et al., 2019; Bajović et al., 2023; Liu & Zhou, 2024) and is of independent interest, even in centralized settings.

**Paper Organization.** The rest of the paper is organized as follows. Section 2 outlines the problem and **DSGD** method, Section 3 presents the main results, Section 4 provides proof sketches and discussions, Section 5 presents numerical experiments, while Section 6 concludes the paper. Appendix contains results omitted from the main body. The remainder of this section introduces the notation used in the paper.

**Notation.** We use $\mathbb{N}$, $\mathbb{R}$ and $\mathbb{R}^d$ to denote positive integers, real numbers and $d$-dimensional vectors. For $m \in \mathbb{N}$, we use $[m] = \{1, \ldots, m\}$ to denote positive integers up to and including $m$. The notation $\langle \cdot, \cdot \rangle$ stands for the Euclidean inner product, while $\| \cdot \|$ is used for both vector and matrix induced norms. We use subscripts to denote users and superscripts to denote the iteration counter, e.g., $x_i^t$ refers to the model of user $i$ in iteration $t$. The "big O" notation $\mathcal{O}(\cdot)$ hides only global constants, unless stated otherwise.

## 2. Problem Setup and DSGD

In this section we introduce the problem of interest and the **DSGD** algorithm. Consider a network of $n \geq 2$ users who can communicate with each other and want to jointly train a model. Formally, the problem can be cast as

$$\arg\min_{x \in \mathbb{R}^d} \left\{ f(x) = \frac{1}{n} \sum_{i \in [n]} f_i(x) \right\}, \qquad (2)$$

## Algorithm 1 DSGD

**Require:** Model initialization $x_i^1 \in \mathbb{R}^d$, $i \in [n]$, step-size schedule $\{\alpha_t\}_{t \in \mathbb{N}}$;
1: **for** $t = 1, 2, \ldots$, each user $i \in [n]$ in parallel **do**
2:     Query the oracle to obtain $\nabla \ell(x_i^t; \xi_i^t)$;
3:     Perform the model update:
       $x_i^{t+1} = \sum_{j \in \mathcal{N}_i} w_{ij}\big(x_j^t - \alpha_t \nabla \ell(x_j^t; \xi_j^t)\big)$;
4: **end for**

where $x \in \mathbb{R}^d$ represents model parameters, $f_i : \mathbb{R}^d \mapsto \mathbb{R}$ is the cost function of user $i \in [n]$, given by $f_i(x) = \mathbb{E}_{\xi_i \sim \mathcal{D}_i}[\ell(x; \xi_i)]$, with $\xi_i \in \Xi$ being a random variable governed by an unknown distribution $\mathcal{D}_i$, while $\ell : \mathbb{R}^d \times \Xi \mapsto \mathbb{R}$ is a loss function. Each user has access to a Stochastic First-order Oracle ($\mathcal{SFO}$), which, when queried by user $i \in [n]$ with input $x \in \mathbb{R}^d$, returns the gradient of $\ell$ evaluated at a random sample, i.e., $\nabla \ell(x; \xi_i)$. The $\mathcal{SFO}$ model subsumes:

1. *Batch (offline) learning* - users have access to a local dataset $\{\xi_{i,l}\}_{l \in [m_i]}$, so that $f_i(x) = \frac{1}{m_i} \sum_{l \in [m_i]} \ell(x; \xi_{i,l})$ and in each round users choose a random sample $\xi_{i,l}$, which is used to compute $\nabla \ell(x; \xi_{i,l})$ and update the model;

2. *Streaming (online) learning* - users do not store a local dataset, but in each round observe a random sample $\xi_i$, which is used to compute $\nabla \ell(x; \xi_i)$ and update the model.

The communication pattern between users is modeled as a static graph $G = (V, E)$, where $V = [n]$ is the set of vertices representing users, while $E \subset V \times V$ is the set of edges representing communication links between users. To solve (2) in decentralized fashion, we consider a version of Decentralized Stochastic Gradient Descent (**DSGD**), based on the Adapt-Then-Combine (i.e., diffusion) approach, e.g., (Lopes & Sayed, 2008; Cattivelli & Sayed, 2010; Chen & Sayed, 2012). The method consists of the following steps. First, users choose a shared step-size schedule $\{\alpha_t\}_{t \in \mathbb{N}}$ and each user $i \in [n]$ chooses an arbitrary, but deterministically selected initial model $x_i^1 \in \mathbb{R}^{d}$.[4] In iteration $t \geq 1$, users query the $\mathcal{SFO}$ with their current model $x_i^t$ and receive $\nabla \ell(x_i^t; \xi_i^t)$. Users then update their models via the rule

$$x_i^{t+1} = \sum_{j \in \mathcal{N}_i} w_{ij}\big(x_j^t - \alpha_t \nabla \ell(x_j^t; \xi_j^t)\big), \qquad (3)$$

which consists of a local update, followed by a consensus step, where $\mathcal{N}_i := \{j \in V : \{i, j\} \in E\} \cup \{i\}$ is the set of users (i.e., *neighbours*) with whom user $i$ can communicate (including $i$ itself), while $w_{ij} > 0$ is the weight user $i$ assigns to user $j$'s model. The method is summarized in Algorithm 1. The version of **DSGD** considered in our work is closely related to the variant based on Combine-Then-Adapt (i.e., consensus+innovation) approach, e.g., (Nedić

& Ozdaglar, 2009; Kar et al., 2012; Kar & Moura, 2013), whose HP convergence is studied in Lu et al. (2024).

## 3. Main Results

In this section we present the main results. Subsection 3.1 states the preliminaries, while Subsections 3.2 and 3.3 provide results for non-convex and strongly convex costs.

### 3.1. Preliminaries

In this subsection we outline the assumptions used in our work. For any $T \geq 1$, let $\{\xi_i^t\}_{t \in [T]}$ be the random samples observed by user $i \in [n]$ up to time $T$ and denote by $\mathcal{F}_T$ the natural filtration with respect to the sequence of user models up to time $T$, i.e., $\mathcal{F}_T := \sigma\left(\left\{\{x_i^1\}_{i \in [n]}, \ldots, \{x_i^T\}_{i \in [n]}\right\}\right)$. For ease of notation, let $z_i^t := \nabla \ell(x_i^t; \xi_i^t) - \nabla f_i(x_i^t)$ and $W \in \mathbb{R}^{n \times n}$, where $[W]_{i,j} := w_{ij}$, denote the stochastic noise and the network communication matrix, respectively.

**(A1)** The network communication matrix $W \in \mathbb{R}^{n \times n}$ is primitive and doubly stochastic.

Assumption **(A1)** is satisfied by connected undirected graphs, as well as a class of strongly-connected directed graphs with doubly stochastic weights, e.g., (Xin et al., 2020).[5] Let $\lambda := \|W - J\|$ denote the network connectivity parameter, where $J := \frac{1}{n} \mathbf{1}_n \mathbf{1}_n^\top \in \mathbb{R}^{n \times n}$ is the ideal communication matrix. It can be shown that **(A1)** implies $\lambda \in [0, 1)$, see, e.g., (Horn & Johnson, 2012).

**(A2)** The global cost $f$ is bounded from below, i.e., $f^\star := \inf_{x \in \mathbb{R}^d} f(x) > -\infty$.

**(A3)** Each local cost $f_i$ has $L$-Lipschitz gradients, i.e., $\|\nabla f_i(x) - \nabla f_i(y)\| \leq L\|x - y\|$, for any $x, y \in \mathbb{R}^d$.

Assumptions **(A2)**-**(A3)** are standard in smooth non-convex optimization, e.g., (Ghadimi & Lan, 2013; Xin et al., 2021). While some works, e.g., (Li et al., 2023), assume gradients of $\ell$ are $L$-Lipschitz continuous, we note that such assumption and **(A3)** aim to exploit different structural properties. Further, it is known that under certain regularity conditions, $L$-Lipschitz gradients of $\ell$ imply $L$-Lipschitz gradients of $f_i$, making such condition slightly stronger than **(A3)**.

**(A4)** The stochastic quantities satisfy the following:

1. The random samples $\{\xi_i^t\}_{i \in [n], t \in [T]}$, are independent across users and iterations.

2. The stochastic gradients are unbiased, i.e., for any $i \in [n]$, $t \geq 1$ and $\mathcal{F}_t$-measurable vector $x \in \mathbb{R}^d$, we have $\mathbb{E}[\nabla \ell(x; \xi_i^t) \mid \mathcal{F}_t] = \nabla f_i(x)$.

3. The gradient noise $z_i^t$ at user $i \in [n]$ and time $t \geq 1$ is

---

[4]While initial models can be any real vectors, possibly different across users, for analysis purposes they need to be deterministic.

[5]Our work can be readily extended to the setting of time-varying networks, see Appendix K for a detailed discussion.

$\sigma_i$-sub-Gaussian, i.e., we have

$$\mathbb{E}\left[\exp\left(\frac{\|z_i^t\|^2}{\sigma_i^2}\right)\mid \mathcal{F}_t\right] \leq \exp(1).$$

The first condition in **(A4)** is standard in decentralized stochastic optimization (Xin et al., 2021; Yu et al., 2026), while the second and third require noise to be unbiased and sub-Gaussian (i.e., *light-tailed*). Light tails are necessary to achieve exponentially decaying tail bounds of the form in (1) if the vanilla stochastic gradient estimator is employed without any modifications (e.g., using clipping operator or estimators like median-of-means) and are widely used in centralized settings, e.g., (Nemirovskiĭ et al., 2009; Ghadimi & Lan, 2013; Li & Orabona, 2020; Harvey et al., 2019; Liu et al., 2023a; Bajović et al., 2023).[6]

**(A5)** The gradients of users have bounded heterogeneity, i.e., for all $x \in \mathbb{R}^d$ and some $A, B \geq 0$,[7] we have $\max_{i \in [n]} \|\nabla f_i(x)\|^2 \leq A^2 + B^2\|\nabla f(x)\|^2$.

A heterogeneity bound of the type in **(A5)** is required to ensure that **DSGD** converges for non-convex costs (Lian et al., 2017; Chang et al., 2020; Koloskova et al., 2020; Wang & Joshi, 2021). Note that **(A5)** is strictly weaker than the uniformly bounded gradient assumption used in (Lu et al., 2024), as it allows users' gradients to grow with the global gradient. Compared to (Koloskova et al., 2020), who analyze the MSE convergence and require a bound on the average heterogeneity, i.e., $\frac{1}{n}\sum_{i \in [n]} \|\nabla f_i(x)\|^2 \leq A^2 + B^2\|\nabla f(x)\|^2$, we impose a slightly stronger condition. While we believe that **(A5)** can be relaxed to average heterogeneity by incorporating a similar analysis technique to the one in (Koloskova et al., 2020), to keep the proofs simple and instructive, we will use assumption **(A5)**.

**(A6)** Each $f_i$ is twice continuously differentiable and $\mu$-strongly convex, i.e., for every $x, y \in \mathbb{R}^d$, we have $f_i(x) \geq f_i(y) + \langle \nabla f_i(y), x - y \rangle + \frac{\mu}{2}\|x - y\|^2$.

Assumption **(A6)** is used for the strongly convex case. It is well known that **DSGD** requires **(A3)** and **(A6)** to hold for each $f_i$, e.g., (Jakovetić et al., 2018; Koloskova et al., 2020; Wang & Joshi, 2021). The heterogeneity bound in **(A5)** and strong convexity of each cost in **(A6)** can be removed by deploying the GT technique, e.g., (Xin et al., 2021), however, this is beyond the scope of the current work. Next, let $\overline{x}^t := \frac{1}{n}\sum_{i \in [n]} x_i^t$, $\overline{g}^t := \frac{1}{n}\sum_{i \in [n]} \nabla \ell(x_i^t; \xi_i^t)$ and $\overline{z}^t := \frac{1}{n}\sum_{i \in [n]} z_i^t$ denote the network-average model, stochastic gradient and noise, respectively. Using (3) and the double stochasticity of $W$, it follows that

$$\overline{x}^{t+1} = \overline{x}^t - \alpha_t \overline{g}^t.$$

---

[6]While it is possible to achieve concentration of the form in (1) under, e.g., sub-Weibull noise, see (Madden et al., 2024) and references therein, the focus of this work is on sub-Gaussian noise.

[7]With at least one of $A, B$ being strictly positive.

We then have the following result.

**Lemma 3.1.** *Let* **(A3)** *hold. If* $\alpha_t \leq \frac{1}{2L}$, *we have*

$$f(\overline{x}^{t+1}) \leq f(\overline{x}^t) - \frac{\alpha_t}{2}\|\nabla f(\overline{x}^t)\|^2 - \alpha_t \langle \nabla f(\overline{x}^t), \overline{z}^t \rangle$$

$$+ \alpha_t^2 L\|\overline{z}^t\|^2 + \frac{\alpha_t L^2}{2n}\sum_{i \in [n]} \|x_i^t - \overline{x}^t\|^2.$$

Lemma 3.1 provides an important deterministic descent-type inequality for **DSGD** and is the starting point for our analysis. The right-hand side of the above inequality consists of terms that arise in centralized **SGD** plus a consensus gap term $\sum_{i \in [n]} \|x_i^t - \overline{x}^t\|^2$, which stems from the decentralized nature of the algorithm and bounding this term is crucial to ensure convergence. To close the subsection, we provide an important result on the behaviour of the stochastic noise in the HP sense. Let $\sigma^2 := \frac{1}{n}\sum_{i \in [n]} \sigma_i^2$ be the average noise parameter. We then have the following result.

**Lemma 3.2.** *If* **(A4)** *holds, then the following are true for any* $t \geq 1$, $i \in [n]$ *and* $\mathcal{F}_t$-*measurable* $v \in \mathbb{R}^d$.

1. $\mathbb{E}\left[\exp\left(\langle v, z_i^t \rangle\right) \mid \mathcal{F}_t\right] \leq \exp\left(\frac{3\sigma_i^2\|v\|^2}{4}\right)$.

2. $\mathbb{E}\left[\exp\left(\langle v, \overline{z}^t \rangle\right) \mid \mathcal{F}_t\right] \leq \exp\left(\frac{3\sigma^2\|v\|^2}{4n}\right)$.

3. $\mathbb{E}\left[\exp\left(\frac{n\|\overline{z}^t\|^2}{15\sigma^2}\right) \mid \mathcal{F}_t\right] \leq 2d\exp(1)$.

Lemma 3.2 provides some properties of noise, importantly showing that the average noise is $\mathcal{O}\left(\frac{\sigma}{\sqrt{n}}\right)$-sub-Gaussian, establishing the variance reduction benefit of decentralized optimization in the HP sense. This result is crucial toward showing that **DSGD** achieves a linear speed-up in the number of users, which we do in the following subsections. We note that the third property introduces a mild logarithmic dependence on the problem dimension $d$, with further discussion on this dependence provided in Section 4 and Appendix L.

### 3.2. Non-Convex Costs

In this subsection we present results for non-convex costs. Let $\Delta_x := \frac{1}{n}\sum_{i \in [n]} \|x_i^1 - \overline{x}^1\|^2$ and $\Delta_f := f(\overline{x}^1) - f^\star$ denote the initial consensus and optimality gap.

**Lemma 3.3.** *If* **(A1)** *and* **(A5)** *hold, then for any* $t \geq 1$

$$\frac{1}{n}\sum_{i \in [n]} \|x_i^{t+1} - \overline{x}^{t+1}\|^2 \leq 2\lambda^{2t}\Delta_x + \frac{4\lambda^2 A^2}{1 - \lambda}\sum_{k \in [t]} \alpha_k^2 \lambda^{t-k}$$

$$+ \frac{4\lambda^2}{n(1 - \lambda)}\sum_{k \in [t]} \alpha_k^2 \lambda^{t-k}\sum_{i \in [n]} \left[\|z_i^k\|^2 + B^2\|\nabla f(x_i^k)\|^2\right].$$

Lemma 3.3 provides a useful deterministic bound on the consensus gap. Define $\overline{\sigma} := \max_{i \in [n]} \sigma_i$. Building on Lemmas 3.1 and 3.3, we get the following result.

**Theorem 3.4.** *Let* **(A1)-(A5)** *hold. If for any* $T \geq 1$, *the step-size is chosen such that* $\alpha_T \equiv \alpha = \min\left\{C, \frac{\sqrt{n}}{\sigma\sqrt{15LT}}\right\}$, *where* $C > 0$ *is a problem related constant satisfying*

$$C \leq \min\left\{\frac{1}{2L}, \frac{n}{9\sigma^2}, \frac{1-\lambda}{\lambda LB\sqrt{48}}, \frac{\sqrt{n}}{3\sigma\sqrt{5L}}, \frac{\sqrt[3]{n}(1-\lambda)^{2/3}}{\overline{\sigma}^{2/3}\lambda^2 L^{2/3}\sqrt[3]{9}}\right\},$$

*then for any* $\delta \in (0,1)$, *with probability at least* $1 - \delta$

$$\frac{1}{nT}\sum_{t\in[T]}\sum_{i\in[n]}\|\nabla f(x_i^t)\|^2 = \mathcal{O}\left(\frac{\sigma\sqrt{L}\big(\Delta_f + \log(2d/\delta)\big)}{\sqrt{nT}}\right.$$
$$\left. + \frac{\Delta_f + \log(1/\delta)}{CT} + \frac{\Delta_x L^2}{(1-\lambda^2)T} + \frac{n\lambda^2 L(A^2 + \sigma^2)}{\sigma^2(1-\lambda)^2 T}\right).$$

Theorem 3.4 establishes HP convergence of **DSGD** to a stationary point (or a set of stationary points), for general smooth non-convex costs and a fixed step-size. We now discuss the bound from a few different perspectives.

*Centralized SGD and linear speed-up.* Compared to centralized **SGD**, which achieves the rate $\mathcal{O}\left(\frac{1}{\sqrt{T}} + \frac{1}{T}\right)$, e.g., (Liu et al., 2023a), we can see that the rate in Theorem 3.4 also consists of two different order terms, namely $\mathcal{O}\left(\frac{1}{\sqrt{nT}} + \frac{n}{T}\right)$. Crucially, the leading term achieves a linear speed-up in the number of users, which is consistent with MSE results, e.g., (Koloskova et al., 2020), showing that **DSGD** retains the benefits of decentralized learning in the HP sense.

*Network effect and transient time.* We can see from the bound in Theorem 3.4 that network connectivity does not affect the leading term $\mathcal{O}\left(\frac{1}{\sqrt{nT}}\right)$, but only affects terms decaying at a faster rate, which is consistent with MSE results, e.g., (Koloskova et al., 2020). Moreover, the rate in Theorem 3.4 implies that the transient time of **DSGD** in the HP sense is of order $\mathcal{O}\left(\frac{n^3}{(1-\lambda)^4}\right)$, matching the transient time of **DSGD** implied by MSE rates, see, e.g., Table I in (Alghunaim & Yuan, 2022). For details on the derivation of the transient time, the reader is referred to Appendix I.

*Effects of noise and heterogeneity.* The noise (through $\sigma, \overline{\sigma}$) and heterogeneity (through $A, B$) mainly affect the higher-order terms,[8] which is again consistent with MSE results. Note that, if $A = 0$, the bound in Theorem 3.4 recovers the convergence rate of noiseless gradient descent, in the sense that, when $\sigma = A = 0$, the above bound becomes $\mathcal{O}\left(\frac{1}{T}\right)$.

*Comparison with Lu et al. (2024).* The authors in (Lu et al., 2024) study HP guarantees of **DSGD** over undirected time-varying networks, under a time-varying step-size schedule $\alpha_t = \frac{a}{\sqrt{t+1}}$, assuming uniformly bounded gradients and establish the rate $\mathcal{O}\left(\frac{\log(T/\delta)}{\sqrt{T}}\right)$. As highlighted next, we provide several important improvements. First, we remove the

uniformly bounded gradient assumption, replacing it with the more general heterogeneity condition **(A5)**, which is facilitated by the use of the "offset trick", see the discussion in Section 4 for details. Next, Lu et al. (2024) are unable to establish a linear speed-up, which is a byproduct of both their step-size choice, as well as their treatment of the noise. In particular, choosing a time-varying step-size for non-convex costs results in a loss of linear speed-up, see Appendix F for a detailed discussion. However, it would be impossible for Lu et al. (2024) to obtain a linear speed-up even under a fixed step-size, as they show no variance reduction benefit of decentralized learning in the HP sense, instead only using the definition of sub-Gaussianity. On the other hand, the variance reduction benefits of decentralized learning established in Lemma 3.2 allow us to achieve linear speed-up in the number of users when a fixed step-size is used, see Appendix J for detailed discussion on the results needed to achieve linear speed-up.[9] Finally, while we consider static networks, our results can be extended to time-varying networks, e.g., (Lu et al., 2024), see Appendix K for details.

### 3.3. Strongly Convex Costs

In this subsection we provide results for the last iterate of strongly convex costs. Recall that strong convexity implies a unique global minimizer, e.g., (Nesterov, 2018), denoted by $x^\star \in \mathbb{R}^d$, with $f^\star := f(x^\star)$, and let $\kappa := \frac{L}{\mu}$ and $\|\nabla \mathbf{f}^\star\|^2 := \sum_{i\in[n]}\|\nabla f_i(x^\star)\|^2$ be the condition number and heterogeneity measure, respectively. It is known from MSE analysis that **DSGD** does not require a heterogeneity bound of the form in **(A5)** for strongly convex costs, e.g., (Jakovetić et al., 2018; Koloskova et al., 2020). To show this is also the case in the HP sense, we next provide a refined version of Lemma 3.3, which carefully leverages properties of strong convexity to bound the MGF of the consensus gap.

**Lemma 3.5.** *Let* **(A1)-(A4)** *and* **(A6)** *hold, let* $a, t_0, K > 0$ *and the step-size be given by* $\alpha_t = \frac{a}{t+t_0}$, *and let* $x_i^1 = x_j^1$, *for all* $i, j \in [n]$. *If* $a = \frac{6}{\mu}$ *and* $t_0 \geq \max\left\{6, \frac{288\overline{\sigma}^2 K}{\mu^2}, \frac{3456\overline{\sigma}^2\lambda^2 K}{\mu^2(1-\lambda)}, \frac{12\lambda L\sqrt{10}}{\mu(1-\lambda)}\right\}$, *then for* $K_{t+1} = (t + t_0 + 2)K$ *and any* $\nu \leq \min\left\{1, \frac{\mu^2}{144\sigma^2 K}\right\}$, *we have*

$$\mathbb{E}\left[\exp\left(\nu K_{t+1}\sum_{i\in[n]}\|x_i^{t+1} - \overline{x}^{t+1}\|^2\right)\right]$$
$$\leq \exp\left(\nu K_{t+1}\left(\sum_{k\in[t]}\lambda^{t-k}S_k + \sum_{k\in[t]}\lambda^{t-k}D_k\right)\right),$$

*where* $S_k, D_k > 0$ *are problem related constants.*

Lemma 3.5 bounds the MGF of the consensus gap, without requiring the bounded heterogeneity condition **(A5)**, with

---

[8]The effect of noise on the leading term can be eliminated by choosing the step-size $\alpha = \min\left\{C, \frac{\sqrt{n}}{\sqrt{LT}}\right\}$.

[9]This is consistent with MSE results, e.g., (Koloskova et al., 2020; Xin et al., 2021; Alghunaim & Yuan, 2022), where linear speed-up in non-convex case is achieved only via a fixed step-size.

$S_k, D_k$ depending on the step-size and other problem parameters, see Appendix G for explicit forms of $S_k, D_k$. We note that the requirement of same model initialization across users can be removed, at the cost of an additional, geometrically decaying term in the exponent on the RHS, however, we use same initialization, for ease of exposition. Prior to stating the main theorem, we provide an important technical result which facilitates sharper bounds and ensures linear speed-up in the number of users.

**Lemma 3.6.** *Let $\{X^t\}_{t\in\mathbb{N}}$ be a sequence of random variables initialized by a deterministic $X^1 > 0$, such that, for some $M \in \mathbb{N}$, $a, t_0 > 0$ and every $t \geq 1$*

$$\mathbb{E}[\exp(X^{t+1})] \leq \mathbb{E}\left[\exp\left(\left(1-\frac{a}{t+t_0}\right)X^t + \sum_{i\in[M]}\frac{C_i}{(t+t_0)^i}\right)\right],$$

*where $C_i > 0$, $i \in [M]$. If $a \in (1,2]$ and $t_0 \geq a$, we have*

$$\mathbb{E}[\exp(X^{t+1})] \leq \exp\left(\frac{(t_0+1)^a X_1}{(t+1+t_0)^a} + \frac{2^a C_1}{a}\right)$$

$$\times \exp\left(\frac{2^a C_2/(a-1)}{t+1+t_0} + \frac{2^a C_3 \log(t+1+t_0)}{(t+1+t_0)^a}\right)$$

$$\times \exp\left(\sum_{j=4}^{M}\frac{2^a t_0^{3-j} C_j}{(j-3)(t+1+t_0)^a}\right).$$

Lemma 3.6 provides a tight bound on the MGF of an "almost decreasing" process, with the optimality gap of strongly convex costs being an instance of such a process. Compared to bounds used in the centralized setting, e.g., (Harvey et al., 2019; Bajović et al., 2023; Liu & Zhou, 2024), Lemma 3.6 is significantly sharper and incorporates higher-order terms, which are crucial for ensuring linear speed-up, see Section 4 and Appendix J for detailed discussions. We are now ready to state the main result for strongly convex costs.

**Theorem 3.7.** *Let (A1)-(A4) and (A6) hold, the step-size be given by $\alpha_t = \frac{a}{t+t_0}$ and let $x_i^1 = x_j^1$, for all $i, j \in [n]$. If $a = \frac{6}{\mu}$, $t_0 \geq \max\left\{6, \frac{3+\lambda}{1-\lambda}, \frac{1960\sigma^2\kappa}{\mu}, \frac{432\overline{\sigma}^2\kappa^2}{\mu}, \frac{12\kappa\lambda\sqrt{10}}{1-\lambda}, \frac{5184\overline{\sigma}^2\lambda^2\kappa^2}{\mu(1-\lambda)}\right\}$ and $\nu = \min\left\{1, \frac{\mu}{432\sigma^2\kappa^2}, \frac{\mu}{72\kappa}\right\}$, then for any $\delta \in (0,1)$ and $T \geq 1$, with probability at least $1 - \delta$, it holds that*

$$\frac{1}{n}\sum_{i\in[n]}\left(f(x_i^T) - f^\star\right) = \mathcal{O}\left(\frac{\nu^{-1}\log(2/\delta) + \sigma^2\kappa\log(2d)/\mu}{n(T+t_0)}\right.$$

$$\left.+ \frac{\lambda^2 L(1+L)(n\sigma^2 + \|\nabla\mathbf{f}^\star\|^{2(1+\kappa^2)/(1-\lambda)})}{(1-\lambda)n(T+t_0)^2}\right),$$

*where $\mathcal{O}(\cdot)$ hides some higher-order terms.*

Theorem 3.7 establishes HP convergence of **DSGD** to the global minimum, for smooth strongly convex costs and time-varying step-size. In Appendix G we provide the full bound,

containing additional higher-order terms. Next, we discuss multiple aspects of our results.

*Centralized SGD and linear speed-up.* Compared to centralized **SGD**, which achieves the HP rate $\mathcal{O}\left(\frac{1}{T} + \frac{1}{T^2}\right)$, e.g., (Liu & Zhou, 2024), the rate in Theorem 3.7 also consists of two terms of different order, namely $\mathcal{O}\left(\frac{1}{nT} + \frac{1}{T^2}\right)$. Similarly to non-convex results, the leading term shows a linear speed-up in the number of users, again matching the corresponding MSE guarantees, e.g., (Koloskova et al., 2020).

*Network effect and transient time.* We can again see that network connectivity only affects higher-order terms. Moreover, the rate in Theorem 3.7 implies a transient time of **DSGD** in the HP sense, of order $\mathcal{O}\left(\max\left\{\frac{n}{1-\lambda}, \frac{1}{(1-\lambda)^2}\right\}\right)$. This is *strictly sharper* than the transient time of **DSGD** implied by MSE rates, of order $\mathcal{O}\left(\frac{n}{(1-\lambda)^2}\right)$, see Table II in (Alghunaim & Yuan, 2022), further highlighting the tightness of our results and improved analysis.

*Effects of noise and heterogeneity.* While heterogeneity only affects higher-order terms (through $\|\nabla\mathbf{f}^\star\|^2$ and $\|\mathbf{x}^1 - \mathbf{x}^\star\|^2$, see Appendix G for the full bound), we can see that the noise affects both the leading and higher-order terms, stemming from the time-varying step-size schedule and the constant $\nu^{-1} = \mathcal{O}\left(\frac{\sigma^2\kappa^2}{\mu}\right)$. Similarly to the discussion after Theorem 3.4, the effect of noise on the leading term can be mitigated by employing a carefully selected fixed step-size. For ease of exposition, we omit the results using a fixed step-size.

*Comparison with Lu et al. (2024).* The authors in (Lu et al., 2024) establish HP convergence of **DSGD** for the more general case of PŁ costs, providing a rate of $\mathcal{O}\left(\frac{1}{T}\right)$, using the same time-varying step-size schedule as in Theorem 3.7. Crucially, the authors require two very strong assumptions: $(i)$ *path-wise uniformly bounded gradients*, i.e., $\|\nabla f_i(x_i^t)\| \leq G$, for some $G > 0$ and all $t \geq 1$, almost surely; $(ii)$ *asymptotically vanishing noise*, i.e., for all $i \in [n]$ and any $t \geq 1$, they require the following

$$\mathbb{E}\left[\exp\left(\frac{\|z_i^t\|^2}{\alpha_t^2\sigma_i^2}\right) \mid \mathcal{F}_t\right] \leq \exp(1),$$

where $\alpha_t = \frac{a}{t+t_0}$ is the step-size. Both conditions are difficult to satisfy, as $(i)$ is a path-wise relaxation of the uniformly bounded gradient condition (which is not satisfied for strongly convex costs), while $(ii)$ implies that the noise is $\frac{\sigma_i}{t+t_0}$-sub-Gaussian at time $t$, meaning that it vanishes at rate $\mathcal{O}\left(\frac{1}{t+t_0}\right)$.[10] Crucially, Lu et al. (2024) are unable to establish linear speed-up, even though they impose much

---

[10]Markov's inequality implies that a $\frac{\sigma}{t+t_0}$-sub-Gaussian random variable $X$ satisfies $\mathbb{P}\left(X^2 \leq \frac{\sigma^2(\log(1/\delta)+1)}{(t+t_0)^2}\right) \geq 1-\delta$, for any $\delta \in (0,1)$. Similarly, Jensen's inequality implies $\mathbb{E}[X^2] \leq \frac{\sigma^2}{(t+t_0)^2}$.

stronger assumptions on the noise and cost. On the other hand, we show optimal convergence rate with linear speed-up using standard assumptions, with sharper transient times compared to MSE results. These improvements stem not only from a tighter analysis and Lemma 3.2, but also Lemmas 3.5 and 3.6, which allow us to remove the bounded heterogeneity condition **(A5)** and ensure linear speed-up is achieved, see Appendix J for a further discussion.

*On the step-size.* Throughout this section we used the step-size $\alpha_t \propto t^{-1}$, known to be optimal for strongly convex costs, e.g., (Rakhlin et al., 2012). We extend the analysis in Appendix M to a general step-size $\alpha_t \propto t^{-\eta}$, $\eta \in (1/2, 1]$, establishing the HP rate $\mathcal{O}\left(\frac{\log(2d/\delta)}{nt^{2\eta-1}} + \frac{1}{t^{3\eta-1}}\right)$, providing an important insight: for strongly convex costs the rate of decay in $n$ is always optimal and independent of the step-size. The reader is referred to Appendix M for details.

# 4. Proof Outlines and Discussion

In this section we briefly outline proof sketches of the main results and provide further discussions.

*Proof sketch of Theorem 3.4.* Using Lemma 3.1, rearranging and summing up the first $T$ terms, we get

$$
\sum_{t \in [T]} \frac{\alpha_t}{2} \|\nabla f(\overline{x}^t)\|^2 \leq \Delta_f - \sum_{t \in [T]} \alpha_t \langle \nabla f(\overline{x}^t), \overline{z}^t \rangle
$$
$$
+ L \sum_{t \in [T]} \alpha_t^2 \|\overline{z}^t\|^2 + \frac{L^2}{2} \sum_{t \in [T]} \frac{\alpha_t}{n} \sum_{i \in [n]} \|x_i^t - \overline{x}^t\|^2. \quad (4)
$$

We use Lemma 3.3 to control the last term on the RHS, while to deal with $\sum_{t \in [T]} \alpha_t \langle \nabla f(\overline{x}^t), \overline{z}^t \rangle$ and remove the need for bounded gradients, we use Lemma 3.2 and the "offset trick", e.g., (Li & Orabona, 2020; Liu et al., 2023a; Armacki et al., 2026a), i.e., we subtract $\sum_{t \in [T]} \frac{9\sigma^2 \alpha_t^2 \|\nabla f(\overline{x}^t)\|^2}{4n}$ from both sides of (4), ensuring that the effects of the inner product is absorbed by the left-hand side. The rest of the proof relies on Lemma 3.2, some technical results and a careful selection of the step-size, see Appendix F for details.

*Proof sketch of Theorem 3.7.* Using Lemma 3.1, properties of strongly convex functions, and defining $F^t := n(t + t_0)\big(f(\overline{x}^t) - f^\star\big)$ and $A_t := \alpha_t(t + t_0 + 1)$, we get

$$
F^{t+1} \leq (1 - \alpha_t \mu) \frac{t + t_0 + 1}{t + t_0} F^t - A_t \langle \nabla f(\overline{x}^t), \overline{z}^t \rangle
$$
$$
+ \alpha_t A_t n L \|\overline{z}^t\|^2 + \frac{A_t L^2}{2} \sum_{i \in [n]} \|x_i^t - \overline{x}^t\|^2. \quad (5)
$$

We use (5), Lemma 3.5, and a careful analysis to show that the MGF of $F^t$ satisfies the conditions of Lemma 3.6, which we can then apply. The proof is completed by using these results to show that both optimality and consensus gaps are small with high probability, see Appendix G for details.

*Comparison with centralized works.* Compared to works studying HP convergence of centralized **SGD**, e.g., (Harvey et al., 2019; Bajović et al., 2023; Liu et al., 2023a; Liu & Zhou, 2024), we make several contributions. First, we face the challenge of controlling the MGF of the consensus gap, stemming from the decentralized nature of the algorithm. To that end, we provide Lemma 3.3 for non-convex, as well as the improved Lemma 3.5 for strongly convex costs, which removes the bounded heterogeneity condition **(A5)**. Next, we establish Lemma 3.2, which shows the variance reduction benefits of decentralized methods in the HP sense and is of independent interest when studying HP guarantees in decentralized settings. Finally, we provide a novel result on the MGF of the optimality gap of strongly convex costs (more broadly, "almost decreasing" processes), in the form of Lemma 3.6. This result provides a tighter control on the MGF compared to state-of-the-art bounds in (Harvey et al., 2019; Bajović et al., 2023; Liu & Zhou, 2024) and is, in addition to Lemma 3.2, paramount to ensuring linear speed-up is achieved for strongly convex costs. In particular, following the same approach as in (Harvey et al., 2019; Bajović et al., 2023), one can show that the MGF of the quantity of interest is uniformly bounded, i.e., that for some $B > 0$ and all $t \geq 1$ and $\nu \in (0, B]$

$$
\mathbb{E}[\exp(\nu F^t)] \leq \exp\left(\frac{\nu}{B}\right), \quad (6)
$$

where $F^t = n(t+t_0)\big(f(\overline{x}^t) - f^\star\big)$. Using (6) and Markov's inequality, setting $\nu = B$, one can show that

$$
\mathbb{P}\left(f(\overline{x}^t) - f^\star \leq \frac{\log(1/\delta) + 1}{nB(t + t_0)}\right) \geq 1 - \delta, \quad (7)
$$

losing the linear speed-up, as $B = \mathcal{O}(1/n)$, which we show in Appendix J. On the other hand, Lemma 3.6 provides a much sharper bound on the MGF, ensuring that linear speed-up is achieved. Similar observations hold with respect to (Liu & Zhou, 2024), see Appendix J for details.

*On the dimension dependence.* Our bounds contain a factor of $\log(d)$, introducing a mild dependence on the problem dimension not present in either MSE results in (Koloskova et al., 2020), or HP bounds in (Lu et al., 2024). This dependence stems from Lemma 3.2, where we show that variance-reduction is achieved in the HP sense, with $\log(d)$ appearing in the exponent bounding the MGF of $\overline{z}^t$. While mild, it is not clear if the dimension dependence for the average of sub-Gaussian vectors can be fully removed, noted also by Jin et al. (2019). For a further discussion on sub-Gaussian vectors and dimension dependence, see Appendix L.

# 5. Numerical Experiments

In this section we provide numerical experiments to evaluate our theoretical results. We consider two sets of experiments,

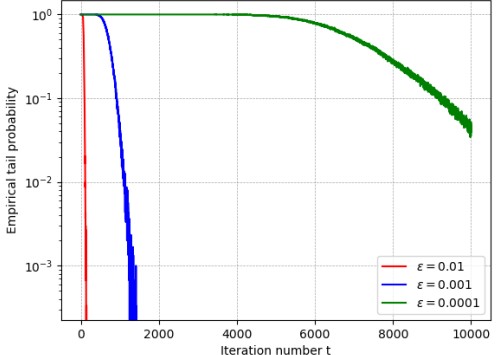
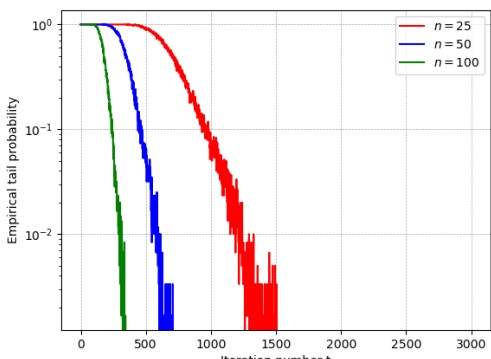

*Figure 1.* Performance of **DSGD** in the HP sense. The left figure presents performance over a fixed network of $n = 30$ users, for varying values of threshold $\varepsilon = \{10^{-2}, 10^{-3}, 10^{-4}\}$. The right figure presents performance for fixed value of threshold $\varepsilon = 10^{-3}$ and varying network size, with $n = \{25, 50, 100\}$ users. We can see that the empirical tail probability consistently decays exponentially fast, with faster decay over larger networks, indicating that linear speed-up is achieved.

strongly convex costs on synthetic data and non-convex costs on real data. For full details on the experimental setup and additional results, the reader is referred to Appendix H.

*Synthetic data.* We consider a strongly convex quadratic problem, where the local cost of each user is given by $f_i(x) = \frac{1}{2}x^\top A_i x + b_i^\top x$, where $A_i \in \mathbb{R}^{d \times d}$ is positive definite, making each $f_i$ strongly convex with unbounded gradients. The users communicate over an undirected network $G$ corresponding to a random Erdős–Rényi graph with connectivity parameter $p = 0.8$, while the weight matrix $W \in \mathbb{R}^{n \times n}$ is computed using the Metropolis-Hastings weight scheme, e.g., (Xiao & Boyd, 2004). When queried by user $i \in [n]$, the $\mathcal{SFO}$ returns $g_i^t = A_i x_i^t + b_i + z_i^t$, where $z_i^t \in \mathbb{R}^d$ is a zero-mean Gaussian random vector, making the noise consistent with assumption **(A4)**. We use the **DSGD** method outlined in Algorithm 1, with step-size $\alpha_t = \frac{1}{t+1}$ and shared initialization $x_i^1 = 0$, for all $i \in [n]$. We aim to test two facets of our theory: (i) *exponentially decaying tails* - whether the tail probability decays at an exponential scale and (ii) *linear speed-up* - whether the tail probability decays faster as the number of users increases. To measure the performance, we use the empirical tail probability

$$\mathbb{P}_{n,\varepsilon}^t = \frac{1}{R} \sum_{r \in [R]} \mathbb{I}\left(\frac{1}{n}\sum_{i \in [n]} \|x_i^{t,r} - x^\star\|^2 > \varepsilon\right),$$

where $x_i^{t,r} \in \mathbb{R}^d$ is the model of user $i$ in iteration $t$ and run $r$, with $x^\star = \arg\min_{x \in \mathbb{R}^d} f(x)$ being the solution of the global problem and $\mathbb{I}(A)$ being the indicator of event $A$. The results are presented in Figure 1. The left plot presents the tail decay rate for a fixed network of $n = 30$ users and varying values of accuracy threshold, given by $\varepsilon = \{10^{-2}, 10^{-3}, 10^{-4}\}$, while the right plot presents the tail decay for a fixed value of accuracy threshold $\varepsilon = 10^{-3}$ and varying number of users $n = \{25, 50, 100\}$. We can

see from the two plots that: (i) the tail probability induced by **DSGD** consistently decays at an exponential rate, for all values of $\varepsilon$ and (ii) the decay is faster for larger networks and linear speed-up is achieved, as predicted by our theory.

*Real data.* We consider a non-convex logistic regression problem, where the local cost of each user is given by $f_i(x) = \frac{1}{m_i}\sum_{r \in [m_i]} \log\left(1 + \exp(-y_{i,r}\langle h_{i,r}, x\rangle)\right) + \eta \sum_{k \in [d]} \frac{[x]_k^2}{1+[x]_k^2}$, where $h_{i,r} \in \mathbb{R}^d$ and $y_{i,r} \in \{+1, -1\}$ are the feature vector and associated label, $\eta > 0$ is a user-specified penalty parameter, while $[x]_k$ is the $k$-th component of vector $x$. We use the "mushroom", "a9a" and "ijcnn1" datasets from the LIBSVM library (Chang & Lin, 2011). To measure the performance, we evaluate the empirical tail probability with respect to the averaged gradient norm-squared, i.e., we compute $\mathbb{P}_{n,\epsilon}^t = \frac{1}{R}\sum_{r \in [R]} \mathbb{I}\left(G_n^{t,r} > \epsilon\right)$, where $G_n^{t,r} = \frac{1}{nt}\sum_{\tau \in [t]}\sum_{i \in [n]}\|\nabla f(x_i^{\tau,r})\|^2$. Due to space constraints, the reader is kindly referred to Appendix H for full results and further details.

## 6. Conclusion

We studied convergence in HP of a variant of **DSGD** under light-tailed noise, showing that it is guaranteed to converge in the HP sense under the same conditions on the cost as in the MSE sense, achieving order-optimal rates and linear speed-up for both non-convex and strongly convex costs. Compared to existing works, we relax strong assumptions like uniformly bounded gradients and asymptotically vanishing noise, while simultaneously showing improved rates and sharper transient times than the ones obtained from MSE rates. Future work includes extending our results to costs satisfying the PŁ condition, incorporating bias-correction mechanisms like GT and exact diffusion to remove bounded heterogeneity, and considering heavy-tailed noise.

## Acknowledgements

The authors would like to thank Haoyuan Cai (EPFL) for help with some of numerical experiments and the anonymous reviewers for their useful feedback.

## Impact Statement

This paper presents work whose goal is to advance the field of Machine Learning. There are many potential societal consequences of our work, none which we feel must be specifically highlighted here.

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

# A. Introduction

Appendix contains results omitted from the main body. Section B discusses the differences between MSE and HP guarantees, Section C collects some important facts used in our proofs, Section D provides some technical results, Section E defines some notions used in the analysis, Sections F and G provide proofs for non-convex and strongly convex costs, respectively, Section H provides additional numerical results, Section I derives the transient times, Section J provides a discussion on the novel results needed for achieving linear speed-up, Section K discusses the extension to time-varying networks, Section L contains further discussion on dimension dependence in our bounds and the notion of sub-Gaussianity used in our work, while Section M provides additional results for strongly convex costs under a more general step-size schedule.

# B. On MSE and HP Results

Although both HP and MSE results establish convergence, the nature of these guarantees is very different. In particular, MSE results quantify the *average behaviour across many runs* of an algorithm, while HP results quantify the behaviour of *an individual run*. As mentioned briefly in the introduction, this distinction is very important in huge-scale applications like LLM training, where it is often impossible to perform more than a single training run, both resource- and time-wise. For example, Narayanan et al. (2021) estimate that the GPT-3 model with 175 billion parameters was trained for 34 days, while the variant with 1 trillion parameters can be trained for approximately 84 days. This is further exacerbated by phenomena like heavy-tailed noise, frequently observed during training of deep learning models and transformers, e.g., (Zhang et al., 2020; Simsekli et al., 2019), which can cause the performance of an individual run to significantly deviate from the average performance. Hence, strong guarantees with respect to an individual run are very important in such applications. While MSE results can be used to provide guarantees with respect to an individual run, they can be quite loose compared to HP bounds. In particular, if we have a MSE bound of the form $\mathbb{E}[X_t^2] = 1/t$, Chebyshev's inequality implies $\mathbb{P}(X_t^2 > 1/\delta t) \leq \delta$, which shows an inversely proportional dependence on the confidence level $\delta \in (0, 1)$. On the other hand, HP results establish guarantees of the type $\mathbb{P}(X_t^2 > \log(1/\delta)/t) \leq \delta$, having a much milder, logarithmic dependence on $1/\delta$. Equivalently, for any threshold $\epsilon > 0$, MSE + Chebyshev implies $\mathbb{P}(X_t^2 > \epsilon) \leq 1/t\epsilon$, while HP gives $\mathbb{P}(X_t^2 > \epsilon) \leq \exp(-t\epsilon)$, providing a much sharper bound on the tail probability. Since our goal is to show that $X_t^2 \leq \epsilon$ (e.g., if $X_t = \|x^t - x^\star\|$, or if $X_t = \min_{k \in [t]} \|\nabla f(x^k)\|$), then $\mathbb{P}(X_t^2 > \epsilon)$ represents the *probability of failure*, with MSE + Chebyshev implying that the probability of failure decays polynomially in $t$, while HP bound implies that the probability of failure decays exponentially fast in $t$, giving much stronger guarantees on the convergence of an individual run.

# C. Useful Inequalities

In this section we outline some well-known inequalities and results used in our proofs. We start with Jensen's inequality for convex/concave functions.

**Proposition C.1** (Jensen's inequality). *Let $X \in \mathbb{R}$ be an integrable random variable. Then, for any convex function $h : \mathbb{R} \mapsto \mathbb{R}$, we have*

$$h(\mathbb{E}[X]) \leq \mathbb{E}[h(X)].$$

*Moreover, if $h$ is concave, the reverse inequality holds, i.e., we then have*

$$\mathbb{E}[h(X)] \leq h(\mathbb{E}[X]).$$

**Proposition C.2** (Cauchy-Schwartz inequality). *For any $a, b \in \mathbb{R}^d$, we have*

$$|\langle a, b \rangle| \leq \|a\| \|b\|.$$

As a consequence of the Cauchy-Schwartz inequality, we have the following result.

**Proposition C.3** (Young's inequality). *For any $a, b \in \mathbb{R}$ and any $\epsilon > 0$, we have*

$$ab \leq \frac{\epsilon a^2}{2} + \frac{b^2}{2\epsilon}.$$

*Furthermore, for any $\theta > 0$, we have*

$$(a + b)^2 \leq (1 + \theta)a^2 + (1 + \theta^{-1})b^2.$$

Young's inequality is also known as the Peter-Paul inequality.

**Proposition C.4** (Hölder's inequality)**.** *For any random variables $X, Y \in \mathbb{R}$ and any $p, q \in [1, \infty]$, such that $\frac{1}{p} + \frac{1}{q} = 1$, we have*

$$\mathbb{E}|XY| \leq \sqrt[p]{\mathbb{E}|X|^p} \sqrt[q]{\mathbb{E}|Y|^q}.$$

The coefficients $p, q \in [1, \infty]$ are known as Hölder coefficients. Note that Hölder's inequality recovers Cauchy-Schwartz inequality for $p = q = 2$. We have the following useful consequence of Hölder's inequality.

**Proposition C.5.** *For any $n \in \mathbb{N}$ and random variables $X_i \in \mathbb{R}$, $i \in [n]$, we have*

$$\mathbb{E}\Big[ \prod_{i \in [n]} |X_i| \Big] \leq \prod_{i \in [n]} \sqrt[n]{\mathbb{E}|X_i|^n}.$$

Next, we state a useful result from (Xin et al., 2021).

**Proposition C.6** ((Xin et al., 2021), Lemma 21)**.** *For any $c, t_0 > 0$ and $0 \leq a \leq b$, we have*

$$\prod_{k=a}^{b} \Big(1 - \frac{c}{k + t_0}\Big) \leq \frac{(a + t_0)^c}{(b + 1 + t_0)^c}.$$

The following result states some important consequences of conditions **(A2)** and **(A3)**, see, e.g., (Bertsekas & Tsitsiklis, 2000; Nesterov, 2018).

**Proposition C.7.** *Let **(A3)** hold. Then, for any $i \in [n]$ and $x, y \in \mathbb{R}^d$, the following statements are true.*

1.  $f_i(x) \leq f_i(y) + \langle \nabla f_i(y), x - y \rangle + \frac{L}{2}\|x - y\|^2$.

2.  *$f$ has L-Lipschitz continuous gradients.*

3.  $f(x) \leq f(y) + \langle \nabla f(y), x - y \rangle + \frac{L}{2}\|x - y\|^2$.

4.  *If in addition **(A2)** holds, then for any $x \in \mathbb{R}^d$, we have $\|\nabla f(x)\|^2 \leq 2L(f(x) - f^\star)$.*

Next, we state some important consequences of **(A6)**, see, e.g., (Nesterov, 2018).

**Proposition C.8.** *Let (A6) hold. Then, the following are true, for all $i \in [n]$ and $x \in \mathbb{R}^d$.*

1.  $\nabla^2 f_i(x) \succeq \mu I_d$.

2.  *$f$ is $\mu$-strongly convex.*

3.  $\|\nabla f(x)\|^2 \geq 2\mu(f(x) - f^\star)$.

4.  $f(x) - f^\star \geq \frac{\mu}{2}\|x - x^\star\|^2$.

Finally, we provide a useful martingale concentration result from (Jin et al., 2019).

**Proposition C.9.** *Let $X_1, \ldots, X_n \in \mathbb{R}^d$ be random vectors and denote by $\mathcal{F}_n = \sigma(\{X_1, \ldots, X_n\})$ the corresponding filtration. If for all $i \in [n]$, the vector $X_i$ satisfies $\mathbb{E}[X_i \mid \mathcal{F}_{i-1}] = 0$ and*

$$\mathbb{P}\big(\|X_i\| > \epsilon \mid \mathcal{F}_{i-1}\big) \leq 2 \exp\Big(-\frac{t^2}{2\sigma_i^2}\Big),$$

*for some $\mathcal{F}_{i-1}$-measurable $\sigma_i > 0$ and all $\epsilon > 0$, then for any $\delta \in (0, 1)$, we have*

$$\mathbb{P}\Big(\|\sum_{i \in [n]} X_i\|^2 > 2\sqrt{2} \sum_{i \in [n]} \sigma_i^2 \log(2d/\delta)\Big) \leq \delta.$$

## D. Technical Results

In this section we prove Lemmas 3.2, 3.6 and provide another technical result used in the proofs. For the reader's convenience, we restate Lemmas 3.2 and 3.6 below.

**Lemma 3.2.** *If* **(A3)** *holds, then the following are true for any* $t \geq 1$, $i \in [n]$ *and* $\mathcal{F}_t$-*measurable* $v \in \mathbb{R}^d$.

1. $\mathbb{E}\left[\exp\left(\langle v, z_i^t \rangle\right) \mid \mathcal{F}_t\right] \leq \exp\left(\frac{3\sigma_i^2 \|v\|^2}{4}\right)$.

2. $\mathbb{E}\left[\exp\left(\langle v, \overline{z}^t \rangle\right) \mid \mathcal{F}_t\right] \leq \exp\left(\frac{3\sigma^2 \|v\|^2}{4n}\right)$.

3. $\mathbb{E}\left[\exp\left(\frac{n\|\overline{z}^t\|^2}{15\sigma^2}\right) \mid \mathcal{F}_t\right] \leq 2d\exp(1)$.

*Proof.* 1. To prove the first property, we follow steps similar to those from Lemma 1 in (Li & Orabona, 2020). Let $y_i = \frac{z_i^t}{\sigma_i}$ and note that $\mathbb{E}\left[\exp\left(\|y_i\|^2\right) \mid \mathcal{F}_t\right] \leq \exp(1)$, from **(A3)**. Assume first that $v \in \mathbb{R}^d$ is such that $\|v\| \leq \frac{4}{3}$. Using the inequality $\exp(a) \leq a + \exp(9a^2/16)$, which holds for any $a \in \mathbb{R}$, we then have

$$
\begin{aligned}
\mathbb{E}\left[\exp\left(\langle v, y_i \rangle\right) \mid \mathcal{F}_t\right] &\leq \mathbb{E}\left[\langle v, y_i \rangle + \exp\left(\frac{9\langle v, y_i \rangle^2}{16}\right) \mid \mathcal{F}_t\right] \overset{(a)}{=} \mathbb{E}\left[\exp\left(\frac{9\langle v, y_i \rangle^2}{16}\right) \mid \mathcal{F}_t\right] \\
&\overset{(b)}{\leq} \mathbb{E}\left[\exp\left(\frac{9\|v\|^2\|y_i\|^2}{16}\right) \mid \mathcal{F}_t\right] \overset{(c)}{\leq} \left(\mathbb{E}\left[\exp\left(\|y_i\|^2\right) \mid \mathcal{F}_t\right]\right)^{9\|v\|^2/16} \\
&\overset{(d)}{\leq} \exp\left(\frac{9\|v\|^2}{16}\right) \leq \exp\left(\frac{3\|v\|^2}{4}\right),
\end{aligned}
$$

where $(a)$ follows from **(A3)** and since $v$ is $\mathcal{F}_t$-measurable, $(b)$ follows from Proposition C.2, $(c)$ follows from the fact that $\frac{9\|v\|^2}{16} \leq 1$ and Proposition C.1, while $(d)$ follows from $\mathbb{E}[\exp(\|y_i\|^2) \mid \mathcal{F}_t] \leq \exp(1)$. Next, if $\|v\| > \frac{4}{3}$, we have

$$
\begin{aligned}
\mathbb{E}[\exp(\langle v, y_i \rangle) \mid \mathcal{F}_t] &\leq \exp\left(\frac{3\|v\|^2}{8}\right) \mathbb{E}\left[\exp\left(\frac{2\|y_i\|^2}{3}\right) \mid \mathcal{F}_t\right] \\
&\leq \exp\left(\frac{2}{3} + \frac{3\|v\|^2}{8}\right) \leq \exp\left(\frac{3\|v\|^2}{4}\right),
\end{aligned}
$$

where the first inequality follows by applying Proposition C.3 with $\epsilon = \frac{4}{3}$ and the fact that $v$ is $\mathcal{F}_t$-measurable, the second follows from Proposition C.1 and $\mathbb{E}[\exp(\|y_i\|^2) \mid \mathcal{F}_t] \leq \exp(1)$, while the third inequality follows from the fact that $\frac{2}{3} < \frac{3\|v\|^2}{8}$, since $\|v\| > \frac{4}{3}$. Combining both cases, we get $\mathbb{E}[\exp(\langle v, y_i \rangle) \mid \mathcal{F}_t] \leq \exp\left(\frac{3\|v\|^2}{4}\right)$, for any $\mathcal{F}_t$-measurable vector $v \in \mathbb{R}^d$. The proof is completed by applying this inequality to $\langle v, z_i^t \rangle = \langle \sigma_i v, y_i \rangle$.

2. Recall that $\overline{z}^t = \frac{1}{n}\sum_{i\in[n]} z_i^t$ and $\sigma^2 = \frac{1}{n}\sum_{i\in[n]} \sigma_i^2$. We then have, for any $\mathcal{F}_t$-measurable $v \in \mathbb{R}^d$

$$
\begin{aligned}
\mathbb{E}[\exp(\langle v, \overline{z}^t \rangle)] = \mathbb{E}\left[\exp\left(\frac{1}{n}\sum_{i\in[n]} \langle v, z_i^t \rangle\right)\right] &\overset{(a)}{=} \prod_{i\in[n]} \mathbb{E}\left[\exp\left(\left\langle \frac{v}{n}, z_i^t \right\rangle\right)\right] \\
&\overset{(b)}{\leq} \prod_{i\in[n]} \exp\left(\frac{3\sigma_i^2 \|v\|^2}{4n^2}\right) \overset{(c)}{=} \exp\left(\frac{3\sigma^2 \|v\|^2}{4n}\right),
\end{aligned}
$$

where $(a)$ follows from the fact that, conditioned on $\mathcal{F}_t$, the noise across users is independent according to **(A3)**, $(b)$ follows from the first part of the proof, while $(c)$ follows from the definition of $\sigma^2$.

3. From **(A4)** we know that conditioned on $\mathcal{F}_t$, the vectors $z_1^t, \ldots, z_n^t$ are independent and zero-mean. Moreover, for each

$i \in [n]$ and any $\epsilon > 0$, we have

$$\mathbb{P}\big(\|z_i^t\| > \epsilon \mid \mathcal{F}_t\big) = \mathbb{P}\bigg( \exp\bigg( \frac{\|z_i^t\|^2}{2\sigma_i^2} \bigg) > \exp\bigg( \frac{\epsilon^2}{2\sigma_i^2} \bigg) \, \bigg| \, \mathcal{F}_t \bigg) \overset{(a)}{\leq} \exp\bigg( -\frac{\epsilon^2}{2\sigma_i^2} \bigg) \mathbb{E}\bigg[ \exp\bigg( \frac{\|z_i^t\|^2}{2\sigma_i^2} \bigg) \, \bigg| \, \mathcal{F}_t \bigg]$$

$$\overset{(b)}{\leq} \exp\bigg( -\frac{\epsilon^2}{2\sigma_i^2} \bigg) \bigg( \mathbb{E}\bigg[ \exp\bigg( \frac{\|z_i^t\|^2}{\sigma_i^2} \bigg) \, \bigg| \, \mathcal{F}_t \bigg] \bigg)^{1/2} \overset{(c)}{\leq} \exp\bigg( \frac{1}{2} - \frac{\epsilon^2}{2\sigma_i^2} \bigg) \overset{(d)}{\leq} 2\exp\bigg( -\frac{\epsilon^2}{2\sigma_i^2} \bigg),$$

where $(a)$ follows from Markov's inequality, $(b)$ follows from Proposition C.1, in $(c)$ we use **(A4)**, while $(d)$ follows form the fact that $\exp(1/2) \leq 2$. Therefore, vectors $z_1^t, \ldots, z_n^t$ satisfy the conditions of Proposition C.9 with respect to $\mathcal{F}_t$, and it follows that, for any $\delta \in (0, 1)$

$$\mathbb{P}\bigg( \|\bar{z}^t\|^2 > \frac{2\sqrt{2}\sigma^2 \log(2d/\delta)}{n} \bigg) \leq \delta,$$

or equivalently, for any $\epsilon > 0$, we have

$$\mathbb{P}\big(\|\bar{z}^t\|^2 > \epsilon\big) \leq 2d\exp\bigg( -\frac{n\epsilon}{2\sigma^2\sqrt{2}} \bigg) \leq 2d\exp\bigg( -\frac{n\epsilon}{3\sigma^2} \bigg). \tag{8}$$

Next, define $Y = \frac{\sqrt{n}\|\bar{z}^t\|}{\sigma\sqrt{3}}$. Using the layer-cake formula for the expectation of a non-negative random variable, we then have, for any integer $p \geq 1$

$$\mathbb{E}\big[\|Y\|^{2p}\big] = \int_0^\infty \mathbb{P}\big(\|Y\|^{2p} > y\big) dy \overset{(a)}{=} p \int_0^\infty s^{p-1} \mathbb{P}\big(\|Y\|^2 > s\big) ds$$

$$= p \int_0^\infty s^{p-1} \mathbb{P}\bigg( \|\bar{z}^t\|^2 > \frac{3\sigma^2 s}{n} \bigg) ds \overset{(b)}{\leq} 2dp \int_0^\infty s^{p-1} \exp(-s) ds \overset{(c)}{\leq} 2dp^{p+1}, \tag{9}$$

where $(a)$ follows by introducing the substitution $y = s^p$, $(b)$ follows from (8), while $(c)$ follows from the definition of $\Gamma$ function and the fact that $\Gamma(p) \leq p^p$. For any $c > 0$, we then have

$$\mathbb{E}\bigg[ \exp\bigg( \frac{cn\|\bar{z}^t\|^2}{3\sigma^2} \bigg) \bigg] = 1 + \sum_{k=1}^\infty \frac{c^k \mathbb{E}\big[\|Y\|^{2k}\big]}{k!} \overset{(i)}{\leq} 1 + \sum_{k=1}^\infty \frac{2dk(ck)^k}{k!} \overset{(ii)}{\leq} 2d\bigg( 1 + \frac{1}{e} \sum_{k=1}^\infty k(ec)^k \bigg),$$

where $(i)$ follows from (9), while $(ii)$ follows from $k! \geq e(k/e)^k$. Next, note that the sequence $\sum_{k=1}^\infty kp^k$ converges if $p < 1$, with $\sum_{k=1}^\infty kp^k = \frac{p}{(1-p)^2}$, therefore, choosing $c < \frac{1}{e}$ in the relation above, we get

$$\mathbb{E}\bigg[ \exp\bigg( \frac{cn\|\bar{z}^t\|^2}{3\sigma^2} \bigg) \bigg] \leq 2d\bigg( 1 + \frac{c}{(1-ec)^2} \bigg). \tag{10}$$

It can be verified that $\frac{c}{(1-ec)^2} \leq e - 1$ for any $c \leq \frac{e-1}{e^2}$. Choosing $c = \frac{1}{5}$ and plugging into (10) completes the proof.

$\square$

We next restate Lemma 3.6.

**Lemma 3.6.** *Let $\{X^t\}_{t \geq 2}$ be a sequence of random variables initialized by a deterministic $X^1 > 0$, such that, for some $M \in \mathbb{N}$, $a, t_0, C_i > 0$, $i \in [M]$ and every $t \geq 1$*

$$\mathbb{E}[\exp(X^{t+1})] \leq \mathbb{E}\bigg[ \exp\bigg( \bigg(1 - \frac{a}{t+t_0}\bigg) X^t + \sum_{i \in [M]} \frac{C_i}{(t+t_0)^i} \bigg) \bigg]. \tag{11}$$

*If $a \in (1, 2]$ and $t_0 \geq a$, we then have*

$$\mathbb{E}[\exp(X^{t+1})] \leq \exp\bigg( \frac{(t_0+1)^a X_1}{(t+1+t_0)^a} + \frac{2^a C_1}{a} + \frac{2^a C_3 \log(t+1+t_0)}{(t+1+t_0)^a} \bigg)$$

$$\times \exp\bigg( \frac{2^a C_2/(a-1)}{t+1+t_0} + \sum_{j=4}^M \frac{2^a t_0^{3-j} C_j}{(j-3)(t+1+t_0)^a} \bigg).$$

*Proof.* Starting from (11), taking the logarithm, defining $Y_t := \log \mathbb{E}[\exp(X^t)]$ and $b_t = 1 - \frac{a}{t+t_0}$, we then have

$$Y^{t+1} \leq \sum_{i \in [M]} \frac{C_i}{(t+t_0)^i} + \log \mathbb{E}[\exp(b_t X^t)] \leq \sum_{i \in [M]} \frac{C_i}{(t+t_0)^i} + \log \left[ \left( \mathbb{E}[\exp(X^t)] \right)^{b_t} \right]$$

$$= b_t Y^t + \sum_{i \in [M]} \frac{C_i}{(t+t_0)^i}, \tag{12}$$

where the second inequality follows from the fact that $b_t \in (0,1)$ and Proposition C.1. Unrolling the recursion (12) and noting that $Y_1 = X_1$, since $X_1 > 0$ is deterministic, we get

$$Y^{t+1} \leq X^1 \prod_{k \in [t]} b_k + \sum_{i \in [M]} C_i \sum_{k \in [t]} \frac{1}{(k+t_0)^i} \prod_{s=k+1}^{t} b_s$$

$$\leq \frac{(t_0+1)^a X^1}{(t+1+t_0)^a} + \sum_{i \in [M]} C_i \sum_{k \in [t]} \frac{1}{(k+t_0)^i} \times \frac{(k+1+t_0)^a}{(t+1+t_0)^a}$$

$$\leq \frac{(t_0+1)^a X^1}{(t+1+t_0)^a} + \sum_{i \in [M]} \frac{2^a C_i}{(t+1+t_0)^a} \sum_{k \in [t]} (k+t_0)^{a-i}, \tag{13}$$

where the second inequality follows from Proposition C.6, while the third inequality follows from the fact that $\left( \frac{k+1+t_0}{k+t_0} \right)^a \leq 2^a$. We now proceed to analyze $\sum_{k \in [t]} (k+t_0)^{a-i}$, for different values of $i \in [M]$. Using Darboux sums and the fact that $a \in (1,2]$, it can be readily verified that

$$\sum_{k \in [t]} (k+t_0)^{a-i} \leq \begin{cases} \frac{(t+1+t_0)^a}{a}, & i = 1 \\ \frac{(t+1+t_0)^{a-1}}{a-1}, & i = 2 \\ \ln(t+t_0+1), & i = 3 \\ \frac{t_0^{3-i}}{i-3}, & i \geq 4 \end{cases}. \tag{14}$$

Plugging (14) into (13), we get

$$Y^{t+1} \leq \frac{(t_0+1)^a X^1}{(t+1+t_0)^a} + \frac{2^a C_1}{a} + \frac{2^a C_2}{(a-1)(t+t_0+1)} + \frac{2^a C_3 \log(t+t_0+1)}{(t+1+t_0)^a} + \sum_{j=4}^{M} \frac{2^a t_0^{3-j} C_j}{(j-3)(t+1+t_0)^a}.$$

Taking the exponent on both sides completes the proof. $\qquad \square$

To complete this section, we provide a useful technical result.

**Lemma D.3.** *Let $\lambda \in [0,1)$ and $\alpha_t = \frac{a}{(t+t_0)^c}$, where $a, t_0 > 0$ and $c \geq 1/2$. If $t_0 \geq \frac{2c-1+\lambda}{1-\lambda}$, then for any $t \geq 1$, we have*

$$\sum_{k \in [t]} \alpha_k \lambda^{t-k} \leq \frac{3\alpha_t}{1-\lambda}.$$

*Proof.* Using the definition of $\alpha_t$, we note that

$$\sum_{k \in [t]} \alpha_k \lambda^{t-k} = \alpha_t \sum_{k \in [t]} \frac{\alpha_k}{\alpha_t} \lambda^{t-k} = \alpha_t \sum_{k \in [t]} \lambda^{t-k} \left( \frac{t+t_0}{k+t_0} \right)^c = \alpha_t \sum_{k \in [t]} \lambda^{t-k} \left( 1 + \frac{t-k}{k+t_0} \right)^c.$$

Next, denote by $\widetilde{\lambda} := 1 - \lambda \in (0,1]$ and use the substitution $s = t - k$, to get

$$\sum_{k \in [t]} \alpha_k \lambda^{t-k} = \alpha_t \sum_{s=0}^{t-1} (1-\widetilde{\lambda})^s \left( 1 + \frac{s}{t-s+t_0} \right)^c \leq \alpha_t \sum_{s=0}^{t-1} \exp\left( -\widetilde{\lambda}s + \frac{cs}{t-s+t_0} \right)$$

$$\leq \alpha_t \sum_{s=0}^{t-1} \exp\left( -s\left( \widetilde{\lambda} - \frac{c}{1+t_0} \right) \right) \leq \alpha_t \sum_{s=0}^{t-1} \exp\left( -\frac{\widetilde{\lambda}s}{2} \right), \tag{15}$$

where we used $1 \pm x \le \exp(\pm x)$ in the first, the fact that $t - s \ge 1$ in the second and the choice of $t_0$ in the third inequality. Next, we use Darboux sums, to get

$$\sum_{s=0}^{t-1} \exp\left(-\frac{\widetilde{\lambda}s}{2}\right) = 1 + \sum_{s=1}^{t-1} \exp\left(-\frac{\widetilde{\lambda}s}{2}\right) \le 1 + \int_0^{t-1} \exp\left(-\frac{\widetilde{\lambda}s}{2}\right) ds \le 1 + \frac{2}{\widetilde{\lambda}} \le \frac{3}{\widetilde{\lambda}}. \tag{16}$$

Plugging (16) into (15) completes the proof. $\qquad\square$

## E. Analysis Setup

In this section we define some notation useful for the analysis. To begin, let $g_i^t := \nabla\ell(x_i^t; \xi_i^t)$ denote the stochastic gradient of user $i$ at time $t$. We can then represent the update rule (3) as

$$x_i^{t+1} = \sum_{j \in \mathcal{N}_i} w_{ij}(x_i^t - \alpha_t g_i^t). \tag{17}$$

It can immediately be seen that $z_i^t = g_i^t - \nabla f_i(x_i^t)$. Next, define the network average stochastic gradient $\overline{g}^t := \frac{1}{n}\sum_{i\in[n]} g_i^t$ and $\overline{\nabla} f_t := \frac{1}{n}\sum_{i\in[n]} \nabla f_i(x_i^t)$, which represents the network average of user's gradients evaluated at their local models. It can then be seen that $\overline{g}^t = \overline{\nabla} f_t + \overline{z}^t$. Combining the definition of the network average model, the fact that the weight matrix is doubly stochastic and (17), it follows that $\overline{x}^{t+1} = \overline{x}^t - \alpha_t\overline{g}^t$.

We now introduce some notation useful for the analysis of decentralized methods, see, e.g., (Nedić & Ozdaglar, 2009; Jakovetić, 2019; Xin et al., 2021). Let $\mathbf{x}^t := col(x_1^t, \ldots, x_n^t) \in \mathbb{R}^{nd}$ denote the column vector stacking users' local models. Using this notation, we can then represent the update rule (17) compactly as

$$\mathbf{x}^{t+1} = \mathbf{W}(\mathbf{x}^t - \alpha_t\mathbf{g}^t), \tag{18}$$

where $\mathbf{W} = W \otimes I_d \in \mathbb{R}^{nd\times nd}$, $\otimes$ denotes the Kronecker product and $I_d \in \mathbb{R}^{d\times d}$ denotes the $d$-dimensional identity matrix, while $\mathbf{g}^t := col(g_1^t, \ldots, g_n^t)$. Next, define the matrix $J := \frac{1}{n}\mathbf{1}_n\mathbf{1}_n^\top \in \mathbb{R}^{n\times n}$, where $\mathbf{1}_n \in \mathbb{R}^n$ is the vector of all ones. The matrix $J$ represent the "ideal" consensus matrix, where all users can communicate with each other.[11] The interaction between matrices $W$ and $J$ represents an important part of any decentralized algorithm and we now list some known properties, see, e.g., (Jakovetić, 2019; Xin et al., 2021) and references therein.

**Proposition E.1.** *Let* **(A1)** *hold. Then, the following are true.*

1. $W\mathbf{1}_n = J\mathbf{1}_n = \mathbf{1}_n$.

2. $\lambda := \|W - J\| \in [0, 1)$.

3. $WJ = JW = J$.

Next, define $\overline{\mathbf{x}}^t := \mathbf{1}_n \otimes \overline{x}^t \in \mathbb{R}^{nd}$, $\mathbf{J} := J \otimes I_d \in \mathbb{R}^{nd\times nd}$ and note that $\overline{\mathbf{x}}^t = \mathbf{J}\mathbf{x}^t$. Combined with (18), it follows that $\overline{\mathbf{x}}^{t+1} = \overline{\mathbf{x}}^t - \alpha_t\overline{\mathbf{g}}^t$. Denoting by $\widetilde{\mathbf{W}} := \mathbf{W} - \mathbf{J}$, it follows from (18) that

$$\mathbf{x}^{t+1} - \overline{\mathbf{x}}^{t+1} = \mathbf{W}(\mathbf{x}^t - \alpha\mathbf{g}^t) - \mathbf{J}(\mathbf{x}^t - \alpha_t\mathbf{g}^t) = \widetilde{\mathbf{W}}(\mathbf{x}^t - \alpha\mathbf{g}^t) = \widetilde{\mathbf{W}}(\mathbf{x}^t - \overline{\mathbf{x}}^t - \alpha\mathbf{g}^t). \tag{19}$$

Finally, recall that we denote the unique global minima of strongly convex costs by $x^\star \in \mathbb{R}^d$. We define two related concepts, namely the column stacking of $x^\star$, i.e., $\mathbf{x}^\star := \mathbf{1}_n \otimes x^\star$ and the stacking of users' gradients evaluated at the global optima, i.e., $\nabla\mathbf{f}^\star := col(\nabla f_1(x^\star), \ldots, \nabla f_n(x^\star))$. Note that $\|\nabla\mathbf{f}^\star\|^2 = \sum_{i\in[n]}\|\nabla f_i(x^\star)\|^2$ is a useful measure of heterogeneity.

## F. Proofs for Non-convex Costs

In this section we prove Lemmas 3.1 and 3.3, as well as Theorem 3.4. For the reader's convenience, we restate the results below, starting with Lemma 3.1.

---

[11]This is equivalent to the client-server setup in terms of the update rule, as the server averages models from all clients in each iteration.

**Lemma 3.1.** *Let* **(A3)** *hold. If* $\alpha_t \leq \frac{1}{2L}$*, we have*

$$f(\overline{x}^{t+1}) \leq f(\overline{x}^t) - \frac{\alpha_t}{2}\|\nabla f(\overline{x}^t)\|^2 - \alpha_t\langle\nabla f(\overline{x}^t), \overline{z}^t\rangle + \alpha_t^2 L\|\overline{z}^t\|^2 + \frac{\alpha_t L^2}{2n}\sum_{i\in[n]}\|x_i^t - \overline{x}^t\|^2.$$

*Proof.* The proof follows similar steps as in, e.g., Lemma 3 from (Xin et al., 2021). Starting from Proposition C.7 and averaging across all costs $i \in [n]$, it readily follows that, for all $x, y \in \mathbb{R}^d$

$$f(x) \leq f(y) + \langle\nabla f(y), x - y\rangle + \frac{L}{2}\|x - y\|^2. \tag{20}$$

Setting $x = \overline{x}^{t+1}$ and $y = \overline{x}^t$ in (20), we get

$$\begin{aligned}
f(\overline{x}^{t+1}) &\leq f(\overline{x}^t) - \alpha_t\langle\nabla f(\overline{x}^t), \overline{g}_t\rangle + \frac{\alpha_t^2 L}{2}\|\overline{g}_t\|^2 \\
&\stackrel{(a)}{\leq} f(\overline{x}^t) - \alpha_t\langle\nabla f(\overline{x}^t), \overline{\nabla}f_t\rangle - \alpha_t\langle\nabla f(\overline{x}^t), \overline{z}^t\rangle + \alpha_t^2 L\|\overline{\nabla}f_t\|^2 + \alpha_t^2 L\|\overline{z}^t\|^2 \\
&\stackrel{(b)}{=} f(\overline{x}^t) - \frac{\alpha_t}{2}\|\nabla f(\overline{x}^t)\|^2 - (1 - 2\alpha_t L)\frac{\alpha_t}{2}\|\overline{\nabla}f_t\|^2 \\
&\quad + \frac{\alpha_t}{2}\|\overline{\nabla}f_t - \nabla f(\overline{x}^t)\|^2 - \alpha_t\langle\nabla f(\overline{x}^t), \overline{z}^t\rangle + \alpha_t^2 L\|\overline{z}^t\|^2 \\
&\stackrel{(c)}{\leq} f(\overline{x}^t) - \frac{\alpha_t}{2}\|\nabla f(\overline{x}^t)\|^2 + \frac{\alpha_t}{2}\|\overline{\nabla}f_t - \nabla f(\overline{x}^t)\|^2 - \alpha_t\langle\nabla f(\overline{x}^t), \overline{z}^t\rangle + \alpha_t^2 L\|\overline{z}^t\|^2,
\end{aligned} \tag{21, 22}$$

where $(a)$ follows by applying Proposition C.3 with $\theta = 1$, $(b)$ follows from the identity $\langle a, b\rangle = \frac{1}{2}\left(\|a\|^2 + \|b\|^2 - \|a - b\|^2\right)$, while $(c)$ follows from the fact that $\alpha_t \leq \frac{1}{2L}$. Recalling the definition of $\overline{\nabla}f_t$, we get

$$\|\overline{\nabla}f_t - \nabla f(\overline{x}^t)\|^2 = \left\|\frac{1}{n}\sum_{i\in[n]}\left[\nabla f_i(x_i^t) - \nabla f_i(\overline{x}^t)\right]\right\|^2 \leq \frac{L^2}{n}\sum_{i\in[n]}\|x_i^t - \overline{x}^t\|^2, \tag{23}$$

where we used Proposition C.1 and **(A3)** in the last inequality. Plugging (23) in (21) gives the desired result. $\square$

Next, we restate Lemma 3.3.

**Lemma 3.3.** *Let* **(A1)** *and* **(A5)** *hold. We then have, for any $t \geq 1$*

$$\begin{aligned}
\frac{1}{n}\sum_{i\in[n]}\|x_i^{t+1} - \overline{x}^{t+1}\|^2 &\leq 2\lambda^{2t}\Delta_x + \frac{4\lambda^2}{n(1-\lambda)}\sum_{k\in[t]}\alpha_k^2\lambda^{t-k}\sum_{i\in[n]}\|z_i^k\|^2 \\
&\quad + \frac{4\lambda^2 A^2}{1-\lambda}\sum_{k\in[t]}\alpha_k^2\lambda^{t-k} + \frac{4\lambda^2 B^2}{n(1-\lambda)}\sum_{k\in[t]}\alpha_k^2\lambda^{t-k}\sum_{i\in[n]}\|\nabla f(x_i^k)\|^2.
\end{aligned}$$

*Proof.* We start by noting that $\sum_{i\in[n]}\|x_i^{t+1} - \overline{x}^{t+1}\|^2 = \|\mathbf{x}^{t+1} - \overline{\mathbf{x}}^{t+1}\|^2$. Next, starting from (19) and unrolling the recursion, we get

$$\mathbf{x}^{t+1} - \overline{\mathbf{x}}^{t+1} = \widetilde{\mathbf{W}}(\mathbf{x}^t - \overline{\mathbf{x}}^t) - \alpha_t\widetilde{\mathbf{W}}\mathbf{g}^t = \ldots = \widetilde{\mathbf{W}}^t(\mathbf{x}^1 - \overline{\mathbf{x}}^1) - \sum_{k\in[t]}\alpha_k\widetilde{\mathbf{W}}^{t+1-k}\mathbf{g}^k,$$

Using Proposition E.1, it follows that

$$\|\mathbf{x}^{t+1} - \overline{\mathbf{x}}^{t+1}\| \leq \|\widetilde{\mathbf{W}}^t(\mathbf{x}^1 - \overline{\mathbf{x}}^1)\| + \sum_{k\in[t]}\alpha_k\|\widetilde{\mathbf{W}}^{t+1-k}\mathbf{g}^k\| \leq \lambda^t\|\mathbf{x}^1 - \overline{\mathbf{x}}^1\| + \lambda\sum_{k\in[t]}\alpha_k\lambda^{t-k}\|\mathbf{g}^k\|,$$

where we used the fact that $\|\widetilde{\mathbf{W}}^k\| = \|\widetilde{\mathbf{W}}\|^k$, for any integer $k \geq 0$. This readily implies

$$\begin{aligned}
\|\mathbf{x}^{t+1} - \overline{\mathbf{x}}^{t+1}\|^2 &\leq 2\lambda^{2t}\|\mathbf{x}^1 - \overline{\mathbf{x}}^1\|^2 + 2\lambda^2\left(\sum_{k\in[t]}\alpha_k\lambda^{t-k}\|\mathbf{g}_k\|\right)^2 \\
&\leq 2\lambda^{2t}\|\mathbf{x}^1 - \overline{\mathbf{x}}^1\|^2 + 2\lambda^2\sum_{s\in[t]}\lambda^{t-s}\sum_{k\in[t]}\alpha_k^2\lambda^{t-k}\|\mathbf{g}^k\|^2,
\end{aligned} \tag{24}$$

where in the second inequality we use Proposition C.2, with $a = [a_1, \ldots, a_t]^\top$ and $b = [b_1, \ldots, b_t]^\top$, setting $a_k = \lambda^{(t-k)/2}$ and $b_k = \alpha_k \lambda^{(t-k)/2} \|\mathbf{g}^k\|$. Next, consider the quantity $\|\mathbf{g}^k\|^2$. Using the fact that $\|\mathbf{g}^k\|^2 = \sum_{i \in [n]} \|g_i^k\|^2$, we get

$$
\|\mathbf{g}^k\|^2 = \sum_{i \in [n]} \|g_i^k\|^2 \overset{(a)}{\leq} 2 \sum_{i \in [n]} \|z_i^k\|^2 + 2 \sum_{i \in [n]} \|\nabla f_i(x_i^k)\|^2
$$

$$
\overset{(b)}{\leq} 2 \sum_{i \in [n]} \|z_i^k\|^2 + 2nA^2 + 2B^2 \sum_{i \in [n]} \|\nabla f(x_i^k)\|^2, \tag{25}
$$

where $(a)$ follows from $g_i^k = \nabla f_i(x_i^k) + z_i^k$ and using Proposition C.3 with $\theta = 1$, while $(b)$ follows from **(A5)**. Plugging (25) into (24), using the fact that $\sum_{k \in [t]} \lambda^{t-k} = \sum_{k=0}^{t-1} \lambda^k \leq \frac{1}{1-\lambda}$, the desired result follows. $\qquad \square$

We are now ready to prove Theorem 3.4, which we restate next, for convenience.

**Theorem 3.4.** *Let* **(A1)**-**(A5)** *hold. If for any $T \geq 1$, the step-size is chosen such that $\alpha_T \equiv \alpha = \min\left\{C, \frac{\sqrt{n}}{\sigma\sqrt{15LT}}\right\}$, where $C > 0$ is a problem related constant satisfying $C \leq \min\left\{\frac{1}{2L}, \frac{n}{9\sigma^2}, \frac{1-\lambda}{\lambda L B \sqrt{48}}, \frac{\sqrt{n}}{3\sigma\sqrt{5L}}, \frac{\sqrt[3]{n}(1-\lambda)^{2/3}}{\sigma^{2/3}\lambda^2 L^{2/3}\sqrt[3]{9}}\right\}$, then for any $\delta \in (0,1)$, with probability at least $1 - \delta$*

$$
\frac{1}{nT} \sum_{t \in [T]} \sum_{i \in [n]} \|\nabla f(x_i^t)\|^2 = \mathcal{O}\left(\frac{\sigma\sqrt{L}(\Delta_f + \log(2d/\delta))}{\sqrt{nT}} + \frac{\Delta_f + \log(1/\delta)}{CT} + \frac{\Delta_x L^2}{(1-\lambda^2)T} + \frac{n\lambda^2 L(A^2 + \sigma^2)}{\sigma^2(1-\lambda)^2 T}\right).
$$

*Proof.* Using Lemma 3.1, rearranging and summing up the first $T$ terms, we get

$$
\sum_{t \in [T]} \frac{\alpha_t}{2} \|\nabla f(\overline{x}^t)\|^2 \leq \Delta_f - \sum_{t \in [T]} \alpha_t \langle \nabla f(\overline{x}^t), \overline{z}^t \rangle + L \sum_{t \in [T]} \alpha_t^2 \|\overline{z}^t\|^2 + \frac{L^2}{2n} \sum_{t \in [T]} \alpha_t \sum_{i \in [n]} \|x_i^t - \overline{x}^t\|^2.
$$

Next, to offset the effect of the inner product term, we subtract $\frac{9\sigma^2}{4n} \sum_{t \in [T]} \alpha_t^2 \|\nabla f(\overline{x}^t)\|^2$ from both sides of the above inequality and note that, choosing $\alpha_t \leq \frac{n}{9\sigma^2}$, we have $\frac{\alpha_t}{2} - \frac{9\alpha_t^2 \sigma^2}{4n} \geq \frac{\alpha_t}{4}$. Therefore, we obtain

$$
\sum_{t \in [T]} \frac{\alpha_t}{4} \|\nabla f(\overline{x}^t)\|^2 \leq \Delta_f + L \sum_{t \in [T]} \alpha_t^2 \|\overline{z}^t\|^2 + \frac{L^2}{2n} \sum_{t \in [T]} \alpha_t \sum_{i \in [n]} \|x_i^t - \overline{x}^t\|^2
$$

$$
- \sum_{t \in [T]} \alpha_t \left( \langle \nabla f(\overline{x}^t), \overline{z}^t \rangle + \frac{9\alpha_t \sigma^2}{4n} \|\nabla f(\overline{x}^t)\|^2 \right). \tag{26}
$$

Using Proposition C.3 with $\theta = 1$ and Lipschitz continuity of gradients of $f$, it readily follows that

$$
\|\nabla f(x_i^t)\|^2 \leq 2\|\nabla f(\overline{x}^t)\|^2 + 2L^2 \|\overline{x}^t - x_i^t\|^2,
$$

implying

$$
\|\nabla f(\overline{x}^t)\|^2 \geq \frac{1}{2n} \sum_{i \in [n]} \|\nabla f(x_i^t)\|^2 - \frac{L^2}{n} \sum_{i \in [n]} \|\overline{x}^t - x_i^t\|^2. \tag{27}
$$

Plugging (27) into (26) and rearranging, we get

$$
\frac{1}{8n} \sum_{t \in [T]} \sum_{i \in [n]} \alpha_t \|\nabla f(x_i^t)\|^2 \leq \Delta_f - \sum_{t \in [T]} \alpha_t \left( \langle \nabla f(\overline{x}^t), \overline{z}^t \rangle + \frac{9\alpha_t \sigma^2}{4n} \|\nabla f(\overline{x}^t)\|^2 \right)
$$

$$
+ L \sum_{t \in [T]} \alpha_t^2 \|\overline{z}^t\|^2 + \frac{3L^2}{4n} \sum_{t \in [T]} \alpha_t \sum_{i \in [n]} \|x_i^t - \overline{x}^t\|^2.
$$

Using Lemma 3.3, we then have

$$
\frac{1}{8n} \sum_{t\in[T]} \sum_{i\in[n]} \alpha_t \|\nabla f(x_i^t)\|^2 \le \Delta_f - \sum_{t\in[T]} \alpha_t \Big( \langle \nabla f(\overline{x}^t), \overline{z}^t \rangle + \frac{9\alpha_t \sigma^2}{4n} \|\nabla f(\overline{x}^t)\|^2 \Big) + L \sum_{t\in[T]} \alpha_t^2 \|\overline{z}^t\|^2
$$
$$
+ \frac{3L^2}{4} \sum_{t\in[T]} \alpha_t \Big( 2\lambda^{2(t-1)} \Delta_x + \frac{4\lambda^2}{n(1-\lambda)} \sum_{k\in[t-1]} \alpha_k^2 \lambda^{t-1-k} \sum_{i\in[n]} \Big[ \|z_i^k\|^2 + A^2 + B^2 \|\nabla f(x_i^k)\|^2 \Big] \Big). \tag{28}
$$

Next, note that, for any sequence $\{a_t\}_{t\in[T]}$, the following identity holds

$$
\sum_{t\in[T]} \sum_{k\in[t-1]} \lambda^{t-1-k} a_k = \sum_{t\in[T]} a_t \sum_{k\in[T-t]} \lambda^{k-1},
$$

implying that $\sum_{t\in[T]} \sum_{k\in[t-1]} \lambda^{t-1-k} a_k \le \frac{1}{1-\lambda} \sum_{t\in[T]} a_t$, if $\{a_t\}_{t\in[T]}$ is non-negative. Since the step-size is non-increasing, applying the previous identity to $\sum_{t\in[T]} \alpha_t \sum_{k\in[t-1]} \alpha_k^2 \lambda^{t-1-k} [\|z_i^k\|^2 + A^2 + B^2 \|\nabla f(x_i^k)\|^2]$, we get

$$
\sum_{t\in[T]} \alpha_t \sum_{k\in[t-1]} \alpha_k^2 \lambda^{t-1-k} \Big[ \|z_i^k\|^2 + A^2 + B^2 \|\nabla f(x_i^k)\|^2 \Big]
$$
$$
\le \sum_{t\in[T]} \sum_{k\in[t-1]} \alpha_k^3 \lambda^{t-1-k} \Big[ \|z_i^k\|^2 + A^2 + B^2 \|\nabla f(x_i^k)\|^2 \Big]
$$
$$
\le \frac{1}{1-\lambda} \sum_{t\in[T]} \alpha_t^3 \Big[ \|z_i^t\|^2 + A^2 + B^2 \|\nabla f(x_i^t)\|^2 \Big]. \tag{29}
$$

Plugging (29) into (28), it follows that

$$
\frac{1}{8n} \sum_{t\in[T]} \sum_{i\in[n]} \alpha_t \|\nabla f(x_i^t)\|^2 \le \Delta_f - \sum_{t\in[T]} \alpha_t \Big( \langle \nabla f(\overline{x}^t), \overline{z}^t \rangle + \frac{9\alpha_t \sigma^2}{4n} \|\nabla f(\overline{x}^t)\|^2 \Big) + L \sum_{t\in[T]} \alpha_t^2 \|\overline{z}^t\|^2
$$
$$
+ \frac{3\Delta_x L^2}{2} \sum_{t\in[T]} \alpha_t \lambda^{2(t-1)} + \frac{3\lambda^2 L^2}{n(1-\lambda)^2} \sum_{t\in[T]} \alpha_t^3 \sum_{i\in[n]} \Big[ \|z_i^t\|^2 + A^2 + B^2 \|\nabla f(x_i^t)\|^2 \Big].
$$

Rearranging and choosing $\alpha_t \le \frac{1-\lambda}{\lambda L B \sqrt{48}}$, we have

$$
\frac{1}{16n} \sum_{t\in[T]} \sum_{i\in[n]} \alpha_t \|\nabla f(x_i^t)\|^2 \le \Delta_f - \sum_{t\in[T]} \alpha_t \Big( \langle \nabla f(\overline{x}^t), \overline{z}^t \rangle + \frac{9\alpha \sigma^2}{4n} \|\nabla f(\overline{x}^t)\|^2 \Big) + L \sum_{t\in[T]} \alpha_t^2 \|\overline{z}^t\|^2
$$
$$
+ \frac{3\Delta_x L^2}{2} \sum_{t\in[T]} \alpha_t \lambda^{2(t-1)} + \frac{3\lambda^2 A^2 L^2}{(1-\lambda)^2} \sum_{t\in[T]} \alpha_t^3 + \frac{3\lambda^2 L^2}{n(1-\lambda)^2} \sum_{t\in[T]} \sum_{i\in[n]} \alpha_t^3 \|z_i^t\|^2. \tag{30}
$$

Define the process $M_T := \frac{1}{16n} \sum_{t\in[T]} \sum_{i\in[n]} \alpha_t \|\nabla f(x_i^t)\|^2 - \Delta_f - \frac{3\Delta_x L^2}{2} \sum_{t\in[T]} \alpha_t \lambda^{2(t-1)} - \frac{3\lambda^2 A^2 L^2}{(1-\lambda)^2} \sum_{t\in[T]} \alpha_t^3$ and consider its' MGF. Using (30), we then get

$$
\mathbb{E}[\exp(M_T)] \le \mathbb{E}\Bigg[ \exp\Big( \sum_{t\in[T]} \underbrace{-\alpha_t \Big( \langle \nabla f(\overline{x}^t), \overline{z}^t \rangle + \frac{9\alpha_t \sigma^2 \|\nabla f(\overline{x}^t)\|^2}{4n} \Big)}_{b_{1,t}} \Big) \exp\Big( \sum_{t\in[T]} \underbrace{\alpha_t^2 L \|\overline{z}^t\|^2}_{b_{2,t}} \Big)
$$
$$
\times \exp\Big( \sum_{t\in[T]} \underbrace{\frac{3\alpha_t^3 \lambda^2 L^2}{n(1-\lambda)^2} \sum_{i\in[n]} \|z_i^t\|^2}_{b_{3,t}} \Big) \Bigg] \le \sqrt[3]{\mathbb{E}[\exp(B_{1,T})] \mathbb{E}[\exp(B_{2,T})] \mathbb{E}[\exp(B_{3,T})]}, \tag{31}
$$

where $B_{k,T} = 3 \sum_{t\in[T]} b_{1,t}$ and $B_{k,0} = 0$ for all $k \in [3]$, while the last step follows by applying Proposition C.5. We now analyze each quantity separately, starting with $\mathbb{E}[\exp(B_{1,T})]$. To that end, we have

$$
\mathbb{E}[\exp(B_{1,T})] = \mathbb{E}\Bigg[ \exp\Big( B_{1,T-1} - \frac{27\alpha_T^2 \sigma^2 \|\nabla f(\overline{x}^T)\|^2}{4n} \Big) \mathbb{E}\Big[ \exp\big( -3\alpha_T \langle \nabla f(\overline{x}^T), \overline{z}^T \rangle \big) \mid \mathcal{F}_T \Big] \Bigg].
$$

Noting that $\nabla f(\overline{x}^T)$ is $\mathcal{F}_T$-measurable and applying Lemma 3.2, we get

$$\mathbb{E}[\exp(B_{1,T})] \le \mathbb{E}\left[\exp\left(B_{1,T-1} - \frac{27\alpha_T^2\sigma^2\|\nabla f(\overline{x}^T)\|^2}{4n} + \frac{27\alpha_T^2\sigma^2\|\nabla f(\overline{x}^T)\|^2}{4n}\right)\right] = \mathbb{E}[\exp(B_{1,T-1})].$$

Unrolling the recursion, it follows that $\mathbb{E}[\exp(B_{1,T})] \le 1$. Next, consider $\mathbb{E}[\exp(B_{2,T})]$. To that end, we have

$$\mathbb{E}[\exp(B_{2,T})] = \mathbb{E}\left[\exp(B_{2,T-1})\mathbb{E}\left[\exp\left(3\alpha_T^2 L\|\overline{z}^T\|^2\right)\mid\mathcal{F}_T\right]\right]$$

$$= \mathbb{E}\left[\exp(B_{2,T-1})\mathbb{E}\left[\exp\left(\frac{45\alpha_T^2\sigma^2 L}{n} \times \frac{n\|\overline{z}^T\|^2}{15\sigma^2}\right)\mid\mathcal{F}_T\right]\right]$$

$$\le \mathbb{E}\left[\exp(B_{2,T-1})\left(\mathbb{E}\left[\exp\left(\frac{n\|\overline{z}^T\|^2}{15\sigma^2}\right)\mid\mathcal{F}_T\right]\right)^{45\alpha_T^2\sigma^2 L/n}\right],$$

where the last inequality follows from $\alpha_t \le \frac{\sqrt{n}}{3\sigma\sqrt{5L}}$ and Proposition C.1. From the third property in Lemma 3.2 we get

$$\mathbb{E}[\exp(B_{2,T})] \le \exp\left(\frac{45\alpha_T^2\sigma^2 L}{n}\left(1 + \log(2d)\right)\right)\mathbb{E}\left[\exp(B_{2,T-1})\right].$$

Unrolling the recursion, it follows that $\mathbb{E}[\exp(B_{2,T})] \le \exp\left(\frac{45\sigma^2 L}{n}\left(1 + \log(2d)\right)\sum_{t\in[T]}\alpha_t^2\right)$. Finally, to bound $\mathbb{E}[\exp(B_{3,T})]$, we can proceed in the same way, using the conditional independence of noise across agents from **(A4)** to note that, if $\alpha_t \le \frac{\sqrt[3]{n}(1-\lambda)^{2/3}}{\sigma^{2/3}\lambda^{2/3}L^{2/3}\sqrt[3]{9}}$, then $\mathbb{E}[\exp(B_{3,T})] \le \exp\left(\frac{9\sigma^2\lambda^2 L^2}{(1-\lambda)^2}\sum_{t\in[T]}\alpha_t^3\right)$. Combining everything, we get

$$\mathbb{E}[\exp(M_T)] \le \exp\left(\frac{15\sigma^2 L}{n}\left(1 + \log(2d)\right)\sum_{t\in[T]}\alpha_t^2 + \frac{3\sigma^2\lambda^2 L^2}{(1-\lambda)^2}\sum_{t\in[T]}\alpha_t^3\right). \tag{32}$$

Using Markov's inequality and (32), we get, for any $\epsilon > 0$

$$\mathbb{P}(M_T > \epsilon) \le \exp(-\epsilon)\mathbb{E}[\exp(M_T)] \le \exp\left(-\epsilon + \frac{15\sigma^2 L}{n}\left(1 + \log(2d)\right)\sum_{t\in[T]}\alpha_t^2 + \frac{3\sigma^2\lambda^2 L^2}{(1-\lambda)^2}\sum_{t\in[T]}\alpha_t^3\right),$$

or equivalently, for any $\delta \in (0, 1)$, with probability at least $1 - \delta$

$$M_T \le \log(1/\delta) + \frac{15\sigma^2 L}{n}\left(1 + \log(2d)\right)\sum_{t\in[T]}\alpha_t^2 + \frac{3\sigma^2\lambda^2 L^2}{(1-\lambda)^2}\sum_{t\in[T]}\alpha_t^3.$$

Using the definition of $M_T$ and that the sequence of step-sizes is non-increasing, we get with probability at least $1 - \delta$

$$\frac{\alpha_T}{16n}\sum_{t\in[T]}\sum_{i\in[n]}\|\nabla f(x_i^t)\|^2 \le \Delta_f + \log(1/\delta) + \frac{3\Delta_x L^2}{2}\sum_{t\in[T]}\alpha_t\lambda^{2(t-1)}$$

$$+ \frac{15\sigma^2 L}{n}\left(1 + \log(2d)\right)\sum_{t\in[T]}\alpha_t^2 + \frac{3\lambda^2 L^2(\sigma^2 + A^2)}{(1-\lambda)^2}\sum_{t\in[T]}\alpha_t^3.$$

Dividing both sides by $\frac{\alpha_T T}{16}$, it follows that, with probability at least $1 - \delta$

$$\frac{1}{nT}\sum_{t\in[T]}\sum_{i\in[n]}\|\nabla f(x_i^t)\|^2 \le \frac{16\left(\Delta_f + \log(1/\delta)\right)}{\alpha_T T} + \frac{24\Delta_x L^2}{\alpha_T T}\sum_{t\in[T]}\alpha_t\lambda^{2(t-1)}$$

$$+ \frac{240\sigma^2 L}{n\alpha_T T}\left(1 + \log(2d)\right)\sum_{t\in[T]}\alpha_t^2 + \frac{48\lambda^2 L^2(\sigma^2 + A^2)}{\alpha_T T(1-\lambda)^2}\sum_{t\in[T]}\alpha_t^3.$$

Next, consider two regimes with respect to the time horizon $T$.

1. *Known time horizon.* In this case, we choose a fixed step-size $\alpha_t \equiv \alpha$, for all $t \in [T]$. Noticing that $\sum_{t \in [T]} \alpha_t \lambda^{2(t-1)} \leq \frac{\alpha}{1-\lambda^2}$, we then have, with probability at least $1 - \delta$

$$\frac{1}{nT} \sum_{t \in [T]} \sum_{i \in [n]} \|\nabla f(x_i^t)\|^2 \leq \frac{16(\Delta_f + \log(1/\delta))}{\alpha T} + \frac{24\Delta_x L^2}{(1-\lambda^2)T} + \frac{240\alpha\sigma^2 L(1 + \log(2d))}{n} + \frac{48\alpha^2\lambda^2 L^2(\sigma^2 + A^2)}{(1-\lambda)^2}.$$

If the step-size satisfies $\alpha \leq \frac{\sqrt{n}}{\sigma\sqrt{15LT}}$, then with probability at least $1 - \delta$

$$\frac{1}{nT} \sum_{t \in [T]} \sum_{i \in [n]} \|\nabla f(x_i^t)\|^2 \leq \frac{16(\Delta_f + \log(1/\delta))}{\alpha T} + \frac{24\Delta_x L^2}{(1-\lambda^2)T} + \frac{16\sigma\sqrt{15L}(1 + \log(2d))}{\sqrt{nT}} + \frac{4n\lambda^2 L(\sigma^2 + A^2)}{\sigma^2(1-\lambda)^2 T}.$$

Setting $C = \min\left\{\frac{1}{2L}, \frac{n}{9\sigma^2}, \frac{1-\lambda}{\lambda LB\sqrt{48}}, \frac{\sqrt{n}}{3\sigma\sqrt{5L}}, \frac{\sqrt[3]{n}(1-\lambda)^{2/3}}{\sigma^{2/3}\lambda^2 L^{2/3}\sqrt[3]{9}}\right\}$ and $\alpha = \min\left\{C, \frac{\sqrt{n}}{\sigma\sqrt{15LT}}\right\}$ guarantees that all the step-size conditions are satisfied and it readily follows that $\frac{1}{\alpha} = \max\left\{\frac{1}{C}, \frac{\sigma\sqrt{15LT}}{\sqrt{n}}\right\} \leq \frac{1}{C} + \frac{\sigma\sqrt{15LT}}{\sqrt{n}}$, implying that $\frac{1}{\alpha T} \leq \frac{\sigma\sqrt{15L}}{\sqrt{nT}} + \frac{1}{CT}$. Therefore, for any $\delta \in (0, 1)$, with probability at least $1 - \delta$, we finally have

$$\frac{1}{nT} \sum_{t \in [T]} \sum_{i \in [n]} \|\nabla f(x_i^t)\|^2 \leq \frac{16\sigma\sqrt{15L}(\Delta_f + \log(2d/\delta) + 1)}{\sqrt{nT}} + \frac{16(\Delta_f + \log(1/\delta))}{CT} + \frac{24\Delta_x L^2}{(1-\lambda^2)T} + \frac{4n\lambda^2 L(\sigma^2 + A^2)}{\sigma^2(1-\lambda)^2 T}.$$

2. *Unknown time horizon.* In this case, we choose a time-varying step-size $\alpha_t = \frac{C'}{\sqrt{t+1}}$, for all $t \geq 1$ and $C' = \sqrt{2}C$, where $C = \min\left\{\frac{1}{2L}, \frac{n}{9\sigma^2}, \frac{1-\lambda}{\lambda LB\sqrt{48}}, \frac{\sqrt{n}}{3\sigma\sqrt{5L}}, \frac{\sqrt[3]{n}(1-\lambda)^{2/3}}{\sigma^{2/3}\lambda^2 L^{2/3}\sqrt[3]{9}}\right\}$, again guaranteeing that all the step-size conditions are satisfied. Noting that $\alpha_t \leq C$, we get $\sum_{t \in [T]} \alpha_t \lambda^{2(t-1)} \leq \frac{C}{(1-\lambda^2)}$, hence, with probability at least $1 - \delta$

$$\frac{1}{nT} \sum_{t \in [T]} \sum_{i \in [n]} \|\nabla f(x_i^t)\|^2 \leq \frac{16\sqrt{2}(\Delta_f + \log(1/\delta))}{C\sqrt{T+1}} + \frac{24\sqrt{2}\Delta_x L^2}{C\sqrt{T+1}(1-\lambda^2)}$$
$$+ \frac{240\sqrt{2}\sigma^2 LC \log(T+1)(1 + \log(2d))}{n\sqrt{T+1}} + \frac{384\lambda^2 L^2(\sigma^2 + A^2)C^2}{(1-\lambda)^2\sqrt{T+1}}.$$

Note that in the unknown time horizon we lose linear speed-up, as even when $C = \frac{\sqrt{n}}{3\sigma\sqrt{5L}}$, we get

$$\frac{1}{nT} \sum_{t \in [T]} \sum_{i \in [n]} \|\nabla f(x_i^t)\|^2 = \mathcal{O}\left(\frac{\sigma\sqrt{L}(\Delta_f + \log(2dT/\delta))}{\sqrt{n(T+1)}} + \frac{\sigma\Delta_x L^{3/2}}{(1-\lambda^2)\sqrt{n(T+1)}} + \frac{n\lambda^2 L(\sigma^2 + A^2)}{\sigma^2(1-\lambda)^2\sqrt{T+1}}\right),$$

since the term which does not attain linear speed-up is no longer of higher order. We note that linear speed-up in the MSE sense is also achieved under a known time horizon and fixed step-size, e.g., (Koloskova et al., 2020; Xin et al., 2021).

$\square$

# G. Proofs for Strongly Convex Costs

In this section we prove Lemma 3.5 and Theorem 3.7. To do so, we follow a similar strategy to the one in, e.g., (Jakovetić et al., 2018), where it is shown that the sequence of iterates generated by **DSGD** is bounded in the MSE sense. However, to establish HP guarantees we instead work with the MGF of the iterates and have the following result.

**Lemma G.1.** *Let assumptions (A1)-(A4) and (A6) hold and let $a, t_0, K > 0$ and $\nu \in (0, 1]$ be positive constants. If for all $t \geq 1$ the step-size satisfies $\alpha_t \leq \min\left\{\frac{1}{\bar{\sigma}\sqrt{2(t+t_0+2)K}}, \frac{1}{\mu}\right\}$, with $\nu \leq \min\left\{1, \frac{\mu}{24a\bar{\sigma}^2 K}\right\}$ and $K_{t+1} = (t + t_0 + 2)K$, then*

$$\mathbb{E}[\exp(\nu K_{t+1}\|\mathbf{x}^{t+1} - \mathbf{x}^\star\|^2)] \leq \exp\left(\nu K_{t+1}\left(\frac{4an\sigma^2\alpha_{t+1}}{a\mu - 1} + \frac{9\|\nabla\mathbf{f}^\star\|^2}{\mu^2} + \frac{(1 + t_0)^{a\mu}\|\mathbf{x}^1 - \mathbf{x}^\star\|^2}{(t + 1 + t_0)^{a\mu}}\right)\right).$$

*Proof.* Consider the update rule (18). We then have

$$\mathbf{x}^{t+1} - \mathbf{x}^\star = \mathbf{W}(\mathbf{x}^t - \mathbf{x}^\star - \alpha_t \mathbf{g}_t) = \mathbf{W}(\mathbf{x}^t - \mathbf{x}^\star - \alpha_t \nabla \mathbf{f}^t - \alpha_t \mathbf{z}^t), \tag{33}$$

where in the first equality we used $\mathbf{W}\mathbf{x}^\star = \mathbf{x}^\star$. Using Taylor's expansion, for each $i \in [n]$ and $x \in \mathbb{R}^d$, we have

$$\nabla f_i(x) = \nabla f_i(x^\star) + \int_0^1 \nabla^2 f_i(x^\star + \tau(x - x^\star))d\tau(x - x^\star) = \nabla f_i(x^\star) + H_i(x)(x - x^\star). \tag{34}$$

Denote by $\mathbf{H}^t := diag(H_1(x_1^t), \ldots, H_n(x_n^t)) \in \mathbb{R}^{nd \times nd}$ the block diagonal matrix and recall that $\nabla \mathbf{f}^\star = col(\nabla f_1(x^\star), \ldots, \nabla f_n(x^\star)) \in \mathbb{R}^{nd}$. Using (34), we can readily see that $\nabla \mathbf{f}^t = \nabla \mathbf{f}^\star + \mathbf{H}^t(\mathbf{x}^t - \mathbf{x}^\star)$, therefore, plugging in (33), we get

$$\mathbf{x}^{t+1} - \mathbf{x}^\star = \mathbf{W}(\mathbf{I} - \alpha_t \mathbf{H}^t)(\mathbf{x}^t - \mathbf{x}^\star) - \alpha_t \mathbf{W}\nabla \mathbf{f}^\star - \alpha_t \mathbf{W}\mathbf{z}^t = C_t + \alpha_t \mathbf{W}\mathbf{z}^t,$$

where $C_t := \mathbf{W}(\mathbf{I} - \alpha_t \mathbf{H}^t)(\mathbf{x}^t - \mathbf{x}^\star) - \alpha_t \mathbf{W}\nabla \mathbf{f}^\star$. Therefore, we have

$$\|\mathbf{x}^{t+1} - \mathbf{x}^\star\|^2 = \|C_t\|^2 - 2\alpha_t \langle \mathbf{W}C_t, \mathbf{z}^t \rangle + \alpha_t^2 \|\mathbf{W}\mathbf{z}^t\|^2 \leq \|C_t\|^2 - 2\alpha_t \langle \mathbf{W}C_t, \mathbf{z}^t \rangle + \alpha_t^2 \|\mathbf{z}^t\|^2$$

$$\overset{(i)}{\leq} (1+\theta)\|\mathbf{W}(\mathbf{I} - \alpha_t \mathbf{H}^t)(\mathbf{x}^t - \mathbf{x}^\star)\|^2 + (1+\theta^{-1})\alpha_t^2\|\mathbf{W}\nabla \mathbf{f}^\star\|^2 - 2\alpha_t \langle \mathbf{W}C_t, \mathbf{z}^t \rangle + \alpha_t^2\|\mathbf{z}^t\|^2$$

$$\overset{(ii)}{\leq} (1+\theta)(1 - \alpha_t\mu)^2\|\mathbf{x}^t - \mathbf{x}^\star\|^2 + (1+\theta^{-1})\alpha_t^2\|\nabla \mathbf{f}^\star\|^2 - 2\alpha_t \langle \mathbf{W}C_t, \mathbf{z}^t \rangle + \alpha_t^2\|\mathbf{z}^t\|^2,$$

where in $(i)$ we used Proposition C.3, for some $\theta > 0$ (to be specified later), while $(ii)$ follows from Proposition E.1 and the fact that $\|\mathbf{I} - \alpha_t \mathbf{H}^t\| \leq (1 - \alpha_t\mu)$ (as a consequence of Proposition C.8). Define $D_t := (1+\theta)(1 - \alpha_t\mu)^2\|\mathbf{x}^t - \mathbf{x}^\star\|^2 + (1+\theta^{-1})\alpha_t^2\|\nabla \mathbf{f}^\star\|^2$ and consider the MGF of $\nu K_{t+1}\|\mathbf{x}^{t+1} - \mathbf{x}^\star\|^2$ conditioned on $\mathcal{F}_t$, where we recall that $K_{t+1} = (t + t_0 + 2)K$ for some $K > 0$ and $\nu \in (0, 1]$. We then have

$$\mathbb{E}_t[\exp(\nu K_{t+1}\|\mathbf{x}^{t+1} - \mathbf{x}^\star\|^2)] \overset{(a)}{\leq} \exp(\nu K_{t+1}D_t)\mathbb{E}_t\left[\exp\left(\nu K_{t+1}\left(-2\alpha_t\langle \mathbf{W}C_t, \mathbf{z}^t\rangle + \alpha_t^2\|\mathbf{z}^t\|^2\right)\right)\right]$$

$$\overset{(b)}{\leq} \exp(\nu K_{t+1}D_t)\sqrt{\mathbb{E}_t[\exp(-4\alpha_t\nu K_{t+1}\langle \mathbf{W}C_t, \mathbf{z}^t\rangle)\mathbb{E}_t[\exp(2\alpha_t^2\nu K_{t+1}\|\mathbf{z}^t\|^2)]}$$

$$\overset{(c)}{\leq} \exp(\nu K_{t+1}D_t)\sqrt{\exp(12\alpha_t^2\overline{\sigma}^2\nu^2 K_{t+1}^2\|\mathbf{W}C_t\|^2 + 2\alpha_t^2 n\sigma^2\nu K_{t+1})}$$

$$\overset{(d)}{\leq} \exp(\nu K_{t+1}D_t + 6\alpha_t^2\overline{\sigma}^2\nu^2 K_{t+1}^2\|C_t\|^2 + \alpha_t^2 n\sigma^2\nu K_{t+1})$$

$$\overset{(e)}{\leq} \exp(\nu K_{t+1}(1 + 6\alpha_t^2\overline{\sigma}^2\nu K_{t+1})D_t + \alpha_t^2 n\sigma^2\nu K_{t+1}), \tag{35}$$

where $(a)$ follows from the fact that $D_t$ is $\mathcal{F}_t$-measurable, in $(b)$ we used Proposition C.4, $(c)$ follows from Lemma 3.2 and $\alpha_t \leq \frac{1}{\overline{\sigma}\sqrt{2(t+t_0+2)K}}$, in $(d)$ we used Proposition E.1, while $(e)$ follows from the definition of $D_t$ and the fact that $\|C_t\|^2 \leq D_t$. We now analyze $(1 + 6\alpha_t^2\overline{\sigma}^2\nu K_{t+1})D_t$. To that end, if we choose $\theta = \frac{\alpha_t\mu}{2}$, it follows that

$$(1 + 6\alpha_t^2\overline{\sigma}^2\nu K_{t+1})D_t = (1 + 6\alpha_t^2\overline{\sigma}^2\nu K_{t+1})\left[(1 + {}^{\alpha_t\mu}/_2)(1 - \alpha_t\mu)^2\|\mathbf{x}^t - \mathbf{x}^\star\|^2 + (1 + {}^2/_{\alpha_t\mu})\alpha_t^2\|\nabla \mathbf{f}^\star\|^2\right]$$

$$\overset{(i)}{\leq} (1 + 6\alpha_t^2\overline{\sigma}^2\nu K_{t+1})\left[(1 - {}^{\alpha_t\mu}/_2)(1 - \alpha_t\mu)\|\mathbf{x}^t - \mathbf{x}^\star\|^2 + (\alpha_t + {}^2/_\mu)\alpha_t\|\nabla \mathbf{f}^\star\|^2\right]$$

$$\overset{(ii)}{\leq} (1 - \alpha_t\mu)\|\mathbf{x}^t - \mathbf{x}^\star\|^2 + 9\alpha_t\|\nabla \mathbf{f}^\star\|^2/2\mu, \tag{36}$$

where in $(i)$ we used the fact that $(1 + \frac{a}{2})(1 - a) \leq (1 - \frac{a}{2})$ for any $a > 0$, while $(ii)$ follows by setting $\nu \leq \frac{\mu}{24a\overline{\sigma}^2 K}$ and from the step-size choice $\alpha_t \leq \frac{1}{\mu}$. Plugging (36) in (35), using the shorthand $E_t = \alpha_t^2 n\sigma^2 + 9\alpha_t\|\nabla \mathbf{f}^\star\|^2/2\mu$ and taking

the full expectation, we get

$$\mathbb{E}[\exp(\nu K_{t+1}\|\mathbf{x}^{t+1} - \mathbf{x}^\star\|^2)] \leq \exp(\nu K_{t+1}E_t)\mathbb{E}[\exp((1 - \alpha_t\mu)\nu K_{t+1}\|\mathbf{x}^t - \mathbf{x}^\star\|^2)]$$

$$\leq \exp(\nu K_{t+1}E_t)\Big(\mathbb{E}[\exp(\nu K_t\|\mathbf{x}^t - \mathbf{x}^\star\|^2)]\Big)^{(1-\alpha_t\mu)\frac{t+t_0+2}{t+t_0+1}}$$

$$\leq \exp\big(\nu K_{t+1}\big(E_t + (1 - \alpha_t\mu)E_{t-1}\big)\big)\Big(\mathbb{E}[\exp((1 - \alpha_{t-1}\mu)\nu K_t\|\mathbf{x}_{t-1} - \mathbf{x}^\star\|^2)]\Big)^{(1-\alpha_t\mu)\frac{t+t_0+2}{t+t_0+1}}$$

$$\leq \ldots \leq \exp\Big(\nu K_{t+1}\sum_{k=1}^{t} E_k \prod_{s=k+1}^{t}(1 - \alpha_k\mu) + \nu K_1\|\mathbf{x}^1 - \mathbf{x}^\star\|^2 \prod_{k=1}^{t}(1 - \alpha_k\mu)\frac{t + t_0 + 2}{t_0 + 2}\Big)$$

$$= \exp\Big(\nu K_{t+1}\Big(\sum_{k=1}^{t} E_k \prod_{s=k+1}^{t}(1 - \alpha_s\mu) + \|\mathbf{x}^1 - \mathbf{x}^\star\|^2 \prod_{k=1}^{t}(1 - \alpha_k\mu)\Big)\Big),$$

where the second inequality follows from Proposition C.1 and the fact that $0 < (1 - \alpha_t\mu)\frac{t+t_0+2}{t+t_0+1} \leq 1$, for any $t \geq 1$, whenever $0 < \alpha_t \leq \frac{1}{\mu}$. Next, we use Proposition C.6, to get

$$\sum_{k=1}^{t} E_k \prod_{s=k+1}^{t}(1 - \alpha_s\mu) \leq \sum_{k=1}^{t}\Big(\alpha_k^2 n\sigma^2 + 9\alpha_k\|\nabla\mathbf{f}^\star\|^2/2\mu\Big)\frac{(k + 1 + t_0)^{a\mu}}{(t + 1 + t_0)^{a\mu}}$$

$$\overset{(i)}{\leq} \frac{1}{(t + t_0 + 1)^{a\mu}}\sum_{k=1}^{t}\Big(4a^2 n\sigma^2(k + 1 + t_0)^{a\mu-2} + \frac{9a\|\nabla\mathbf{f}^\star\|^2}{\mu}(k + t_0 + 1)^{a\mu-1}\Big)$$

$$\overset{(ii)}{\leq} \frac{4a^2 n\sigma^2}{(a\mu - 1)(t + t_0 + 1)} + \frac{9\|\nabla\mathbf{f}^\star\|^2}{\mu^2},$$

where $(i)$ follows from the choice $\alpha_k \leq \frac{1}{\mu}$, while in $(ii)$ we use the lower Darboux sum. Combining everything, we get

$$\mathbb{E}[\exp(\nu K_{t+1}\|\mathbf{x}^{t+1} - \mathbf{x}^\star\|^2)] \leq \exp\Big(\nu K_{t+1}\Big(\frac{4an\sigma^2\alpha_{t+1}}{a\mu - 1} + \frac{9\|\nabla\mathbf{f}^\star\|^2}{\mu^2} + \frac{(1 + t_0)^{a\mu}\|\mathbf{x}^1 - \mathbf{x}^\star\|^2}{(t + t_0 + 1)^{a\mu}}\Big)\Big).$$

$\square$

Lemma G.1 is an important building block for bounding the consensus gap. We next restate and prove Lemma 3.5.

**Lemma 3.5.** *Let* **(A1)-(A4)** *and* **(A6)** *hold, let* $a, t_0, K > 0$ *and the step-size be given by* $\alpha_t = \frac{a}{t+t_0}$, *and let* $x_i^1 = x_j^1$, *for all* $i, j \in [n]$. *If* $a = \frac{6}{\mu}$ *and* $t_0 \geq \max\Big\{6, \frac{288\overline{\sigma}^2 K}{\mu^2}, \frac{3456\overline{\sigma}^2\lambda^2 K}{\mu^2(1-\lambda)}, \frac{12\lambda L\sqrt{10}}{\mu(1-\lambda)}\Big\}$, *then for* $K_{t+1} = (t + t_0 + 2)K$ *and any* $\nu \leq \min\Big\{1, \frac{\mu^2}{144\sigma^2 K}\Big\}$, *we have*

$$\mathbb{E}\Big[\exp\Big(\nu K_{t+1}\sum_{i\in[n]}\|x_i^{t+1} - \overline{x}^{t+1}\|^2\Big)\Big] \leq \exp\Big(\nu K_{t+1}\Big(\sum_{k\in[t]}\lambda^{t-k}S_k + \sum_{k\in[t]}\lambda^{t-k}D_k\Big)\Big),$$

*where* $S_k := \alpha_k^2\lambda^2\Big(n\sigma^2 + \frac{5\|\nabla\mathbf{f}^\star\|^2}{1-\lambda}\Big)$ *and* $D_k := \frac{5\alpha_k^2\lambda^2 L^2}{1-\lambda}\Big(\frac{4an\sigma^2\alpha_k}{5} + \frac{9\|\nabla\mathbf{f}^\star\|^2}{\mu^2} + \frac{(1+t_0)^6\|\mathbf{x}^1 - \mathbf{x}^\star\|^2}{(k+t_0)^6}\Big)$.

*Proof.* Note that $\sum_{i\in[n]}\|x_i^{t+1} - \overline{x}^{t+1}\|^2 = \|\mathbf{x}^{t+1} - \overline{\mathbf{x}}^{t+1}\|^2$ and recall $\widetilde{\mathbf{W}} = \mathbf{W} - \mathbf{J}$ and the update rule (18). Then

$$\mathbf{x}^{t+1} - \overline{\mathbf{x}}^{t+1} = \widetilde{\mathbf{W}}(\mathbf{x}^t - \overline{\mathbf{x}}^t - \alpha_t\nabla\mathbf{f}^t - \alpha_t\mathbf{z}^t).$$

Denote the consensus difference by $\widetilde{\mathbf{x}}^{t+1} := \mathbf{x}^{t+1} - \overline{\mathbf{x}}^{t+1}$ and let $C_t := \widetilde{\mathbf{W}}(\widetilde{\mathbf{x}}_t - \alpha_t\nabla\mathbf{f}^t)$. Noting that $\|\widetilde{\mathbf{x}}^{t+1}\|^2 = \|C_t\|^2 - 2\alpha_t\langle\widetilde{\mathbf{W}}C_t, \mathbf{z}^t\rangle + \alpha_t\|\widetilde{\mathbf{W}}\mathbf{z}^t\|^2$, we then consider the MGF of $\nu K_{t+1}\|\widetilde{\mathbf{x}}^{t+1}\|^2$ conditioned on $\mathcal{F}_t$, where we recall that $K_{t+1} = (t + t_0 + 2)K$, for some $K > 0$ and $\nu \in (0, 1]$. If $\alpha_t \leq \frac{1}{\overline{\sigma}\lambda\sqrt{2(t+t_0+2)K}}$, we have

$$\mathbb{E}_t[\exp(\nu K_{t+1}\|\widetilde{\mathbf{x}}^{t+1}\|^2)] \leq \exp(\nu K_{t+1}\|C_t\|^2)\sqrt{\mathbb{E}_t\big[\exp(-4\alpha_t\nu K_{t+1}\langle\widetilde{\mathbf{W}}C_t, \mathbf{z}^t\rangle)\big]\mathbb{E}_t\big[\exp(2\alpha_t^2\lambda^2\nu K_{t+1}\|\mathbf{z}^t\|^2)\big]}$$

$$\leq \exp\Big(\nu K_{t+1}\Big((1 + 6\alpha_t^2\overline{\sigma}^2\lambda^2\nu K_{t+1})\|C_t\|^2 + \alpha_t^2\lambda^2 n\sigma^2\Big)\Big), \tag{37}$$

where the first inequality follows from the fact that $C_t$ is $\mathcal{F}_t$-measurable and using Proposition C.4, while the second follows from Lemma 3.2. Next, using Proposition E.1 and defining $\widetilde{\lambda} := 1 - \lambda \in (0, 1]$, we get

$$
\begin{aligned}
\|C_t\|^2 \leq \lambda^2 \|\widetilde{\mathbf{x}}_t - \alpha_t \nabla \mathbf{f}^t\|^2 &\overset{(i)}{\leq} \lambda^2 (1+\theta)\|\widetilde{\mathbf{x}}_t\|^2 + \alpha_t^2 \lambda^2 (1+\theta^{-1})\|\nabla \mathbf{f}^t\|^2 \\
&= (1-\widetilde{\lambda})(1+\theta)\lambda\|\widetilde{\mathbf{x}}_t\|^2 + \alpha_t^2 \lambda^2 (1+\theta^{-1})\|\nabla \mathbf{f}^t\|^2 \\
&\overset{(ii)}{\leq} (1-\widetilde{\lambda}/2)\lambda\|\widetilde{\mathbf{x}}_t\|^2 + \alpha_t^2 \lambda^2 (1+2/\widetilde{\lambda})\|\nabla \mathbf{f}^t\|^2 \\
&\overset{(iii)}{\leq} (1-\widetilde{\lambda}/2)\lambda\|\widetilde{\mathbf{x}}_t\|^2 + \frac{6\alpha_t^2 \lambda^2}{1-\lambda}\|\nabla \mathbf{f}^\star\|^2 + \frac{6\alpha_t^2 \lambda^2 L^2}{1-\lambda}\|\mathbf{x}^t - \mathbf{x}^\star\|^2,
\end{aligned}
\tag{38}
$$

where $(i)$ follows from Proposition C.3, in $(ii)$ we set $\theta = \frac{\widetilde{\lambda}}{2}$, while $(iii)$ follows from the fact that $\|\nabla \mathbf{f}^t\|^2 \leq 2L^2\|\mathbf{x}^t - \mathbf{x}^\star\| + 2\|\nabla \mathbf{f}^\star\|^2$ (recall Proposition C.7). Choosing $\alpha_t \leq \frac{\sqrt{1-\lambda}}{2\overline{\sigma}\lambda\sqrt{6(t+t_0+2)K}}$ and plugging (38) in (37), we get

$$
\begin{aligned}
\mathbb{E}_t[\exp(\nu K_{t+1}\|\widetilde{\mathbf{x}}^{t+1}\|^2)] \leq \exp\bigg( &\nu K_{t+1}\Big(1+\frac{\widetilde{\lambda}}{4}\Big)\Big[\lambda\Big(1-\frac{\widetilde{\lambda}}{2}\Big)\|\widetilde{\mathbf{x}}_t\|^2 \\
&+ \frac{4\alpha_t^2 \lambda^2}{1-\lambda}\|\nabla \mathbf{f}^\star\|^2 + \frac{4\alpha_t^2 \lambda^2 L^2}{1-\lambda}\|\mathbf{x}^t - \mathbf{x}^\star\|^2\Big] + \alpha_t^2 n\sigma^2 \lambda^2 \nu K_{t+1}\bigg) \\
\leq \exp\bigg( &\nu K_{t+1}\Big(\lambda\Big(1-\frac{\widetilde{\lambda}}{4}\Big)\|\widetilde{\mathbf{x}}_t\|^2 + \frac{5\alpha_t^2 \lambda^2}{1-\lambda}\|\nabla \mathbf{f}^\star\|^2 + \frac{5\alpha_t^2 \lambda^2 L^2}{1-\lambda}\|\mathbf{x}^t - \mathbf{x}^\star\|^2 + \alpha_t^2 n\sigma^2 \lambda^2\Big)\bigg).
\end{aligned}
$$

Taking the full expectation and introducing the shorthand $S_t := \alpha_t^2 \lambda^2\Big(n\sigma^2 + \frac{5\|\nabla \mathbf{f}^\star\|^2}{1-\lambda}\Big)$, we get

$$
\begin{aligned}
\mathbb{E}[\exp(\nu K_{t+1}\|\widetilde{\mathbf{x}}^{t+1}\|^2)] &\leq \exp(S_t \nu K_{t+1})\mathbb{E}\bigg[\exp\bigg(\lambda \nu K_{t+1}\Big(1-\frac{\widetilde{\lambda}}{4}\Big)\|\widetilde{\mathbf{x}}_t\|^2 + \frac{5\alpha_t^2 \lambda^2 L^2 \nu K_{t+1}}{1-\lambda}\|\mathbf{x}^t - \mathbf{x}^\star\|^2\bigg)\bigg] \\
&\leq \exp(S_t \nu K_{t+1})\Big(\mathbb{E}\big[\exp(\lambda \nu K_{t+1}\|\widetilde{\mathbf{x}}_t\|^2)\big]\Big)^{1-\widetilde{\lambda}/4}\bigg(\mathbb{E}\bigg[\exp\Big(\frac{20\alpha_t^2 \lambda^2 L^2 \nu K_{t+1}}{(1-\lambda)^2}\|\mathbf{x}^t - \mathbf{x}^\star\|^2\Big)\bigg]\bigg)^{\widetilde{\lambda}/4} \\
&\leq \exp(S_t \nu K_{t+1})\Big(\mathbb{E}\big[\exp(\nu K_t\|\widetilde{\mathbf{x}}_t\|^2)\big]\Big)^{\lambda(1-\widetilde{\lambda}/4)\frac{t+t_0+2}{t+t_0+1}}\bigg(\mathbb{E}\bigg[\exp\Big(\frac{20\alpha_t^2 \lambda^2 L^2 \nu K_{t+1}}{(1-\lambda)^2}\|\mathbf{x}^t - \mathbf{x}^\star\|^2\Big)\bigg]\bigg)^{\widetilde{\lambda}/4},
\end{aligned}
\tag{39}
$$

where the second inequality follows by applying Proposition C.4 with $p = (1-\widetilde{\lambda}/4)^{-1}$ and $q = \frac{4}{\widetilde{\lambda}}$, while the third follows from the fact that $\lambda\frac{t+t_0+2}{t+t_0+1} \leq 1$ for $t_0 \geq \frac{1}{1-\lambda}$ and applying Proposition C.1. If $\alpha_t \leq \frac{1-\lambda}{2\lambda L\sqrt{10}}$, we get

$$
\begin{aligned}
\bigg(\mathbb{E}\bigg[\exp\Big(\frac{20\alpha_t^2 \lambda^2 L^2 \nu K_{t+1}}{(1-\lambda)^2}\|\mathbf{x}^t - \mathbf{x}^\star\|^2\Big)\bigg]\bigg)^{\widetilde{\lambda}/4} &\leq \Big(\mathbb{E}\big[\exp(\nu K_t\|\mathbf{x}^t - \mathbf{x}^\star\|^2)\big]\Big)^{\frac{5\alpha_t^2(t+t_0+2)\lambda^2 L^2}{(1-\lambda)(t+t_0+1)}} \\
&\leq \exp(\nu K_{t+1}D_t),
\end{aligned}
\tag{40}
$$

where we used Proposition C.1 in the first and Lemma G.1 in the second inequality, with $D_t = \frac{5\alpha_t^2 \lambda^2 L^2}{1-\lambda}\Big(\frac{4an\sigma^2\alpha_t}{a\mu-1} + \frac{9\|\nabla \mathbf{f}^\star\|^2}{\mu^2} + \frac{(1+t_0)^{a\mu}\|\mathbf{x}^1 - \mathbf{x}^\star\|^2}{(t+t_0)^{a\mu}}\Big)$. Similarly, we use (39), to get

$$
\begin{aligned}
\Big(\mathbb{E}\big[\exp(\nu K_t\|\widetilde{\mathbf{x}}_t\|^2)\big]\Big)^{\lambda(1-\widetilde{\lambda}/4)\frac{t+t_0+2}{t+t_0+1}} &\leq \exp(\lambda(1-\widetilde{\lambda}/4)S_{t-1}\nu K_{t+1})\Big(\mathbb{E}\big[\exp(\nu K_{t-1}\|\widetilde{\mathbf{x}}_{t-1}\|^2)\big]\Big)^{\lambda^2(1-\widetilde{\lambda}/4)^2\frac{t+t_0+2}{t+t_0}} \\
&\quad \times \bigg(\mathbb{E}\bigg[\exp\Big(\frac{20\alpha_t^2 \lambda^2 L^2 \nu K_t}{(1-\lambda)^2}\|\mathbf{x}_{t-1} - \mathbf{x}^\star\|^2\Big)\bigg]\bigg)^{\lambda(1-\widetilde{\lambda}/4)\widetilde{\lambda}/4}.
\end{aligned}
\tag{41}
$$

Plugging (40) and (41) into (39), we get

$$\mathbb{E}[\exp(\nu K_{t+1}\|\widetilde{\mathbf{x}}^{t+1}\|^2)] \leq \exp\left(\nu K_{t+1}\big(S_t + S_{t-1}\lambda(1-\widetilde{\lambda}/4)\big) + D_t + \lambda(1-\widetilde{\lambda}/4)D_{t-1}\right)$$
$$\times \left(\mathbb{E}\big[\exp(\nu K_{t-1}\|\widetilde{\mathbf{x}}_{t-1}\|^2)\big]\right)^{\lambda^2(1-\widetilde{\lambda}/4)^2 \frac{t+t_0+2}{t+t_0}}.$$

Unrolling the recursion, it follows that

$$\mathbb{E}[\exp(\nu K_{t+1}\|\widetilde{\mathbf{x}}^{t+1}\|^2)] \leq \exp\left(\nu K_{t+1}\Big(\sum_{k=1}^{t}\lambda^{t-k}S_k + \sum_{k=1}^{t}\lambda^{t-k}D_k + \lambda^t\|\widetilde{\mathbf{x}}_1\|^2\Big)\right)$$
$$= \exp\left(\nu K_{t+1}\Big(\sum_{k=1}^{t}\lambda^{t-k}S_k + \sum_{k=1}^{t}\lambda^{t-k}D_k\Big)\right),$$

where the last equality follows from the fact that $x_i^1 = x_j^1$, for all $i, j \in [n]$. □

We are now ready to prove Theorem 3.7. Prior to that, we restate it, for convenience.

**Theorem 3.7.** *Let* **(A1)-(A4)** *and* **(A6)** *hold, the step-size be given by* $\alpha_t = \frac{a}{t+t_0}$ *and let* $x_i^1 = x_j^1$, *for all* $i, j \in [n]$. *If* $a = \frac{6}{\mu}$, $t_0 \geq \max\left\{6, \frac{3+\lambda}{1-\lambda}, \frac{1960\sigma^2\kappa}{\mu}, \frac{432\overline{\sigma}^2\kappa^2}{\mu}, \frac{12\kappa\lambda\sqrt{10}}{1-\lambda}, \frac{5184\overline{\sigma}^2\lambda^2\kappa^2}{\mu(1-\lambda)}\right\}$ *and* $\nu = \min\left\{1, \frac{\mu}{432\sigma^2\kappa^2}, \frac{\mu}{72\kappa}\right\}$, *then for any* $\delta \in (0,1)$ *and* $T \geq 1$, *with probability at least* $1 - \delta$, *it holds that*

$$\frac{1}{n}\sum_{i\in[n]}\big(f(x_i^t) - f^\star\big) = \mathcal{O}\Bigg(\frac{\nu^{-1}\log(2/\delta) + \sigma^2\kappa\big(1+\log(2d)\big)/\mu}{n(t+t_0)} + \frac{\kappa\lambda^2(1+L)(n\sigma^2 + \|\nabla\mathbf{f}^\star\|^{2(1+9\kappa^2)}/(1-\lambda))}{\mu(1-\lambda)n(t+t_0)^2}$$
$$+ \frac{(2+t_0)^3\Delta_f}{n(t+t_0)^3} + \frac{\kappa^3\sigma^2\lambda^2(\kappa\log(t+t_0)+1)}{\mu(1-\lambda)^2(t+t_0)^3} + \frac{\kappa^3\lambda^2 L(1+t_0)\|\mathbf{x}^1 - \mathbf{x}^\star\|^2}{(1-\lambda)^2 n(t+t_0)^3} + \frac{\kappa^2\lambda^2 L(1+t_0)^6\|\mathbf{x}^1 - \mathbf{x}^\star\|^2}{(1-\lambda)^2 n(t+t_0)^8}\Bigg).$$

*Proof.* Starting from Lemma 3.1 and using property 3 of Proposition C.8 with $x = \overline{x}^t$, we have

$$f(\overline{x}^{t+1}) \leq f(\overline{x}^t) - \alpha_t\mu\big(f(\overline{x}^t) - f^\star\big) - \alpha_t\langle\nabla f(\overline{x}^t), \overline{z}_t\rangle + \alpha_t^2 L\|\overline{z}_t\|^2 + \frac{\alpha_t L^2}{2n}\sum_{i\in[n]}\|x_i^t - \overline{x}^t\|^2.$$

Subtracting $f^\star$ from both sides and defining $F_t = n(t+t_0)(f(\overline{x}_t) - f^\star)$, it then follows that

$$F_{t+1} \leq (1-\alpha_t\mu)\frac{t+t_0+1}{t+t_0}F_t - \alpha_t n(t+t_0+1)\langle\nabla f(\overline{x}^t), \overline{z}_t\rangle$$
$$+ \alpha_t^2 n(t+t_0+1)L\|\overline{z}_t\|^2 + \frac{\alpha_t(t+t_0+1)L^2}{2}\|\mathbf{x}^t - \overline{\mathbf{x}}^t\|^2.$$

Next, consider the MGF of $F_{t+1}$ conditioned on $\mathcal{F}_t$. Let $\nu \in (0,1]$ be a positive constant, we then have

$$\mathbb{E}_t\big[\exp(\nu F_{t+1})\big] \overset{(a)}{\leq} \exp\left((1-\alpha_t\mu)\frac{t+t_0+1}{t+t_0}\nu F_t + \frac{\alpha_t\nu(t+t_0+1)L^2}{2}\|\mathbf{x}^t - \overline{\mathbf{x}}^t\|^2\right)$$
$$\times \mathbb{E}_t\Big[\exp\big(-\alpha_t\nu n(t+t_0+1)\langle\nabla f(\overline{x}^t), \overline{z}_t\rangle + \alpha_t^2\nu n(t+t_0+1)L\|\overline{z}_t\|^2\big)\Big]$$
$$\overset{(b)}{\leq} \exp\left((1-\alpha_t\mu)\frac{t+t_0+1}{t+t_0}\nu F_t + \frac{\alpha_t\nu(t+t_0+1)L^2}{2}\|\mathbf{x}^t - \overline{\mathbf{x}}^t\|^2\right)$$
$$\times \sqrt{\mathbb{E}_t\Big[\exp\big(-2\alpha_t\nu n(t+t_0+1)\langle\nabla f(\overline{x}^t), \overline{z}_t\rangle\big)\Big]\mathbb{E}_t\Big[\exp\big(2\alpha_t^2\nu n(t+t_0+1)L\|\overline{z}_t\|^2\big)\Big]}$$
$$\overset{(c)}{\leq} \exp\left((1-\alpha_t\mu)\frac{t+t_0+1}{t+t_0}\nu F_t + \frac{\alpha_t\nu(t+t_0+1)L^2}{2}\|\mathbf{x}^t - \overline{\mathbf{x}}^t\|^2\right)$$
$$\times \exp\left(\frac{3\alpha_t^2\nu^2 n(t+t_0+1)^2\sigma^2\|\nabla f(\overline{x}^t)\|^2}{2} + 15\alpha_t^2\nu\sigma^2(t+t_0+1)L\big(1+\log(2d)\big)\right)$$
$$\overset{(d)}{\leq} \exp\left(\nu\Big(b_t F_t + c_t\|\mathbf{x}^t - \overline{\mathbf{x}}^t\|^2 + d_t\Big)\right),$$

where in $(a)$ we used the fact that $F_t$ and $\|\mathbf{x}^t - \overline{\mathbf{x}}^t\|^2$ are $\mathcal{F}_t$-measurable, $(b)$ follows from Proposition C.4, in $(c)$ we use Lemma 3.2, Proposition C.1 and impose the condition $\alpha_t \leq \frac{1}{\sigma\sqrt{30(t+t_0+1)L}}$, while $(d)$ follows from Proposition C.7 and the definition of $F_t$, with $b_t = \left(1 - \alpha_t\mu + 3\alpha_t^2\nu(t+t_0+1)L\right)\frac{t+t_0+1}{t+t_0}$, $c_t = \frac{\alpha_t(t+t_0+1)L^2}{2}$ and $d_t = 15\alpha_t^2\sigma^2(t+t_0+1)L\left(1 + \log(2d)\right)$. Taking the full expectation and applying Proposition C.4, we get

$$
\begin{aligned}
\mathbb{E}\left[\exp(\nu F_{t+1})\right] &\leq \exp\left(\nu d_t\right)\mathbb{E}\left[\exp\left(\nu b_t F_t + \nu c_t\|\mathbf{x}^t - \overline{\mathbf{x}}^t\|^2\right)\right] \\
&\leq \exp\left(\nu d_t\right)\sqrt[p]{\mathbb{E}\left[\exp\left(\nu p b_t F_t\right)\right]}\sqrt[q]{\mathbb{E}\left[\exp\left(\nu q c_t\|\mathbf{x}^t - \overline{\mathbf{x}}^t\|^2\right)\right]}
\end{aligned}
\tag{42}
$$

for some $p, q \in [1, \infty]$. We next analyze the expression $pb_t$. Recalling the definition of $b_t$, we get

$$
pb_t = p\left(1 - \frac{a\mu}{t+t_0} + \frac{3a^2\nu(t+t_0+1)L}{(t+t_0)^2}\right)\frac{t+t_0+1}{t+t_0} \leq p\left(1 - \frac{a(\mu - 6a\nu L)}{t+t_0}\right)\frac{t+t_0+1}{t+t_0}.
$$

Choosing $\nu \leq \frac{\mu}{12aL}$ and $p = 1 + \frac{\alpha_t\mu}{4}$, it follows that

$$
pb_t \leq p\left(1 - \frac{a\mu}{2(t+t_0)}\right)\frac{t+t_0+1}{t+t_0} \leq \left(1 - \frac{a\mu}{4(t+t_0)}\right)\left(1 + \frac{1}{t+t_0}\right) \leq 1,
\tag{43}
$$

where the last inequality follows since $a\mu > 4$. Next, note that the choice of $p = 1 + \frac{\alpha_t\mu}{4}$ implies that $q = 1 + \frac{4}{\alpha_t\mu}$. From the definition of $c_t$, we then have

$$
qc_t = \left(1 + \frac{4}{\alpha_t\mu}\right)\frac{\alpha_t(t+t_0+1)L^2}{2} = \left(\frac{\alpha_t}{2} + \frac{2}{\mu}\right)(t+t_0+1)L^2 \leq \frac{3L^2}{\mu}(t+t_0+1),
\tag{44}
$$

where the first inequality follows from $\alpha_t \leq \frac{1}{\mu}$. Using (43) and (44) in (42), we get

$$
\begin{aligned}
\mathbb{E}\left[\exp(\nu F_{t+1})\right] &\leq \exp\left(d_t\nu\right)\sqrt[p]{\left(\mathbb{E}\left[\exp(\nu F_t)\right]\right)^{pb_t}}\sqrt[q]{\mathbb{E}\left[\exp\left(\nu(t+t_0+1)3\kappa L\|\mathbf{x}^t - \overline{\mathbf{x}}^t\|^2\right)\right]} \\
&= \exp\left(d_t\nu\right)\left(\mathbb{E}\left[\exp(\nu F_t)\right]\right)^{b_t}\sqrt[q]{\mathbb{E}\left[\exp\left(\nu(t+t_0+1)3\kappa L\|\mathbf{x}^t - \overline{\mathbf{x}}^t\|^2\right)\right]}.
\end{aligned}
\tag{45}
$$

Using Lemma 3.5 with $K = 3\kappa L$, we get

$$
\mathbb{E}\left[\exp\left(\nu q c_t\|\mathbf{x}^t - \overline{\mathbf{x}}^t\|^2\right)\right] \leq \mathbb{E}\left[\exp\left(\nu K_t\|\mathbf{x}^t - \overline{\mathbf{x}}^t\|^2\right)\right] \leq \exp\left(\nu K_t\left(\sum_{k=1}^{t-1}\lambda^{t-1-k}S_k + \sum_{k=1}^{t-1}\lambda^{t-1-k}D_k\right)\right),
$$

where we recall that $S_k = \alpha_k^2\lambda^2\left(n\sigma^2 + \frac{5\|\nabla\mathbf{f}^\star\|^2}{1-\lambda}\right)$ and $D_k = \frac{5\alpha_k^2\lambda^2L^2}{1-\lambda}\left(\frac{4an\sigma^2\alpha_k}{a\mu-1} + \frac{9\|\nabla\mathbf{f}^\star\|^2}{\mu^2} + \frac{(1+t_0)^{a\mu}\|\mathbf{x}^1-\mathbf{x}^\star\|^2}{(k+t_0)^{a\mu}}\right)$. To further bound the above expression, we use Lemma D.3, to get

$$
\begin{aligned}
\sum_{k=1}^{t-1}\lambda^{t-1-k}(S_k + D_k) &\leq \frac{4a^2\lambda^2\left(n\sigma^2 + 5\|\nabla\mathbf{f}^\star\|^2(1+9L^2/\mu^2)/(1-\lambda)\right)}{(1-\lambda)(t+t_0)^2} \\
&\quad + \frac{32a^4n\sigma^2\lambda^2L^2}{(1-\lambda)^2(t+t_0)^3} + \frac{20a^2\lambda^2L^2(1+t_0)^6\|\mathbf{x}^1-\mathbf{x}^\star\|^2}{(1-\lambda)^2(t+t_0)^8}.
\end{aligned}
$$

Noting that $\frac{1}{q} = \frac{\alpha_t\mu}{4+\alpha_t\mu} \leq \frac{\alpha_t\mu}{4}$, we finally get

$$
\sqrt[q]{\mathbb{E}\left[\exp\left(\nu q c_t\|\mathbf{x}^t - \overline{\mathbf{x}}^t\|^2\right)\right]} \leq \exp\left(\frac{\nu K_t\alpha_t\mu}{4}N_t\right) \leq \exp\left(\frac{3aL^2\nu N_t}{2}\right),
$$

where $N_t := \frac{4a^2\lambda^2(n\sigma^2+5\|\nabla\mathbf{f}^\star\|^2(1+9L^2/\mu^2)/(1-\lambda))}{(1-\lambda)(t+t_0)^2} + \frac{32a^4n\sigma^2\lambda^2L^2}{(1-\lambda)^2(t+t_0)^3} + \frac{20a^2\lambda^2L^2(1+t_0)^6\|\mathbf{x}^1-\mathbf{x}^\star\|^2}{(1-\lambda)^2(t+t_0)^8}$. Define $G_1 := \frac{1080\kappa\sigma^2(1+\log(2d))}{\mu}$, $G_2 := \frac{1296\kappa^2\lambda^2(n\sigma^2+5\|\nabla\mathbf{f}^\star\|^2(1+9\kappa^2)/(1-\lambda))}{\mu(1-\lambda)}$, $G_4 := \frac{6480\kappa^3\lambda^2L(1+t_0)^6\|\mathbf{x}^1-\mathbf{x}^\star\|^2}{(1-\lambda)^2}$ and $G_3 := \frac{373248n\kappa^4\sigma^2\lambda^2}{\mu(1-\lambda)^2}$, and plug into (45), to get

$$
\mathbb{E}\left[\exp\left(\nu F_{t+1}\right)\right] \leq \left(\mathbb{E}\left[\exp\left(\nu F_t\right)\right]\right)^{b_t}\exp\left(\sum_{i\in[3]}\frac{\nu G_i}{(t+t_0)^i} + \frac{\nu G_4}{(t+t_0)^8}\right).
$$

Recalling the definition of $b_t$ and (43), it follows that $b_t \leq 1 - \frac{a\mu/2-1}{t+t_0} = 1 - \frac{2}{t+t_0}$, therefore we can bound the MGF of $\nu F_{t+1}$ using Lemma 3.6 with $a = 2$, $M = 8$, $C_i = G_i$ for $i \in [3]$, $C_8 = G_4$ and $C_j = 0$, for $j \in \{4, \ldots, 7\}$, to finally get

$$\mathbb{E}\big[\exp\big(\nu F_{t+1}\big)\big] \leq \exp\left(\frac{(t_0+2)^3\nu\Delta_f}{(t+1+t_0)^2} + 4\nu G_1 + \frac{4\nu G_2}{t+1+t_0}\right)$$
$$\times \exp\left(\frac{4\nu G_3\log(t+t_0+1)}{(t+t_0+1)^2} + \frac{4\nu G_4}{5(t_0+1)^5(t+1+t_0)^2}\right). \tag{46}$$

Applying Markov's inequality, we then get, for any $\epsilon > 0$

$$\mathbb{P}\left(f(\overline{x}^{t+1}) - f^\star > \epsilon\right) = \mathbb{P}\big(\nu F_{t+1} > \nu n(t+1+t_0)\epsilon\big) \leq \exp\left(-\nu n(t+1+t_0)\epsilon\right)\mathbb{E}\big[\big(\nu F_{t+1}\big)\big].$$

Using (46), it can be readily verified that for any $\delta \in (0,1)$, choosing

$$\epsilon_t^1 = \frac{\nu^{-1}\log(1/\delta) + 4G_1}{n(t+t_0)} + \frac{4G_2}{n(t+t_0)^2} + \frac{(t_0+2)^3\Delta_f + 4G_3\log(t+t_0) + 4G_4/5(t_0+1)^5}{n(t+t_0)^3}, \tag{47}$$

results in $\mathbb{P}\left(f(\overline{x}^t) - f^\star > \epsilon_t^1\right) \leq \delta$. Next, using Proposition C.7 with $x = x_i^t$ and $y = \overline{x}^t$, we get

$$f(x_i^t) \leq f(\overline{x}^t) + \langle \nabla f(\overline{x}^t), x_i^t - \overline{x}^t \rangle + \frac{L}{2}\|x_i^t - \overline{x}^t\|^2$$
$$\overset{(i)}{\leq} f(\overline{x}^t) + \frac{1}{2L}\|\nabla f(\overline{x}^t)\|^2 + \frac{L}{2}\|x_i^t - \overline{x}^t\|^2 + \frac{L}{2}\|x_i^t - \overline{x}^t\|^2$$
$$\overset{(ii)}{\leq} f(\overline{x}^t) + f(\overline{x}^t) - f^\star + L\|x_i^t - \overline{x}^t\|^2,$$

where in $(i)$ we used Proposition C.3 with $\epsilon = L$, while $(ii)$ follows from Proposition C.7. Subtracting $f^\star$ from both sides and averaging over all users $i \in [n]$, we get

$$\frac{1}{n}\sum_{i\in[n]}\left(f(x_i^t) - f^\star\right) \leq 2\big(f(\overline{x}^t) - f^\star\big) + \frac{L}{n}\|\mathbf{x}^t - \overline{\mathbf{x}}^t\|^2. \tag{48}$$

We now consider two events, $A_{t,\epsilon} \coloneqq \left\{\omega : f(\overline{x}^t) - f^\star > \epsilon\right\}$ and $B_{t,\epsilon} \coloneqq \left\{\omega : \frac{L}{n}\|\mathbf{x}^t - \overline{\mathbf{x}}^t\|^2 > \epsilon\right\}$. From the previous analysis, we know that, for any $\delta \in (0,1)$ and $\epsilon_t^1$ from (47), we have $\mathbb{P}\big(A_{t,\epsilon_t^1}\big) \leq \delta$. Similarly, using Markov's inequality and Lemma 3.5 with $K = L$, we have, for any $\epsilon > 0$

$$\mathbb{P}\left(\frac{L}{n}\|\mathbf{x}^t - \overline{\mathbf{x}}^t\|^2 > \epsilon\right) = \mathbb{P}\left(\nu K_t\|\mathbf{x}^t - \overline{\mathbf{x}}^t\|^2 > \nu n(t+t_0+1)\epsilon\right)$$
$$\leq \exp(-\epsilon\nu n(t+t_0+1))\mathbb{E}\big[\exp\big(\nu K_t\|\mathbf{x}^t - \overline{\mathbf{x}}^t\|^2\big)\big]$$
$$\leq \exp\left(-\epsilon\nu n(t+t_0+1) + K_t\nu\left(\frac{4a^2\lambda^2(n\sigma^2 + 5\|\nabla\mathbf{f}^\star\|^2(1+9L^2/\mu^2)/(1-\lambda))}{(1-\lambda)(t+t_0)^2}\right)\right)$$
$$\times \exp\left(K_t\nu\left(\frac{32a^4n\sigma^2\lambda^2 L^2}{(1-\lambda)^2(t+t_0)^3} + \frac{20a^2\lambda^2 L^2(1+t_0)^6\|\mathbf{x}^1 - \mathbf{x}^\star\|^2}{(1-\lambda)^2(t+t_0)^8}\right)\right).$$

Therefore, it can be readily seen that for any $\delta \in (0,1)$, choosing

$$\epsilon_t^2 = \frac{\nu^{-1}\log(1/\delta)}{n(t+t_0+1)} + \frac{144\kappa\lambda^2(n\sigma^2 + 5\|\nabla\mathbf{f}^\star\|^2(1+9\kappa^2)/(1-\lambda))}{\mu(1-\lambda)n(t+t_0)^2}$$
$$+ \frac{41472\kappa^3\sigma^2\lambda^2}{\mu(1-\lambda)^2(t+t_0)^3} + \frac{720\kappa^2\lambda^2 L(1+t_0)^6\|\mathbf{x}^1 - \mathbf{x}^\star\|^2}{(1-\lambda)^2 n(t+t_0)^8}, \tag{49}$$

results in $\mathbb{P}\big(B_{t,\epsilon_t^2}\big) \leq \delta$. Finally, let $C_t \coloneqq \left\{\omega : \frac{1}{n}\sum_{i\in[n]}\left(f(x_i^t) - f^\star\right) > 2\epsilon_t^1 + \epsilon_t^2\right\}$. From (48) it readily follows that, for any $\delta \in (0, 1/2)$, we have

$$\mathbb{P}(C_t) \leq \mathbb{P}\big(A_{t,\epsilon_t^1} \cap B_{t,\epsilon_t^2}\big) \leq \mathbb{P}\big(A_{t,\epsilon_t^1}\big) + \mathbb{P}\big(B_{t,\epsilon_t^2}\big) \leq 2\delta.$$

Therefore, for any $\delta \in (0, 1)$, with probability at least $1 - \delta$, we get

$$\frac{1}{n} \sum_{i \in [n]} \left( f(x_i^t) - f^\star \right) = \mathcal{O}\left( \frac{\nu^{-1} \log(2/\delta) + \sigma^2 \kappa (1 + \log(2d))/\mu}{n(t + t_0)} + \frac{\kappa \lambda^2 (1 + L)(n\sigma^2 + \|\nabla \mathbf{f}^\star\|^{2(1+9\kappa^2)/(1-\lambda)})}{\mu(1 - \lambda)n(t + t_0)^2} \right.$$

$$\left. + \frac{(2 + t_0)^3 \Delta_f}{n(t + t_0)^3} + \frac{\kappa^3 \sigma^2 \lambda^2 (\kappa \log(t + t_0) + 1)}{\mu(1 - \lambda)^2 (t + t_0)^3} + \frac{\kappa^3 \lambda^2 L (1 + t_0) \|\mathbf{x}^1 - \mathbf{x}^\star\|^2}{(1 - \lambda)^2 n(t + t_0)^3} + \frac{\kappa^2 \lambda^2 L (1 + t_0)^6 \|\mathbf{x}^1 - \mathbf{x}^\star\|^2}{(1 - \lambda)^2 n(t + t_0)^8} \right).$$

Finally, it can be verified that the conditions on $a$, $t_0$ and $\nu$ in the statement of the theorem ensure that all the step-size conditions are satisfied, completing the proof. $\qquad\square$

We remark that, similarly to the proof of Theorem 3.4, one can analyze the strongly convex case with a fixed step-size, resulting in the dependence on some terms, e.g., optimality and iterate gaps $\Delta_f$ and $\|\mathbf{x}^1 - \mathbf{x}^\star\|$, decaying exponentially fast, i.e., $\mathcal{O}\left( (\|\mathbf{x}^1 - \mathbf{x}^\star\|^2 + \Delta_f)e^{-CT} \right)$, for some $C > 0$, as shown in, e.g., (Koloskova et al., 2020) for MSE guarantees in decentralized, or (Liu & Zhou, 2024) for HP guarantees in centralized settings. For simplicity, we omit this analysis.

# H. Numerical Experiments

In this section we provide some further numerical results and details omitted from the main body. Subsection H.1 presents results on synthetic data, while Subsection H.2 presents results on real data.

## H.1. Synthetic Data

**Methodology.** We consider an instance of (2), with $f_i(x) = \frac{1}{2} x^\top A_i x + b_i^\top x$, where $A_i \in \mathbb{R}^{d \times d}$ is positive definite, making each $f_i$ strongly convex, with *unbounded gradients*. We consider an undirected communication network $G = (V, E)$, corresponding to a randomly generated Erdős–Rényi graph with connectivity parameter $p = 0.8$, while the weight matrix $W \in \mathbb{R}^{n \times n}$ is computed using the Metropolis-Hastings weight scheme, e.g., (Xiao & Boyd, 2004). When queried by user $i \in [n]$ in iteration $t \geq 1$, the $\mathcal{SFO}$ returns a noisy gradient of $f_i$ evaluated at $x_i^t$, i.e., $g_i^t = A_i x_i^t + b_i + z_i^t$, where $z_i^t \in \mathbb{R}^d$ is a zero-mean Gaussian random vector, making the noise consistent with assumption **(A4)**. We use the **DSGD** method outlined in Algorithm 1, with the time-varying step-size schedule $\alpha_t = \frac{1}{t+1}$ and shared initializatio $x_i^1 = 0$, for all $i \in [n]$. We are interested in testing the following two facets of our theory.

1. *Exponentially decaying tails* - we want to verify that the tail probability decays at an exponential scale, as defined in (1) and predicted in Theorem 3.7.

2. *Linear speed-up* - we want to verify that the tail probability decays faster as the number of users increases, as predicted in Theorem 3.7.

To verify these two facets, we measure the performance in terms of the *empirical tail probability* $\mathbb{P}_{n,\varepsilon}^t$, computed as follows. We first run **DSGD** for $T$ iterations and repeat it over $R$ runs. Next, for each $t \in [T]$, we use Monte-Carlo sampling to create a set $S_t$ of indices from $[R]$, of some fixed length $|S_t| = S$. For any $\varepsilon > 0$, the empirical probability is computed as

$$\mathbb{P}_{n,\varepsilon}^t = \frac{1}{S} \sum_{r \in S_t} \mathbb{I}\left( \frac{1}{n} \sum_{i \in [n]} \|x_i^{t,r} - x^\star\|^2 > \varepsilon \right),$$

where $x_i^{t,r} \in \mathbb{R}^d$ is the model of user $i$ in iteration $t$ and run $r$, with $x^\star = \arg\min_{x \in \mathbb{R}^d} f(x)$ being the solution of the global problem and $\mathbb{I}(A)$ being the indicator of event $A$. The empirical tail probability is a proxy to the true tail probability and is the main metric in our experiments. To further illustrate our results, we also compute the *empirical mean-squared error* $\mathbb{E}_n^t$, which is the optimality gap at time $t$, averaged across all users and runs, i.e., $\mathbb{E}_n^t = \frac{1}{nR} \sum_{i \in [n]} \sum_{r \in [R]} \|x_i^{t,r} - x^\star\|^2$. We next present the results.

**Exponentially decaying tails.** To verify that the (empirical) tail probability decays at an exponential rate, we consider a fixed network of $n = 30$ users. For local costs, each matrix $A_i$ is generated using python's `sklearn` library function `make_sparse_spd_matrix`, with dimension $d = 50$ and value `alpha = 0.9`, while vectors $b_i$ are drawn from a multivariate normal distribution $\mathcal{N}(\mathbf{0}_d, \sigma_i^2 I_d)$, where $\sigma_i^2 = 1$ for $i \in \{1, \ldots, 10\}$, $\sigma_i^2 = 2$ for $i \in \{11, \ldots, 20\}$, and $\sigma_i^2 = 4$

for $i \in \{21, \ldots, 30\}$, ensuring that the data is heterogeneous across users. We run **DSGD** for $T = 10000$ iterations and repeat across $R = 5000$ runs. To compute the empirical probability, in each iteration we draw $S = 3000$ Monte-Carlo samples and consider threshold values $\varepsilon = \left\{10^{-2}, 10^{-3}, 10^{-4}\right\}$. The results are presented in Figure 2, where the left plot shows the MSE behaviour, while the right plot shows the tail probability behaviour. We can see that the empirical tail probability decays exponentially fast for all values of $\varepsilon$, as predicted by our theory. Note that for the threshold value $\varepsilon = 10^{-4}$, the tail probability starts decaying exponentially after approximately $t = 6000$ iterations, which is consistent with the MSE behaviour on the left plot, where we can see that it takes **DSGD** around the same number of iterations to reach the average accuracy $\mathbb{E}_n^t = 10^{-4}$.

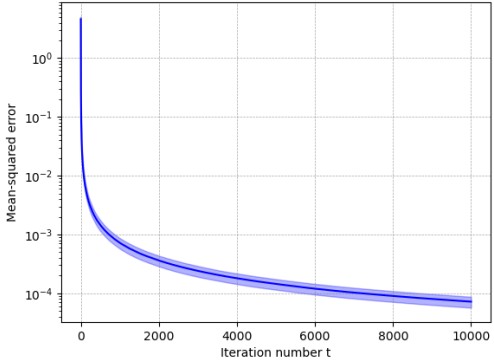 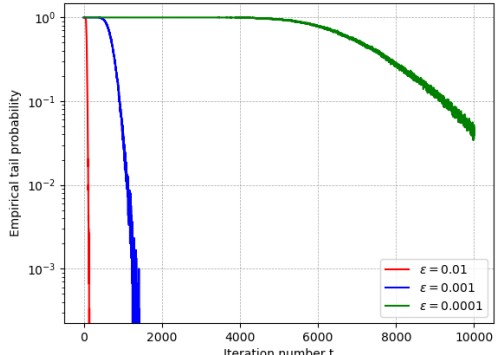

*Figure 2.* Performance of **DSGD**, in the MSE sense (left) and HP sense (right). We can see that **DSGD** achieves an exponential tail decay for all values of threshold $\varepsilon$. For the threshold $\varepsilon = 10^{-4}$, the tail probability starts decaying exponentially after approximately $t = 6000$ iterations, which is consistent with the MSE behaviour, where we can see that **DSGD** takes around the same number of iterations to reach the average accuracy $\mathbb{E}_n^t = 10^{-4}$.

**Linear speed-up.** To verify that linear speed-up in the number of users is achieved, we consider three networks, with $n = \{25, 50, 100\}$ users. In all three cases, we use Erdős–Rényi communications graphs with Metropolis-Hastings weights that satisfy $\lambda = \|W - J\| \approx 0.6$. To ensure that the effect of heterogeneity is consistent across different networks, we fix the matrices $A_i$, i.e., $A_i \equiv A + I_d$, where $A \in \mathbb{R}^{d \times d}$ is generated using sklearn's `make_sparse_spd_matrix`, with $d = 50$ and `alpha = 0.9`, while vectors $b_i$ are given by $b_i = \beta_i \mathbf{1}_d$, where $\beta_i \in \{-2, -1, 0, 1, 3\}$ are selected uniformly at random and in equal proportion across users (i.e., one fifth of users has $\beta_i = -2$, one fifth has $\beta_i = -1$, etc). Generating the networks and costs in this manner ensures that both network connectivity and user heterogeneity are constant across all three network settings, allowing us to properly capture the effect of linear speed-up. We run **DSGD** for $T = 3000$ iterations and repeat across $R = 1000$ runs. To compute the empirical probability, in each iteration we draw $S = 600$ Monte-Carlo samples, with thresholds $\varepsilon = \left\{10^{-2}, 10^{-3}, 10^{-4}\right\}$. The results are presented in Figure 3, where the plots left to right and top to bottom respectively show the MSE and tail probability behaviours for different values of $\varepsilon$. We can again see that the empirical tail probability decays exponentially fast for all values of $\varepsilon$, with the decay being consistently faster for larger number of users, demonstrating linear speed-up in the HP sense. Finally, note that in the lower right plot (i.e., threshold $\varepsilon = 10^{-4}$) the network with $n = 25$ users has a constant tail probability $\mathbb{P}_{n,\varepsilon}^t = 1$, while for the network with $n = 50$ the tail probability is slowly starting to decrease, which is again consistent with the MSE behaviour on the upper left plot, as the accuracy $\mathbb{E}_n^t = 10^{-4}$ for these two networks is not reached (on average) in the allocated number of iterations.

## H.2. Real Data

**Methodology.** We validate our theory on a non-convex problem, considering a binary logistic regression classification task with a non-convex regularizer, e.g., (Antoniadis et al., 2011), with users' local costs given by

$$f_i(x) = \frac{1}{m_i} \sum_{r \in [m_i]} \log\left(1 + \exp(-y_{i,r} \langle h_{i,r}, x \rangle)\right) + \eta \sum_{k \in [d]} \frac{[x]_k^2}{1 + [x]_k^2},$$

where $h_{i,r} \in \mathbb{R}^d$ and $y_{i,r} \in \{+1, -1\}$ are the feature vector and associated label, $\eta > 0$ is a user-specified penalty parameter, while $[x]_k$ denotes the $k$-th component of the model vector $x$. To evaluate the performance, we use the "mushroom",

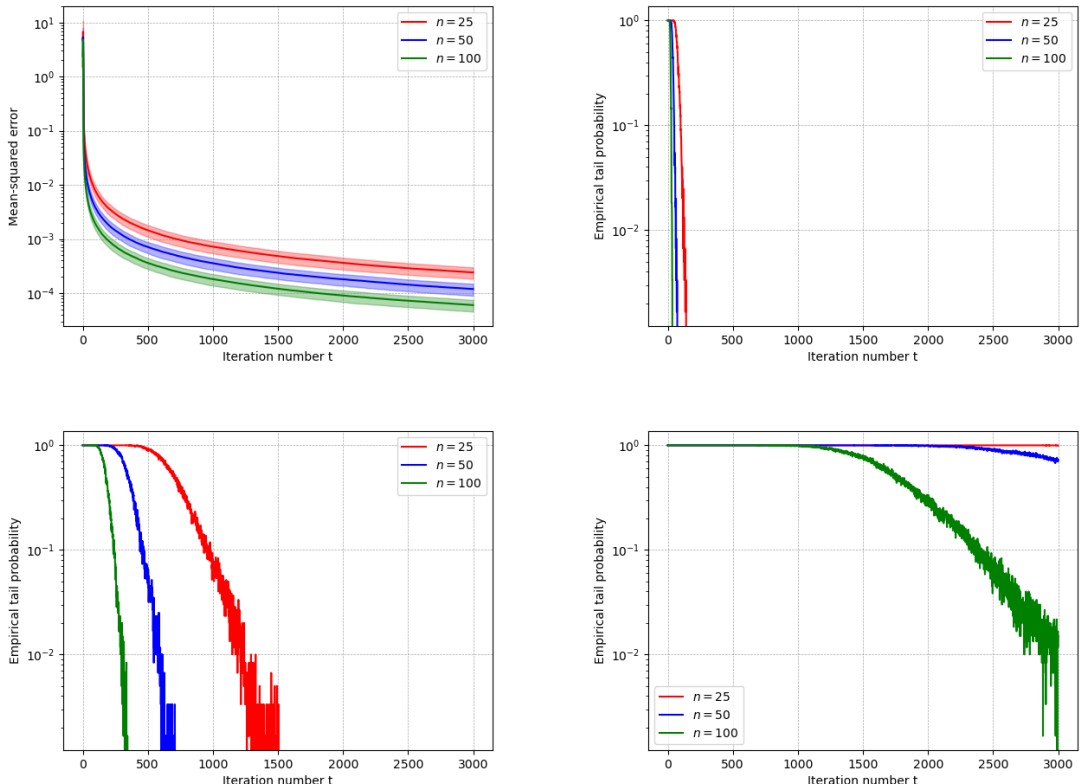

*Figure 3.* Linear speed-up of **DSGD**, in the MSE and HP sense. Left to right and top to bottom: MSE performance and tail decay with threshold $\varepsilon = \left\{ 10^{-2}, 10^{-3}, 10^{-4} \right\}$. We can see that **DSGD** consistently achieves faster exponential tail decay for larger networks, across all values of threshold $\varepsilon$, illustrating the effect of the linear speed-up in the HP sense.

"a9a" and "ijcnn1" datasets from the LIBSVM library (Chang & Lin, 2011), each providing a varying degree of heterogeneity. We split the data uniformly across agents, so that all agents have an equal-sized local dataset before training, i.e., $m_i = \frac{m}{n}$, where $m$ is the size of the original dataset. Similarly to synthetic data experiments, we use Erdős-Rényi graphs with Metropolis-Hastings weights. For each experiment, we fix the following parameters: step-size $\alpha = 0.1$ and penalty parameter $\eta = 0.1$, giving a large learning rate and a non-trivial effect of the non-convex regularizer. Since the gap to the global minima is not computable in the non-convex case, we evaluate the performance using the average gradient norm-squared, i.e., $G_n^{t,r} = \frac{1}{nt} \sum_{\tau \in [t]} \sum_{i \in [n]} \| \nabla f(x_i^{\tau,r}) \|^2$, so that the empirical tail probability is computed as $\mathbb{P}_{n,\epsilon}^t = \frac{1}{R} \sum_{r \in [R]} \mathbb{I}\big( G_n^{t,r} > \epsilon \big)$. Similarly to the previous section, we also compute and visualize the average performance across all runs, i.e., $\mathbb{E}_n^t = \frac{1}{R} \sum_{r \in [R]} G_n^{t,r}$. We note that in our experiments on real data, the stochastic noise comes from the mini-batch choice, hence the resulting noise is *not necessarily sub-Gaussian*.

**Exponentially decaying tails.** We start by testing the decay rate of the empirical tail probability. We fix a network of $n = 30$ agents communicating over an Erdős-Rényi graph. We run both methods for $T = 1000$ iterations, repeated across $R = 100$ runs. The results are presented in Figure 4, where the top and bottom rows respectively correspond to "a9a" and "mushroom" datasets, while figures left to right visualizes the average error $\mathbb{E}_n^t$ and the empirical tail probability $\mathbb{P}_{n,\epsilon}^t$. The threshold values are chosen as $\varepsilon = \{0.05, 0.02, 0.008\}$ for "a9a" and $\varepsilon = \{0.05, 0.03, 0.019\}$ for "mushroom" datasets, based on the achieved average error. As can be seen from the figure, **DSGD** achieves exponential tail decay across both datasets and all values of threshold $\varepsilon$.

**Linear speedup.** To verify that **DSGD** achieves linear speed-up, we again consider three networks with $n = \{10, 30, 50\}$ agents, communicating over Erdős–Rényi graphs. To ensure that the network connectivity is consistent, we enforce the condition $\lambda = \|W - J\| \approx 0.6$ on the resulting weight matrices, with $\alpha, T, R$ remaining unchanged. We consider the "a9a"

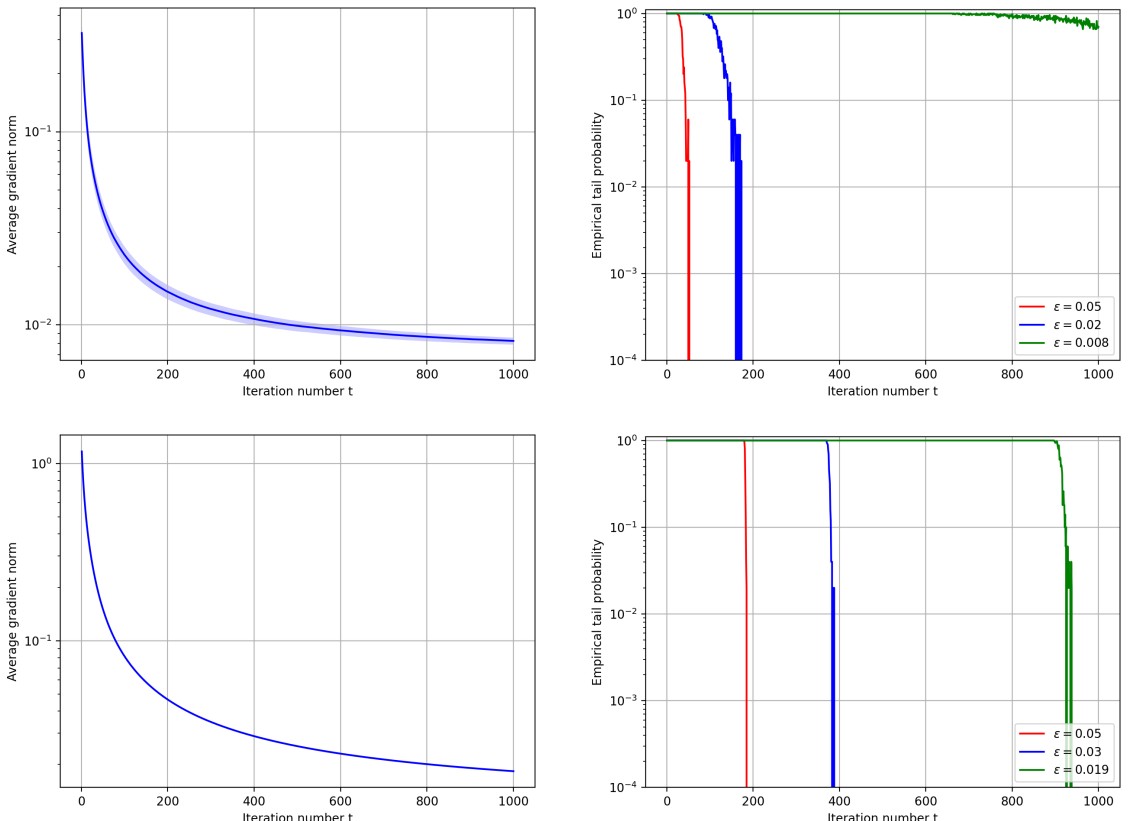

*Figure 4.* Exponential tail decay on real data. Left to right: average gradient norm across all runs and its empirical tail probability. Top to bottom: performance on "`a9a`" and "`mushroom`" datasets. For the empirical tail probability, we use threshold values $\epsilon = \{0.01, 0.05, 0.01\}$ for the respective datasets. We can see that **DSGD** consistently achieves exponentially decaying tails, across both datasets and different threshold values.

and "`ijcnn1`" datasets, plotting the empirical tail probability versus the number of users for different error thresholds. In particular, we use the thresholds $\varepsilon = \{0.05, 0.02, 0.01\}$ for "`a9a`" and $\varepsilon = \{0.005, 0.0035, 0.003\}$ for "`ijcnn1`" dataset. The results are presented in Figures 5 and 6, respectively. We can again see that the average gradient norm and empirical tail probability decay faster with larger $n$ across both datasets and different values of $\varepsilon$, with the speed-up effect more evident on the more heterogeneous "`a9a`" dataset.

## I. Deriving the Transient Time

In this section we provide details on the transient time resulting from our HP bounds. Similarly to Alghunaim & Yuan (2022), we focus on the dependence on network connectivity and number of users, ignoring other problem related constants.

**Transient time for non-convex costs.** Recall that the bound in Theorem 3.4 is of the form

$$\mathcal{O}\left( \frac{1}{\sqrt{nT}} + \frac{1}{CT} + \frac{\Delta_x}{(1-\lambda^2)T} + \frac{n}{(1-\lambda)^2 T} \right), \tag{50}$$

where the problem related constant $C$ depends on the network connectivity via the expression $C = \mathcal{O}\left( \min\left\{ \frac{1-\lambda}{\lambda}, \frac{\sqrt[3]{n}(1-\lambda)^{2/3}}{\lambda^2} \right\} \right)$. Since we are interested in the worst-case transient time (corresponding to poor network connectivity, i.e., $\lambda \approx 1$), it follows that $C = \mathcal{O}\left( \frac{1-\lambda}{\lambda} \right)$. For simplicity, assume that users have a shared initialization, i.e., $x_i^1 = x_j^1$, for all $i, j \in [n]$, so that $\Delta_x = 0$. In this case, the bound in (50) becomes

$$\mathcal{O}\left( \frac{1}{\sqrt{nT}} + \frac{1}{(1-\lambda)T} + \frac{n}{(1-\lambda)^2 T} \right). \tag{51}$$

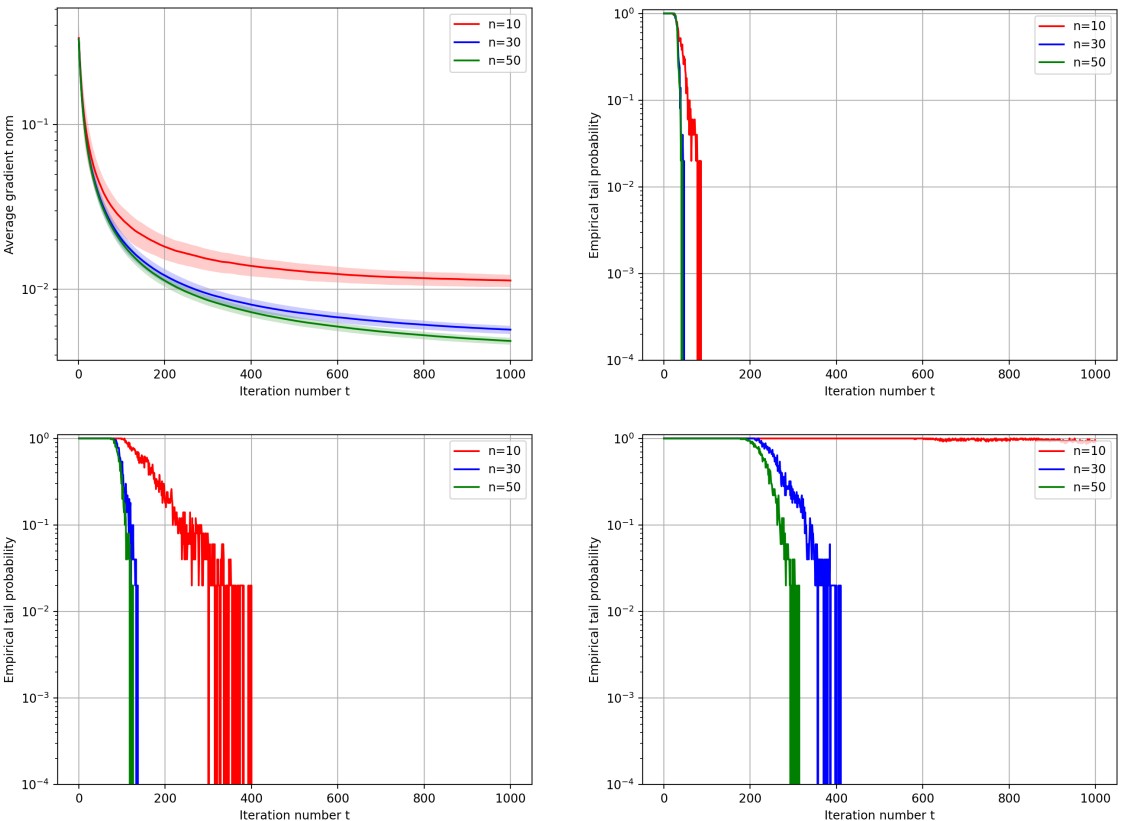

*Figure 5.* Linear speed-up of **DSGD** on the "a9a" dataset. Left to right and top to bottom: MSE performance and tail decay with threshold $\varepsilon = \{0.05, 0.02, 0.01\}$. We can see that **DSGD** consistently achieves faster exponential tail decay for larger networks, across all values of threshold $\varepsilon$, illustrating the effect of the linear speed-up in the HP sense.

It is now evident that, for any $T \geq \max\left\{\frac{n}{(1-\lambda)^2}, \frac{n^3}{(1-\lambda)^4}\right\} = \frac{n^3}{(1-\lambda)^4}$, the bound in (51) becomes $\mathcal{O}\left(\frac{1}{\sqrt{nT}}\right)$, implying a transient time of at most $\mathcal{O}\left(\frac{n^3}{(1-\lambda)^4}\right)$. This matches the transient time of **DSGD** obtained from the MSE rate in (Koloskova et al., 2020), see, e.g., Table I in (Alghunaim & Yuan, 2022).

**Transient time for strongly convex costs.** Recalling the expressions in (47) and (49), it follows that the full bound for strongly convex costs is of the form

$$\mathcal{O}\Bigg(\frac{1}{n(T+t_0)} + \frac{1}{(1-\lambda)(T+t_0)^2} + \frac{1}{n(1-\lambda)^2(T+t_0)^2} + \frac{t_0^3}{n(T+t_0)^3}$$
$$+ \frac{\log(T+t_0)}{(1-\lambda)^2(T+t_0)^3} + \frac{t_0}{n(1-\lambda)^2(T+t_0)^3} + \frac{t_0^6}{n(1-\lambda)^2(T+t_0)^8}\Bigg), \tag{52}$$

where $t_0$ depends on the network connectivity via the expression $t_0 = \Omega\left(\max\left\{\frac{\lambda}{1-\lambda}, \frac{\lambda^2}{1-\lambda}, \frac{1}{1-\lambda}\right\}\right)$. Since we are again interested in the worst-case transient time, it follows that $t_0 = \Omega\left(\frac{1}{1-\lambda}\right)$, so that the bound in (52) becomes

$$\mathcal{O}\Bigg(\frac{1}{n(T+t_0)} + \frac{1}{(1-\lambda)(T+t_0)^2} + \frac{1}{n(1-\lambda)^2(T+t_0)^2}$$
$$+ \frac{\log(T+t_0)}{(1-\lambda)^2(T+t_0)^3} + \frac{1}{n(1-\lambda)^3(T+t_0)^3} + \frac{1}{n(1-\lambda)^8(T+t_0)^8}\Bigg). \tag{53}$$

It can be seen that for $T \geq \max\left\{\frac{n}{1-\lambda}, \frac{1}{(1-\lambda)^2}, \frac{\sqrt{n}}{1-\lambda}, \frac{1}{(1-\lambda)^{\frac{3}{2}}}, \frac{1}{(1-\lambda)^{\frac{8}{7}}}\right\} = \max\left\{\frac{n}{1-\lambda}, \frac{1}{(1-\lambda)^2}\right\}$, the bound in (53) becomes

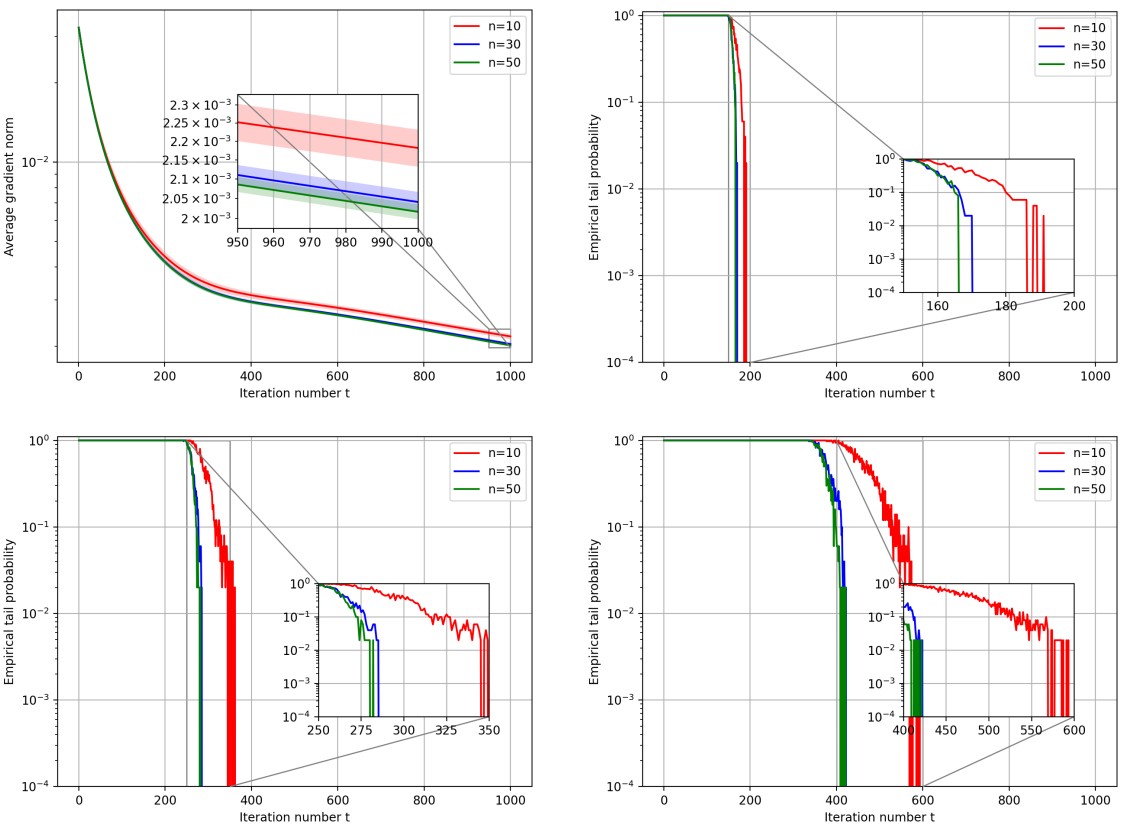

*Figure 6.* Linear speed-up of **DSGD** on the "`ijcnn1`" dataset. Left to right and top to bottom: MSE performance and tail decay with threshold $\varepsilon = \{0.005, 0.0035, 0.003\}$. We can see that **DSGD** consistently achieves faster exponential tail decay for larger networks, across all values of threshold $\varepsilon$, illustrating the effect of the linear speed-up in the HP sense.

$\mathcal{O}\left(\frac{\log(T+t_0)}{n(T+t_0)}\right)$, implying a transient time of at most $\mathcal{O}\left(\max\left\{\frac{n}{1-\lambda}, \frac{1}{(1-\lambda)^2}\right\}\right)$. Interestingly, this is strictly sharper than the transient time of **DSGD** obtained from the MSE rate in (Koloskova et al., 2020), which is of order $\mathcal{O}\left(\frac{n}{(1-\lambda)^2}\right)$, see, e.g., Table II in (Alghunaim & Yuan, 2022). As such, our HP analysis provides either matching or strictly sharper transient times for **DSGD** compared to the transient times stemming from MSE results, further highlighting the tightness of our analysis.

## J. Achieving Linear Speed-up

In this section we provide further discussions on the main results and techniques needed to obtain linear speed-up.

**Linear speed-up for non-convex costs.** The crucial result for obtaining linear speed for non-convex costs is Lemma 3.2, which establishes the variance reduction benefits of decentralized methods in the HP sense. To see why this is the case, note that using a fixed step-size $\alpha_t \equiv \alpha$ and following the same steps as in the proof of Theorem 3.4, while relying only on assumption **(A4)** to control the noise, one would get a final bound where the leading term is of the form

$$\mathcal{O}\left(\frac{1}{\alpha T} + \alpha\sigma^2\right). \tag{54}$$

It is obvious that no choice of $\alpha$ leading to a linear speed-up is possible, i.e., such that (54) becomes $\mathcal{O}\left(\frac{1}{\sqrt{nT}}\right)$. On the other hand, using Lemma 3.2 to handle the noise, we are able to establish a bound where the leading term is of the form

$$\mathcal{O}\left(\frac{1}{\alpha T} + \frac{\alpha\sigma^2}{n}\right), \tag{55}$$

where it is easy to see that choosing $\alpha = \Theta\left(\frac{\sqrt{n}}{\sqrt{T}}\right)$ leads to the desired linear speed-up. Works like (Lu et al., 2024; Lu, 2024; Xu et al., 2024) provide no results on the variance reduction benefit of decentralized learning in the HP sense, instead relying only on the definition of sub-Gaussianity.[12] As such, even if a fixed step-size was used, the authors would be unable to obtain a bound of the form (55) and achieve linear speed-up. As discussed in Appendix F, using a time-varying step-size for non-convex costs inevitably leads to a loss of linear speed-up, as all terms decay at the same rate, leading to the bound

$$\mathcal{O}\left(\frac{1}{\sqrt{nT}} + \frac{n}{\sqrt{T}}\right).$$

On the other hand, using a fixed step-size $\alpha = \Theta\left(\frac{\sqrt{n}}{\sqrt{T}}\right)$, we get the bound

$$\mathcal{O}\left(\frac{1}{\sqrt{nT}} + \frac{n}{T}\right),$$

where the term not achieving linear speed-up is of higher order. This is consistent with MSE guarantees, e.g., (Koloskova et al., 2020; Xin et al., 2021; Alghunaim & Yuan, 2022), where a fixed step-size is also needed to achieve linear speed-up.

**Linear speed-up for strongly convex costs.** In addition to Lemma 3.2, a tighter control of the MGF of the process of interest is required, which is achieved in Lemma 3.6. To see why this is the case, consider the existing techniques in centralized settings, e.g., (Harvey et al., 2019; Bajović et al., 2023), where the authors show via induction that the MGF of the process of interest is uniformly bounded, i.e., that there exists a $B > 0$, such that, for all $t \geq 1$ and $\nu \in (0, B]$

$$\mathbb{E}[\exp(\nu F^t)] \leq \exp\left(\frac{\nu}{B}\right). \tag{56}$$

Recalling that $F^t = n(t + t_0)\big(f(\overline{x}^t) - f^\star\big)$, it then follows from (56) and Markov's inequality that, for any $\epsilon > 0$

$$\mathbb{P}\left(f(\overline{x}^t) - f^\star > \frac{\epsilon + 1}{n\nu(t + t_0)}\right) = \mathbb{P}\big(\nu F^t > \epsilon + 1\big) \leq \exp(-\epsilon).$$

Choosing $\nu = B$, we then get for any $\delta \in (0, 1)$, with probability at least $1 - \delta$

$$f(\overline{x}^t) - f^\star \leq \frac{\log(1/\delta) + 1}{nB(t + t_0)}. \tag{57}$$

While the above bound seemingly achieves linear speed-up, we now show that the constant $B$ must depend on $n$, effectively canceling the linear speed-up. To see that this is the case, we go back to (56) and note that for $t = 1$, we have

$$\mathbb{E}[\exp(\nu F^1)] = \exp\left(\nu n(t_0 + 1)\big(f(\overline{x}^1) - f^\star\big)\right) \leq \exp\left(\frac{\nu}{B}\right),$$

where the equality follows from the definition of $F^t$ and the fact that the initial models are chosen deterministically, while the inequality follows if and only if $B \leq \frac{1}{n(t_0+1)(f(\overline{x}^1)-f^\star)}$. As such, in order for (56) to hold, the constant $B$ has to decay with $n$, canceling the linear speed-up in (57) and resulting in a HP bound of the form

$$f(\overline{x}^t) - f^\star \leq \frac{\log(1/\delta) + 1}{B'(t + t_0)},$$

where $B' > 0$ is independent of $n$ and the linear speed-up is lost. On the other hand, Lemma 3.6 implies that

$$\mathbb{E}\big[\exp\big(\nu F_{t+1}\big)\big] \leq \exp\left(\frac{(t_0 + 2)^3\nu\Delta_f}{(t + 1 + t_0)^2} + 4\nu G_1 + \frac{4\nu G_2}{t + 1 + t_0} + \frac{4\nu G_3\log(t + t_0 + 1)}{(t + t_0 + 1)^2} + \frac{4\nu G_4}{5(t_0 + 1)^5(t + 1 + t_0)^2}\right),$$

for some problem related constants $G_i > 0$, $i \in [4]$, providing a tighter and more fine-grained control of the MGF compared to (56) and crucially allowing for a bound where the leading term achieves linear speed-up in the number of users (as $G_1$ is

---

[12]Lu (2024) consider the problem of online noncooperative games and show HP convergence guarantees in terms of dynamic regret $R_i(t)$, for *each individual* $i \in [n]$. While this is different from the metrics considered in our work (i.e., *user-averaged* gradient norm/optimality gap), the analysis in (Lu, 2024) suffers from the same shortcomings as other decentralized works, i.e., no result showing variance reduction in the HP sense is provided, making linear speed-up impossible to achieve even if $\frac{1}{n}\sum_{i\in[n]} R_i(t)$ is considered.

independent of $n$). On the other hand, using a different approach to Harvey et al. (2019); Bajović et al. (2023), Liu & Zhou (2024) establish the following HP bound for the last iterate of **SGD** in centralized setting and smooth, strongly convex costs

$$f(x^t) - f^\star = \mathcal{O}\left(\frac{\log(t)\log(1/\delta)}{t + t_0} + \frac{1}{(t + t_0)^2}\right).$$

While such a result could potentially be leveraged to maintain linear speed-up (provided that the constant growing linearly with $n$ affects only the higher-order term), it still accrues a factor of $\log(t)$ in the leading term, whereas the $\log$ term in our bound only affects higher-order terms. As such, Lemma 3.6 provides the most fine-grained control of the MGF and is of independent interest when studying "almost decreasing" type processes, even in centralized settings.

## K. Extending to Time-varying Networks

As discussed in the main body, to model the communication network, we use assumption **(A1)**, which subsumes fixed undirected networks, as well as a class of fixed directed networks. The authors in (Lu et al., 2024) consider a more general case of directed, time-varying networks, facilitated by the following condition.

**(A1′)** At any time $t \geq 1$, the network is given by a time-varying directed graph $G^t = (V, E^t, W^t)$, where $E^t = \{(i, j) :$ user $i$ can send a message to user $j$ in iteration $t\}$ and $W^t \in \mathbb{R}^{n \times n}$ is such that $[W^t]_{ij} \geq l > 0$ if $(i, j) \in E^t$, otherwise $[W^t]_{ij} = 0$, where $l \in (0, 1)$. Moreover, $W^t$ is row-stochastic and balanced for any $t \geq 1$, i.e., $W^t \mathbf{1}_n = \mathbf{1}_n$ and $\sum_{j \in [n]}[W^t]_{ij} = \sum_{j \in [n]}[W^t]_{ji}$, for all $i \in V$. Finally, for some $U \in \mathbb{N}$ and all $t \geq 1$, the graph $G_U^t = (V, E_U^t)$, where $E_U^t = \bigcup_{k=tU}^{(t+1)U-1} E^k$, is strongly connected.[13]

Our work can be readily extended to the setting of time-varying networks, with **(A1)** replaced by **(A1′)**. To see why this is the case, note that in our proofs we only use the fact that under **(A1)**, we have $\lambda = \|W - J\| \in [0, 1)$. Lu et al. (2024) use a similar result for time-varying in networks (see Lemma 1 in (Lu et al., 2024)), namely that under **(A1′)**, for any $t \geq s$

$$\left|[\Phi(t, s)]_{ij} - \frac{1}{n}\right| \leq M\rho^{t-s}, \tag{58}$$

where $\Phi(t, s) = W^{t-1} \times \ldots \times W^s$ is the transition matrix from time $s$ to $t$ (with $\Phi(t, t) = I_n$), while $M > 0$ and $\rho \in (0, 1)$ are graph related constants. As discussed in (Lu et al., 2024), the constant $\rho$ plays the same role of the network connectivity parameter, as $\lambda$ does in our work. It can be readily verified that replacing **(A1)** with **(A1′)** and using (58) to control the consensus gap, our results readily go through, with $\lambda$ being replaced by $\rho$. As such, our results can be easily extended to the more general (directed) time-varying network setting considered in (Lu et al., 2024).

## L. On Sub-Gaussian Vectors and Dimension Dependence

As mentioned in Section 4 in the main body, our rates in Theorems 3.4 and 3.7 exhibit a mild dependence on the problem dimension, of order $\log(d)$, with the dependence stemming from Lemma 3.2. This is a consequence of working with random vectors, where we simultaneously need to show a bound on the MGF of the *inner product* $\langle \bar{z}^t, v \rangle$ for any $\mathcal{F}_t$-measurable vector $v \in \mathbb{R}^d$ (shown in point 2 of Lemma 3.2), as well as on the MGF of the *squared norm* $\|\bar{z}^t\|^2$ (shown in point 3 of Lemma 3.2). While the inner product maintains some desirable properties, such as being zero-mean and linear (in the sense that $\langle \bar{z}^t, v \rangle = \frac{1}{n}\sum_{i \in [n]}\langle z_i^t, v \rangle$), this is not the case with the squared norm, which is neither zero-mean, nor linear. Therefore, trying to directly establish the variance reduction benefit of decentralized learning, i.e., that $\bar{z}^t$ is $\mathcal{O}\left(\frac{\sigma}{\sqrt{n}}\right)$-sub-Gaussian, in the sense of condition **(A4)**, fails to yield the desired result. In particular, recalling that $\sigma^2 = \frac{1}{n}\sum_{i \in [n]}\sigma_i^2$, we then have

$$\mathbb{E}\left[\exp\left(\frac{\|\bar{z}^t\|^2}{\sigma^2}\right) \mid \mathcal{F}_t\right] \overset{(i)}{\leq} \mathbb{E}\left[\exp\left(\frac{\left(\sum_{i \in [n]}\|z_i^t\|\right)^2}{n\sum_{i \in [n]}\sigma_i^2}\right) \mid \mathcal{F}_t\right] \overset{(ii)}{\leq} \mathbb{E}\left[\exp\left(\frac{1}{n}\sum_{i \in [n]}\frac{\|z_i^t\|^2}{\sigma_i^2}\right) \mid \mathcal{F}_t\right]$$

$$\overset{(iii)}{\leq} \prod_{i \in [n]}\left(\mathbb{E}\left[\exp\left(\frac{\|z_i^t\|^2}{\sigma_i^2}\right) \mid \mathcal{F}_t\right]\right)^{1/n} \overset{(iv)}{\leq} \prod_{i \in [n]}\exp\left(\frac{1}{n}\right) = \exp(1), \tag{59}$$

---

[13]Recall that a directed graph $G = (V, E)$ is strongly connected if for any two vertices $i, j \in V$, there exists a path from $i$ to $j$ (and vice versa), see, e.g., (Cvetković et al., 1997; Chung, 1997).

where $(i)$ follows from Proposition C.1, $(ii)$ follows from Sedrakyan's inequality, namely that

$$\frac{(\sum_{i \in [n]} a_i)^2}{\sum_{i \in [n]} b_i} \leq \sum_{i \in [n]} \frac{a_i^2}{b_i},$$

which holds for any $n \in \mathbb{N}$, $a_i \in \mathbb{R}$ and $b_i > 0$, in $(iii)$ we used Proposition C.1 and the fact that noise is conditionally independent across users, while $(iv)$ follows from **(A4)**. Therefore, a direct approach yields that $\overline{z}^t$ is $\sigma$-sub-Gaussian, failing to show the variance reduction benefit. To circumvent this issue, we use a different argument, namely that condition **(A4)** is equivalent to *norm-sub-Gausiannity*, i.e., that for any $\epsilon > 0$, we have

$$\mathbb{P}\big(\|z_i^t\|^2 > \epsilon \mid \mathcal{F}_t\big) \leq 2\exp\left(\frac{\epsilon^2}{2\sigma_i^2}\right),$$

which, combined with Proposition C.9, implies that

$$\mathbb{P}\big(\|\overline{z}^t\|^2 > \epsilon\big) \leq 2d\exp\left(\frac{n\epsilon}{3\sigma^2}\right).$$

This result can be further leveraged to show the desired bound on the MGF of $\overline{z}^t$, while introducing the mild $\log(d)$ dependence. Note that when there is no noise, i.e., $\sigma = 0$, the dimension dependence in our bounds vanishes, highlighting that this dependence stems from the noise. As mentioned in (Jin et al., 2019), while mild, it is not clear if the dependence on $d$ can be completely removed. We now highlight three important facts. First, if a different definition of sub-Gaussian vectors is used, namely the one via inner products, which states that, for any $\mathcal{F}_t$-measurable vector $v$, we have

$$\mathbb{E}\big[\exp\left(\langle v, z_i^t \rangle\right) \mid \mathcal{F}_t\big] \leq \exp\left(\frac{\sigma_i^2 \|v\|^2}{2}\right),$$

this would inevitably lead to a worse dependence on the problem dimension, as in order to bound the MGF of the squared norm of the average noise, i.e., $\|\overline{z}^t\|^2$, one would need to use a similar argument as the one for Lemma 1 in (Jin et al., 2019), via covers of the unit sphere, picking up a factor of $d$ in the final rate, much worse than our $\log(d)$ dependence. Second, while the notion of sub-Gaussian random vectors used in **(A4)** is equivalent to the notion of norm-sub-Gaussian vectors in (Jin et al., 2019) and stronger than the notion of sub-Gaussian vectors defined via inner products above, it is in fact the standard notion of sub-Gaussianity used in almost all HP results, in both centralized and decentralized settings, see (Nemirovskiĭ et al., 2009; Ghadimi & Lan, 2013; Li & Orabona, 2020; Liu et al., 2023a; Liu & Zhou, 2024; Lu et al., 2024; Lu, 2024; Xu et al., 2024) and references therein. Finally, we note that in the extreme regime of $n \ll \log(d)$, we can simply use the argument in (59), to conclude that $\overline{z}^t$ is $\sigma$-sub-Gaussian, completely removing the dependence on dimension $d$, at the cost of losing linear speed-up in the number of users.

## M. General Step-Size for Strongly Convex Costs

In this section we extend the results for strongly convex costs under a more general step-size schedule. To that end, we consider a time-varying step-size schedule $\alpha_t \propto t^{-\eta}$, where $\eta \in (1/2, 1]$. We start by providing some technical result. The first extends Proposition C.6 for a general step-size schedule.

**Lemma M.1.** *For any $c, t_0 > 0$, $\eta \in (1/2, 1]$ and $0 \leq a \leq b$, we have*

$$\prod_{k=a}^{b} \left(1 - \frac{c}{(k+t_0)^\eta}\right) \leq \frac{(a+1+t_0)^{c\eta}}{(b+2+t_0)^{c\eta}}.$$

*Proof.* Using the inequality $1 - x \leq e^{-x}$, it follows that

$$\prod_{k=a}^{b} \left(1 - \frac{c}{(k+t_0)^\eta}\right) \leq \prod_{k=a}^{b} \exp\left(-\frac{c}{(k+t_0)^\eta}\right) = \exp\left(-\sum_{k=a}^{b} \frac{c}{(k+t_0)^\eta}\right). \tag{60}$$

Next, note that for any $k \geq 1$

$$\frac{1}{(k+t_0)^\eta} = \int_k^{k+1} \frac{1}{(k+t_0)^\eta} ds \geq \int_k^{k+1} \frac{1}{(s+t_0)^\eta} ds,$$

therefore, we have

$$\sum_{k=a}^{b} \frac{c}{(k+t_0)^\eta} \geq \sum_{k=a}^{b} \int_{k}^{k+1} \frac{c}{(s+t_0)^\eta} ds.$$

We now differentiate between two cases: first, if $\eta = 1$, then $\int_{k}^{k+1} \frac{c}{(s+t_0)^\eta} ds = c \log\left(\frac{k+t_0}{k+1+t_0}\right)$, therefore

$$\sum_{k=a}^{b} \frac{c}{(k+t_0)^\eta} \geq \sum_{k=a}^{b} c \log\left(\frac{k+t_0}{k+1+t_0}\right) = c \log\left(\frac{a+t_0}{b+t_0+1}\right).$$

Plugging into (60), we get

$$\prod_{k=a}^{b} \left(1 - \frac{c}{(k+t_0)^\eta}\right) \leq \left(\frac{a+t_0}{b+t_0+1}\right)^c. \tag{61}$$

Next, if $\eta \in (1/2, 1)$, we use the fact that $\frac{1}{(s+t_0)^\eta} \geq \frac{\eta}{s+t_0+1}$, implying $\int_{k}^{k+1} \frac{c}{(s+t_0)^\eta} ds \geq c\eta \log\left(\frac{k+t_0+1}{k+t_0+2}\right)$, therefore

$$\sum_{k=a}^{b} \frac{c}{(k+t_0)^\eta} \geq c\eta \sum_{k=a}^{b} \log\left(\frac{k+1+t_0}{k+2+t_0}\right) = c\eta \log\left(\frac{a+1+t_0}{b+2+t_0}\right).$$

Plugging into (60), we get

$$\prod_{k=a}^{b} \left(1 - \frac{c}{(k+t_0)^\eta}\right) \leq \left(\frac{a+1+t_0}{b+2+t_0}\right)^{c\eta}. \tag{62}$$

The claim follows by combining (61)-(62) and noting that $\left(\frac{a+t_0}{b+1+t_0}\right)^c \leq \left(\frac{a+1+t_0}{b+2+t_0}\right)^c$. $\qquad \square$

The next result extends Lemma 3.6 for a general step-size.

**Lemma M.2.** *Let $\{X^t\}_{t\in\mathbb{N}}$ be a sequence of random variables initialized by a deterministic $X^1 > 0$, such that, for some $\eta \in (1/2, 1]$, some $C_1, C_2, C_3 > 0$ and every $t \geq 1$*

$$\mathbb{E}[\exp(X^{t+1})] \leq \mathbb{E}\left[\exp\left(\left(1 - \frac{2}{(t+t_0)^\eta}\right)X^t + \frac{C_1}{(t+t_0)^{2\eta}} + \frac{C_2}{(t+t_0)^{1+\eta}} + \frac{C_3}{t+t_0}\right)\right]. \tag{63}$$

*If $t_0 \geq 2$, we have*

$$\mathbb{E}[\exp(X^{t+1})] \leq \exp\left(\frac{X_1}{(t+1+t_0)^{2\eta}} + \frac{C_1}{(t+t_0)^{2\eta-1}} + \frac{C_2}{(t+t_0)^\eta} + C_3\right).$$

*Proof.* Starting from (63), taking the logarithm, defining $Y_t := \log \mathbb{E}[\exp(X^t)]$ and $b_t = 1 - \frac{2}{t+t_0}$, we then have

$$\begin{aligned} Y^{t+1} &\leq \frac{C_1}{(t+t_0)^{2\eta}} + \frac{C_2}{(t+t_0)^{1+\eta}} + \frac{C_3}{t+t_0} + \log \mathbb{E}[\exp(b_t X^t)] \\ &\leq b_t Y^t + \frac{C_1}{(t+t_0)^{2\eta}} + \frac{C_2}{(t+t_0)^{1+\eta}} + \frac{C_3}{t+t_0}, \end{aligned} \tag{64}$$

where the second inequality follows from the fact that $b_t \in (0, 1)$ and Proposition C.1. Unrolling the recursion (64) and noting that $Y_1 = X_1$, since $X_1 > 0$ is deterministic, we get

$$Y^{t+1} \leq X^1 \prod_{k\in[t]} b_k + \sum_{k\in[t]} \frac{C_1}{(k+t_0)^{2\eta}} \prod_{s=k+1}^{t} b_s + \sum_{k\in[t]} \frac{C_2}{(k+t_0)^{1+\eta}} \prod_{s=k+1}^{t} b_s + \sum_{k\in[t]} \frac{C_3}{(k+t_0)} \prod_{s=k+1}^{t} b_s.$$

Using Lemma M.1, it follows that $\prod_{s=k+1}^{t} b_s \leq \left(\frac{k+t_0}{t+t_0}\right)^{2\eta}$, therefore using Darboux sums, we get

$$Y^{t+1} \leq \frac{X^1}{(t+t_0)^{2\eta}} + \frac{C_1}{(t+t_0)^{2\eta-1}} + \frac{C_2}{(t+t_0)^\eta} + C_3. \tag{65}$$

Taking the exponent on both sides completes the proof. $\qquad \square$

The next two results generalize Lemmas G.1 and 3.5. The proofs follow essentially the same steps as those of Lemmas G.1 and 3.5, with Proposition C.6 replaced by Lemma M.1. For brevity, we omit the details.

**Lemma M.3.** *Let assumptions* **(A1)-(A4)** *and* **(A6)** *hold and let* $a, t_0, K > 0$ *and* $\nu \in (0, 1]$ *be some positive constants. If the step-size satisfies* $\alpha_t \leq \min\left\{\frac{1}{\bar{\sigma}\sqrt{2(t+t_0+2)^{\eta-1}K}}, \frac{1}{\mu}\right\}$ *for all* $t \geq 1$, *with* $\nu \leq \min\left\{1, \frac{\mu}{24a\bar{\sigma}^2 K}\right\}$ *and* $K_{t+1} = (t + t_0 + 2)^{2\eta-1}K$, *then*

$$\mathbb{E}[\exp(\nu K_{t+1}\|\mathbf{x}^{t+1} - \mathbf{x}^\star\|^2)] \leq \exp\left(\nu K_{t+1}\left(\frac{nC_1}{(t+t_0+2)^{2\eta-1}} + (t+t_0+2)^{1-\eta}C_2 + \frac{C_3}{(t+t_0+2)^{a\mu\eta}}\right)\right),$$

*for some problem dependent constants* $C_i > 0$, $i \in [3]$, *all independent of* $n$.

**Lemma M.4.** *Let* **(A1)-(A4)** *and* **(A6)** *hold, let* $a, t_0, K > 0$ *and the step-size be given by* $\alpha_t = \frac{a}{(t+t_0)^\eta}$ *for* $\eta \in (1/2, 1]$, *and let* $x_i^1 = x_j^1$, *for all* $i, j \in [n]$. *If* $a = \frac{6}{\mu}$ *and* $t_0 \geq \max\left\{6^{\frac{1}{\eta}}, \frac{1}{1-\lambda}, \frac{7776\bar{\sigma}^2\lambda^2 K}{\mu^2(1-\lambda)}, \left(\frac{12\lambda L\sqrt{10}}{\mu(1-\lambda)}\right)^{\frac{1}{\eta}}\right\}$, *then for* $K_{t+1} = (t + t_0 + 2)^{2\eta-1}K$ *and any* $\nu \leq \min\left\{1, \frac{\mu^2}{144\sigma^2 K}\right\}$, *we have*

$$\mathbb{E}\left[\exp\left(\nu K_{t+1}\sum_{i\in[n]}\|x_i^{t+1} - \overline{x}^{t+1}\|^2\right)\right] \leq \exp\left(\nu K_{t+1}\left(\sum_{k=1}^t \lambda^{t-k}S_k + \sum_{k=1}^t \lambda^{t-k}D_k\right)\right),$$

*where* $D_t = \alpha_t^2\left(\frac{nC_1'}{(t+t_0+2)^{2\eta-1}} + (t+t_0+2)^{1-\eta}C_2' + \frac{C_3'}{(t+t_0+2)^{a\mu\eta}}\right)$ *and* $S_t = \alpha_t^2 nC_4'$, *for some problem related constants* $C_i' > 0$, $i \in [4]$, *all independent of* $n$.

We are now ready to state the main result.

**Theorem M.5.** *Let* **(A1)-(A4)** *and* **(A6)** *hold, the step-size be given by* $\alpha_t = \frac{a}{(t+t_0)^\eta}$ *for* $\eta \in (1/2, 1]$ *and let* $x_i^1 = x_j^1$, *for all* $i, j \in [n]$. *If* $a = \frac{6}{\mu}$, $t_0 \geq \max\left\{6^{\frac{1}{\eta}}, \frac{1+\lambda}{1-\lambda}, \frac{23328\bar{\sigma}^2\kappa^2\lambda^2}{\mu(1-\lambda)}, \frac{9720\sigma^2\kappa}{\mu}, \left(\frac{12\kappa\lambda\sqrt{10}}{1-\lambda}\right)^{\frac{1}{\eta}}\right\}$ *and* $\nu = \min\left\{1, \frac{\mu}{432\sigma^2\kappa^2}, \frac{\mu}{72\kappa}\right\}$, *then for any* $\delta \in (0, 1)$ *and* $T \geq 1$, *with probability at least* $1 - \delta$, *it holds that*

$$\frac{1}{n}\sum_{i\in[n]}\left(f(x_i^t) - f^\star\right) = \mathcal{O}\left(\frac{\log(2d/\delta)}{n(t+t_0)^{2\eta-1}} + \frac{1}{(t+t_0)^{3\eta-1}}\right).$$

Theorem M.5 extends the results of Theorem 3.7 for the polynomially decaying step-size $\alpha_t \propto t^{-\eta}$ for $\eta \in (1/2, 1]$, recovering the results of Theorem 3.7 when $\eta = 1$. Interestingly, it offers the following insight: *for strongly convex costs, the step-size does not determine the rate in the number of users* $n$, *but only the rate in time* $t$. In other words, **DSGD** achieves the optimal linear speed-up for strongly convex costs independent of the step-size schedule. This is intuitive, as the step-size in Theorem M.5 does not depend on $n$, whereas for non-convex costs the step-size needs to be chosen as $\alpha \propto (nT)^{-1/2}$ to guarantee linear speed-up. We now provide a proof sketch, highlighting the main differences to the proof of Theorem 3.7.

*Proof sketch of Theorem 3.7.* The main difference stems from changing the scaling when defining the quantity $F^t$, which we now define as $F^t := n(t+t_0)^{2\eta-1}\left(f(\overline{x}^t) - f^\star\right)$. Denoting by $A_t := \alpha_t(t+t_0+1)^{2\eta-1}$, using Lemma 3.1 and the properties of strongly convex functions, we get

$$F^{t+1} \leq (1 - \alpha_t\mu)\left(\frac{t+t_0+1}{t+t_0}\right)^{2\eta-1}F^t - A_t\langle\nabla f(\overline{x}^t), \overline{z}^t\rangle + \alpha_t A_t nL\|\overline{z}^t\|^2 + \frac{A_t L^2}{2}\sum_{i\in[n]}\|x_i^t - \overline{x}^t\|^2. \quad (66)$$

Starting from (66), we follow similar steps as in the original proof, with Lemmas 3.5 and 3.6 replaced by Lemmas M.4 and M.2. To bound the consensus gap, we use Lemma M.4 and additionally proceed as follows

$$\mathbb{E}\left[\exp\left(\nu q c_t\|\mathbf{x}^t - \overline{\mathbf{x}}^t\|^2\right)\right] \leq \mathbb{E}\left[\exp\left(\nu K_t\|\mathbf{x}^t - \overline{\mathbf{x}}^t\|^2\right)\right] \leq \exp\left(\nu K_t\sum_{k=1}^{t-1}\lambda^{t-1-k}\left(D_k + S_k\right)\right),$$

where $D_k = \alpha_k^2 \left( \frac{nC_1'}{(k+t_0+2)^{2\eta-1}} + C_2'(k+t_0+2)^{1-\eta} + \frac{C_3'}{(k+t_0+2)^{a\mu\eta}} \right)$ and $S_k = \alpha_k^2 nC_4'$. To further bound the above expression, we use Lemma D.3, to get

$$nC_1' \sum_{k=1}^{t-1} \frac{\lambda^{t-1-k}\alpha_k^2}{(k+t_0)^{2\eta-1}} = \sum_{k=1}^{t-1} \lambda^{t-1-k} \frac{nC_1'}{(k+t_0)^{4\eta-1}} \leq \frac{nC_1'}{(t+t_0)^{4\eta-1}}$$

$$C_2' \sum_{k=1}^{t-1} \lambda^{t-1-k}\alpha_k^2(k+t_0)^{1-\eta} = \sum_{k=1}^{t-1} \lambda^{t-1-k} \frac{C_2'}{(k+t_0)^{3\eta-1}} \leq \frac{C_2'}{(t+t_0)^{3\eta-1}}$$

$$C_3' \sum_{k=1}^{t-1} \lambda^{t-1-k} \frac{\alpha_k^2}{(k+t_0)^{a\mu\eta}} = \sum_{k=1}^{t-1} \lambda^{t-1-k} \frac{C_3'}{(k+t_0)^{(a\mu+2)\eta}} \leq \frac{C_3'}{(t+t_0)^{\eta(2+a\mu)}}$$

$$nC_4' \sum_{k=1}^{t-1} \lambda^{t-1-k}\alpha_k^2 = \sum_{k=1}^{t-1} \lambda^{t-1-k} \frac{nC_4'}{(k+t_0)^{2\eta}} \leq \frac{nC_4'}{(t+t_0)^{2\eta}}.$$

Noting that $2\eta - 1 \geq 0$ and that $a\mu \geq 1$, it then follows that

$$\mathbb{E}\left[ \exp\left( \nu q c_t \|\mathbf{x}^t - \overline{\mathbf{x}}^t\|^2 \right) \right] \leq \exp\left( \nu K_t \left( \frac{G_1}{(t+t_0)^{3\eta-1}} + \frac{nG_2}{(t+t_0)^{2\eta}} \right) \right),$$

where $G_1 = C_2' + C_3'$ and $G_2 = C_1' + C_4'$. Since $\frac{1}{q} = \frac{\alpha_t\mu}{4+\alpha_t\mu} \leq \frac{\alpha_t\mu}{4}$, we finally get

$$\sqrt[q]{\mathbb{E}\left[ \exp\left( \nu q c_t \|\mathbf{x}^t - \overline{\mathbf{x}}^t\|^2 \right) \right]} \leq \exp\left( \frac{\nu\mu\alpha_t K_t}{4} \left( \frac{G_1}{(t+t_0)^{3\eta-1}} + \frac{nG_2}{(t+t_0)^{2\eta}} \right) \right) \leq \exp\left( \frac{\nu G_1}{(t+t_0)^{2\eta}} + \frac{\nu nG_2}{(t+t_0)^{1+\eta}} \right),$$

where the last inequality follows from the definition of $K_t$ and the step-size. The rest of the proof follows the same steps as that of Theorem 3.7 and is omitted, for brevity.

