# OpenReview forum: "High-Probability Convergence Guarantees of Decentralized SGD"
_ICML.cc/2026/Conference — ICML 2026 regular_

### Official Review · Reviewer_z5vY · 2026-02-26

**Soundness:** 4
**Presentation:** 1
**Significance:** 2
**Originality:** 2
**Overall Recommendation:** 5
**Confidence:** 4

**Summary:**

This paper establishes sharp high probability convergence results for decentralized SGD iterates. In particular, the paper claims to rigorously establish the linear speed-up, enjoyed by decentralized SGD, in its high-probability bound, and highlights the use of a certain ``offset-trick'' to establish this result.

**Compliance With Llm Reviewing Policy:**

Affirmed.

**Final Justification:**

Rebuttal addressed all my concerns

**Key Questions For Authors:**

## Questions

I list my minor comments in the form of questions.

- In Assumption (A4), what does "noise" mean? Does it imply gradient noise $z^t$ or the random samples $\xi^t$?
- Can Assumption (A3) be replaced by a stochastic Lipschitz condition? (Assumption 2 in [2]).

**Limitations:**

Yes.

**Strengths And Weaknesses:**

## Strengths

**Soundness**: The paper is theoretically solid, and all claims are substantiated by corresponding proofs.

**Presentation**: The paper is mostly clearly written and legible. There still remains some issues with presentations that I subsequently mention in the weakness and question sections. In general, the paper currently feels like an accumulation of theoretical results and discussion of proof techniques. There are also some typos, eg. P7, right column, l. 5: "Morevoer" → Moreover.

**Significance**: The paper is primarily theoretical in nature, and as such its theoretical contribution can be considered new.

**Originality**: The proof techniques are standard.

## Weaknesses

I list my major concerns in the following. I would gladly my score if satisfactory rebuttals are provided.

- The assumptions are not properly articulated. For example, Assumption (A4).3 is stronger than usual sub-Gaussian conditions, and is, in fact, essentially equivalent to the norm-sub Gaussian condition in [1]. A simple calculation based on this definition makes the $\sqrt{d}$ factor redundant. I think the authors themselves mention in Page 36 that "Using a similar argument to the one in Lemma 6 and Corollary 7 in (Jin et al., 2019), we could then show that $z_t$ is $\sigma/\sqrt{n}$-sub-Gaussian, while reducing the dependence on problem dimension to $\log(d)$." I think this should be properly clarified; either one derives a rate without the $\sqrt{d}$ term under norm-sub Gaussianity, or maintain the correct rate but derive it from the usual, linear-projection based sub-Gaussianity definition.

- In the set-up of Theorem 3.4, one looks at an average of $f(x_i^t)$ over both the $n$ clients and the $t$ iterates, and therefore, it is fair to naively expect a rate of $O_{\mathbb{P}}(1/\sqrt{nT})$ anyway. Moreover, in such a set-up, it is well-known that the SGD with constant learning rates does not converge to the global minima, and instead converges to a stationary distribution [2,3]. Therefore, the comparison with [4], and saying that they do not attend linear speed-up and we do, is slightly unfair. On the other hand, the quantity $R_i(t)$ controlled by [5] with high probability, is slightly different compared to the left-hand sides in Theorems 3.4 and 3.7 in this paper; therefore the comparisons aren't exactly apples-to-apples, and this should be properly clarified.

- It is also well-known that to ensure convergence with fixed learning rate, one can make the learning rate $\alpha$ depend on the number of iterates. However, such choices eg $\alpha \asymp 1/T$ essentially makes an online algorithm offline by pre-specifying the number of samples required.

- On the other hand, I would expect a rate of $O_{\mathbb{P}}(1/\sqrt{nT})$ anyway with Polyak-Ruppert averaged iterates with the polynomially decaying learning rates $\alpha_t \propto t^{-\alpha}$. However, no such result is explicitly provided. Theorem 3.7 is a welcome step in that direction, but it only looks at end-terms, and not at PR-averaged version. Additionally, Theorem 3.7 only deals with the choice of $\alpha_t \propto 1/t$, which is prohibitively slow from a computational perspective. However, it would be good to show (i) at least a result like Theorem 3.7 for a general, polynomially decaying learning rate, or ideally (ii) a result for PR-averaged versions for a general, polynomially decaying learning rate.

- I understand the inevitable denseness of a theoretical paper. But it would greatly facilitate readability if adequate explanation behind the assumptions and the conditions in the theorems, is provided. For example, for an uninitiated reader, the particular choice of $\alpha$ in Theorem 3.4 is quite opaque, and merits a discussion. Moreover, tin the discussion following Theorem 3.4, it is not clear if the dependence on dimension $d$ is optimal. Additionally, it is unclear what the authors mean by "We can see from the bound in Theorem 3.4 that network connectivity only affects higher-order terms, decaying at rate $O(1/T)$". A similar concern involves the discussion following Theorem 3.7. For example, the effect of the term $\nu$, which combines the strong-convexity and the condition number of the optimization problem, is not at all discussed. In general, a lack of nuance is reflected from the discussion and remarks, which barely goes beyond superficially expressing the rates in english prose. I would appreciate if the authors provide more in-depth insights into the rates, especially highlighting how the high-probability perspective helps us beyond MSE bounds.

**References:**

[1] *A Short Note on Concentration Inequalities for Random Vectors with SubGaussian Norm*, Jin, Chi, et al. 2019.

[2] *Stochastic Gradient Descent: A Nonlinear Time Series Perspective*. Li, Jiaqi et al. 2024.

[3] *Tight Analysis of Decentralized SGD: A Markov Chain Perspective*, Versini, Lucas, Paul Mangold, and Aymeric Dieuleveut. arXiv preprint arXiv:2601.07021 (2026).

[4] *A unified and refined convergence analysis for non-convex decentralized learning*, Alghunaim, Sulaiman A., and Kun Yuan. IEEE Transactions on Signal Processing 70 (2022).

[5] *Online distributed algorithms for online noncooperative games with stochastic cost functions: high probability bound of regrets.* Lu, Kaihong. IEEE Transactions on Automatic Control 69.12 (2024): 8860-8867.

---

> ### Author Rebuttal · Authors · 2026-03-29
>
> ## **On dimension dependence**
>
> We thank the reviewer for their astute observation, which helped to significantly relax the problem dimension dependence. As the reviewer correctly noted, the notion of sub-Gaussainity used in our work is equivalent to norm-sub-Gaussianity, which, after a careful revision, allowed us to relax the dependence on problem dimension in our bounds to $\log(d)$, for both non-convex and strongly convex costs. Due to space limits, the reviewer is kindly referred to our response to reviewer oxLX for details, while noting briefly that the notion of sub-Gaussianity in (A4) is standard in both centralized and decentralized HP works.
>
> ## **On Polyak-Ruppert and general step**
>
> A result for Polyak-Ruppert average  (PRA) under a slower decaying step follows directly from Thm 3.4 and (A6). Recalling that Thm 3.4 with step $\alpha \propto \frac{1}{\sqrt{nT}}$ gives $\frac{1}{nT}\sum_{i,t}||\nabla f(x_i^t)||^2 = \mathcal{O}\Big(\frac{1}{\sqrt{nT}}\Big),$ defining the PRA as $\tilde{x}^T = \frac{1}{nT}\sum_{i,t}x_i^t$ and using (A6), we get
> $$2\mu(f(\tilde{x}^T) - f^\star) \leq \frac{1}{nT}\sum_{i,t}||\nabla f(x_i^t)||^2 = \mathcal{O}\Big(\frac{1}{\sqrt{nT}}\Big),$$ giving the desired result. Further, Thm 3.7 can be readily extended to a general step $\alpha_{t} \propto t^{-\eta}$, for $\eta \in [1/2,1)$, resulting in a slower, sub-optimal rate. However, Thm 3.7 provides *stronger guarantees*, in multiple facets: (i) the step-size and rate in Thm 3.7 are *optimal* for strongly convex costs, e.g., [1]; (ii) the last iterate is practically more viable than PRA and does not require users maintaining an extra variable (the running average); (iii) the analysis for the last iterate is significantly more involved, as demonstrated in the paper, whereas the result for PRA readily follows from Thm 3.4. We will provide a discussion on extending our results for PRA and general step-size in the final version.
>
> ## **On our results and claims**
>
> There are a few important things to note regarding our results and claims. First, while we agree with the reviewer that it is intuitive to expect linear speed-up (which in part motivates our work), it is *highly non-trivial* to attain, as highlighted by our results, as well as prior works, which were unable to show linear speed-up. Next, Thm 3.4 is not intended to indicate convergence to a global minimum; instead, we measure performance via the standard gradient norm-squared metric, implying convergence to a *stationary point*. Finally, as discussed in Appendix E and I, a fixed step-size is *necessary* to achieve linear speed-up for non-convex costs and the same is true for MSE results [2], i.e., it is not a byproduct of our analysis. For strongly convex costs we use a time-varying step-size, facilitating an online implementation. In our comparison with existing works, we never claimed that [3] does not achieve linear speed-up; in fact, we reference [3] for transient times, which rely on having linear speed-up. In relation to [4], the quantity $R_i(t)$ is indeed different from the quantities in our results. However, the analysis in [4] suffers from the same shortcomings as the other works mentioned in Appendix I, i.e., no result showing variance reduction is provided, making linear speed-up impossible to achieve even if an average of $R_i$'s is considered. We will clarify all of this in the final version.
>
> ## **On discussion**
>
> We thank the reviewer for bringing this to our attention; we will incorporate all of the reviewer's suggestions and provide a more illuminating discussion in the final version. Due to a lack of space, we skip a detailed discussion here.
>
> ## **Key questions**
>
> In (A4) noise refers to $z^t$. Regarding smoothness, while similar, (A3) and Assumption 2 in [5] are different, as in general, neither is implied by the other. However, under certain regularity conditions, Assumption 2 in [5] is known to imply (A3), making it slightly stronger in that sense. We will clarify both points in the final version.
>
> ## **Final comments**
>
> We hope that our responses clarified the reviewer's concerns. In light of our responses, we kindly ask the reviewer to consider increasing the score. We would be happy to answer any further questions during discussion week.
>
> ### **References**
>
> [1] Liu, Z., et al. (2024). *Revisiting the Last-Iterate Convergence of Stochastic Gradient Methods*. 12th ICLR.
>
> [2] Koloskova, A., et al. (2020). *A Unified Theory of Decentralized SGD with Changing Topology and Local Updates*. 37th ICML.
>
> [3] Alghunaim, S. A., et al. (2022). *A Unified and Refined Convergence Analysis for Non-Convex Decentralized Learning*. IEEE Transactions on Signal Processing.
>
> [4] Lu, K. (2024). *Online Distributed Algorithms for Online Noncooperative Games With Stochastic Cost Functions: High Probability Bound of Regrets*. IEEE Transactions on Automatic Control.
>
> [5] Li, J., et al. (2024). *Stochastic Gradient Descent: A Nonlinear Time Series Perspective*. Preprint.

---

> > ### Author Rebuttal · Reviewer_z5vY · 2026-04-03
> >
> > I thank the authors for their detailed response, resolving most of my concerns. Regarding the rebuttal on PRA and step-size, I agree with the authors that the last iterate rate for $\alpha_t \propto t^{-\alpha}$, $\alpha \in (1/2,1)$ is indeed suboptimal; however, this is where PRA comes in. With PRA, even with this step-size, one should get a $1/ \sqrt{nT}$ rate. As you can understand, this rate cannot be readily obtained from your theorem 3.7 as you could do with $\alpha_t \propto t^{-1}$; the rate you would achieve from Theorem 3.7 would not be the optimal rate for PRA. Therefore it is important to do a separate analysis for PRA.

---

> > > ### Author Response · Authors · 2026-04-04
> > >
> > > We are glad that our response addressed most of the reviewer's concerns and are thankful to the reviewer for providing us with an opportunity to further clarify and improve some of our statements and results.
> > >
> > > ## **On slower decaying step-size**
> > > To further address the reviewer's request for a result under a polynomially decaying, time-varying step-size of the form $\alpha_t \propto t^{-\eta}$, for $\eta \in (1/2,1]$, we rigorously revisited the proof of Thm 3.7 (and the supporting lemmas), establishing the following HP guarantee of the last iterate, for any $t \geq 1$
> > > $$
> > > \frac{1}{n}\sum_{i \in [n]}(f(x_i^t) - f^*) = \mathcal{O}\bigg(\frac{1}{nt^{2\eta-1}} + \frac{1}{t^{3\eta-1}} \bigg).
> > > $$
> > > This result strictly generalizes Thm 3.7 to include a polynomially decaying step-size schedule and recovers results of Thm 3.7 for $\eta = 1$. Setting $\eta = 3/4$, one gets the desired rate $\mathcal{O}\Big(\frac{1}{n\sqrt{t}} + \frac{1}{t^{5/4}}\Big)$ , with an optimal dependence on $n$, strictly better than in $\mathcal{O}(1/\sqrt{nt})$ mentioned by the reviewer, while matching it in $t$. Moreover, this result provides an important insight: *for strongly convex costs, the step-size schedule does not determine the rate in number of users $n$, but only affects the rate in time $t$*! This is intuitive, as the step-size in Thm 3.7 (and in the new generalized results) does not depend on $n$, whereas for non-convex costs in Thm 3.4 the step-size needs to be chosen as $\alpha \propto 1/\sqrt{nT}$ to guarantee linear speed-up. As such, our HP results provide further important insight into the behavior of DSGD for strongly convex costs and general time-varying step-sizes. While we are unable to provide a link to the full proof due to conference policy, at a high level we replace the original $(t+t_0)$ scaling of the quantity $F_t$, defined in line 1477 on page 27, with $(t+t_0)^{2\eta-1}$ and repeat similar steps as in the original proof. We  would be happy to discuss further and will include the result in the final version.
> > >
> > > ## **On the sub-optimality statement**
> > > What we meant when stating that the last iterate with a general step-size would provide a sub-optimal rate is the fact that *any rate other than* $\mathcal{O}(1/nt)$ is *sub-optimal for strongly convex costs*. As shown in the point above, the rate for the last iterate under general step-size $\alpha_t \propto t^{-\eta}$, $\eta \in (1/2,1)$ has a better dependence on $n$ than the rate suggested by the reviewer to be achievable for PRA, however, both are strictly sub-optimal compared to the rate in Thm 3.7. While the reviewer is correct that a slower decaying step-size is important in some cases, e.g., for non-convex costs, where a too aggressive step-size decay can result in getting stuck at a local maxima or a saddle point, that is not the case in Thm 3.7. In that result, we assume strong convexity, which ensures *a single stationary point exists and corresponds to the global minima*, hence there is no danger of getting stuck at a bad point and it is of interest to converge as fast as possible. The step-size $\alpha_t \propto t^{-1}$ used in Thm 3.7 is known to provide the exact optimal convergence rate in time $t$, with our extended analysis further demonstrating that the optimal dependence on $n$ always holds for the last iterate in the strongly convex case. We will clarify this in the final version.
> > >
> > > ## **On Polyak-Ruppert**
> > > While a result for PRA and a fixed step-size $\alpha \propto 1/\sqrt{nT}$ follows from Thm 3.4 (as shown in our previous response), the reviewer is correct that the time-varying step-size case for PRA does not directly follow from our existing results. Deriving such a result would require a completely different approach, which we believe to be beyond the timeline of the discussion period. Moreover, as mentioned in our first point, the last iterate under a polynomially decaying step-size achieves a rate with an *optimal dependence on $n$*, better than the one intuited by the reviewer for PRA. While it would be interesting to derive a result for PRA under a general time-varying step, we believe it is beyond the scope of current work and hope that our result for PRA under a fixed step-size and the generalized result for the last iterate are sufficient to address the reviewer's concerns for polynomial steps. We will discuss possible extensions for PRA in the final version.
> > >
> > > ## **Final comments**
> > > We want to again thank the reviewer for their insightful comments, which helped to significantly improve the results of this paper, by: (i) greatly relaxing the dependence on $d$; (ii) generalizing the step-size schedule for strongly convex costs and providing novel insights on the effect of step-size and linear speed-up; (iii) highlighting ambiguities and helping to improve the discussion of our results. We hope that our responses address the last of the reviewer's concerns. If that is the case, we kindly ask the reviewer to consider increasing the score.

---

### Official Review · Reviewer_oxLX · 2026-03-10

**Soundness:** 4
**Presentation:** 4
**Significance:** 3
**Originality:** 3
**Overall Recommendation:** 5
**Confidence:** 4

**Summary:**

This paper studies high-probability convergence guarantees for decentralized SGD. Compared with prior work, it removes overly restrictive assumptions such as uniformly bounded gradients and establishes high-probability guarantees under the same conditions that are typically used for MSE results. The paper also proves that decentralized SGD achieves a linear speed-up in the high-probability sense.

**Compliance With Llm Reviewing Policy:**

Affirmed.

**Final Justification:**

My concerns have been adequately addressed. I will keep my positive score.

**Key Questions For Authors:**

1. Is the dimension dependence essential, or mainly due to the proof technique? In what practical settings is it reasonable to assume that $z_i^t$ is $\sigma_i$-norm-sub-Gaussian?

2. Assumption (A5) is necessary for constant stepsize decentralized SGD, and it can be removed using gradient tracking. What are the main technical difficulties in proving high-probability convergence for decentralized gradient tracking methods? While closing the question might actually be out of scope of this paper, some insights on this question would be appreciated.

**Limitations:**

yes

**Strengths And Weaknesses:**

**Strengths**

1. The paper is well written and the overall logic is clear.

2. The paper establishes high-probability guarantees under the same assumptions as MSE results, which is an important contribution.

3. Linear speed-up for decentralized SGD in the high-probability sense is proved.

**Weaknesses**

The bounds depend on the problem dimension $d$.

---

> ### Author Rebuttal · Authors · 2026-03-29
>
> We are glad that the reviewer appreciated the merits of our work and found the paper well-written and clear. We answer below the two questions raised by the reviewer and would be happy to answer any further questions during discussion week.
>
> ### **On dimension dependence**
>
> In light of the question on dimension dependence raised by this reviewer and reviewer z5vY, we revisited our results and managed to substantially improve the dependence on the problem dimension, reducing it from $\sqrt{d}$ and $d$ for non-convex and strongly convex costs, to $\log(d)$ in both cases. This was achieved by noting that the sub-Gaussian condition in (A4) is equivalent to norm-sub-Gaussianity defined in [1], and leveraging this (e.g., using Corollary 7 in [1]) to show an improved bound on the moment-generating function (MGF) of the average noise $\overline{z}^t$, from $\mathbb{E}\Big[\exp\Big(\frac{n||\overline{z}^t||^2}{120d\sigma^2} \Big) \Big] \leq \exp(1)$ to $\mathbb{E}\Big[\exp\Big(\frac{n||\overline{z}^t||^2}{15\sigma^2} \Big) \Big] \leq 2d\exp(1) = \exp(\log(2d) + 1)$. We now highlight two important facts: first, the sub-Gaussian condition used in (A4) is the *standard notion* of sub-Gaussianity used in both centralized and decentralized settings, see, e.g., [2]-[5] and references therein. In that sense, while the notion of sub-Gaussianity used in our work is equivalent to norm-sub-Gaussianity, it is *on par* with the notion used in other works. Second, while the $\log(d)$ factor introduces a mild dependence on the dimension, it is *not clear* whether the dependence on $d$ can be completely removed, which is also noted by the authors in [1]. The worse dimension dependence in our original submission stems from a more conservative analysis, which will be corrected in the final version. Moreover, we will provide a discussion on how the notion of sub-Gaussianity used in our and existing works relates to both norm-sub-Gaussianity, as well as the classical definition of vector sub-Gaussianity via inner products.
>
> ### **Incorporating gradient tracking**
>
> Indeed, as the reviewer correctly notes, the condition (A5) can be removed when gradient tracking is incorporated. However, gradient tracking introduces additional challenges in the analysis, which we outline next. First, unlike DSGD, where each user maintains only the local model estimate, GT requires each user to maintain another variable, which tracks the global gradient. This introduces additional algorithm dynamics and the need for different recursions and inequalities, which do not readily follow from the existing relations developed for DSGD. Second, it is well-known from the MSE analysis that bounds accounting for the joint effect of the local model deviations and the local gradient tracker deviations are required, e.g., provided in the form of a linear time-invariant system in [6]. While this approach is feasible in the MSE sense due to the linearity of the expectation operator, the MGF used in HP analysis is highly non-linear, making joint bounds difficult to obtain. Even if for some $\alpha,\delta \in (0,1)$ and $\beta, \gamma > 0$, we have bounds of the form
> $$\mathbb{E}[\exp(X_{t+1})] \leq \mathbb{E}[\exp(\alpha X_t + \beta Y_t)],$$ and $$\mathbb{E}[\exp(Y_{t+1})] \leq \mathbb{E}[\exp(\gamma X_t + \delta Y_t)],$$ it is not possible to obtain a linear system with respect to MGFs of $X_t, Y_t$, unless $\beta, \gamma < 1$, which can not be guaranteed even in the MSE setting, where $\alpha,\delta \in (0,1)$ suffices. This is further exacerbated in the strongly convex/PL case, where in addition to the model and tracker consensus gap, one needs to account for the joint effect of the cost optimality gap $f(\overline{x}^t) - f^\star$ as well. Therefore, naively following the strategy from the MSE analysis does not directly translate in the HP setup and a different approach is required. While this is beyond the scope of the current work, it provides an interesting future challenge.
>
> ### **References**
>
> [1] Jin, C., et al. (2019). *A Short Note on Concentration Inequalities for Random Vectors with SubGaussian Norm*. ArXiv preprint.
>
> [2] Li, X. & Orabona, F. (2020). *A High Probability Analysis of Adaptive SGD with Momentum*. Workshop “Beyond first-order methods in ML systems”, 37th ICML.
>
> [3] Liu, Z., et al. (2023). *High Probability Convergence of Stochastic Gradient Methods*. 40th ICML, PMLR, 202, pp. 21884–21914.
>
> [4] Liu, Z. & Zhou, Z. (2024). *Revisiting the Last-Iterate Convergence of Stochastic Gradient Methods*. 12th ICLR.
>
> [5] Lu, K., et al. (2024). *Convergence in High Probability of Distributed Stochastic Gradient Descent Algorithms*. IEEE Transactions on Automatic Control, 69, pp. 2189–2204.
>
> [6] Xin, R., et al. (2021) *An Improved Convergence Analysis for Decentralized Online Stochastic Non-Convex Optimization.* IEEE Transactions on Signal Processing, 69, pp. 1842-1858.

---

> > ### Author Rebuttal · Reviewer_oxLX · 2026-04-01
> >
> > Thank you for your detailed response addressing my concerns. The proposed revision to improve the dimension dependence is helpful and would further strengthen the paper. I will keep my positive score.

---

> > > ### Author Response · Authors · 2026-04-05
> > >
> > > We are glad that all of the reviewer's concerns are successfully addressed and that the reviewer appreciated the improvements in our results achieved during the rebuttal. We thank the reviewer once again for their positive evaluation of our work.

---

### Official Review · Reviewer_WSFW · 2026-03-10

**Soundness:** 3
**Presentation:** 3
**Significance:** 2
**Originality:** 3
**Overall Recommendation:** 3
**Confidence:** 4

**Summary:**

The paper conducts high-probability (HP) analysis of decentralized stochastic gradient descent (DSGD) rather than the more conventional mean-squared error (MSE) analysis commonly done in the literature. The paper improves existing HP analysis since it does not need the bounded gradients assumption recently used in HP analysis of distributed optimization algorithms. Furthermore, it is able to work with light-tailed noise for SGD approximation error rather than the uniformly bounded noise often needed for MSE analysis.

**Compliance With Llm Reviewing Policy:**

Affirmed.

**Key Questions For Authors:**

Please see weaknesses.

**Limitations:**

Yes

**Strengths And Weaknesses:**

## Strengths
1. Thorough literature review in Sec. 1.1.

2. Theorems 3.4 and 3.7 are followed by extensive discussion on its novelty and implication of each term in the bounds.

## Weaknesses
1. The MSE error could be related to the HP analysis through Chebyshev's inequality, but the authors do not discuss this in Sec. 1.

2. Also, it is not clear why HP has garnered attention recently, i.e., what was prior MSE analysis missing that HP is addressing.

3. It is stated in the abstract that numerical experiments are provided to validate the theory, but they are only given later in the Appendix.

4. Also, the current experimental results in Appendix G are only on synthetic data. It would be better to run simulations using real-world datasets.

5. Could the authors provide some insights and ideally provide numerical experiments for setups where only HP analysis is applicable, or is superior to, the MSE analysis. For instance, for some particular choice of ML model/ML dataset, MSE analysis would not satisfy convergence guarantees but HP analysis would.

6. The paper has some formatting issues, e.g., the margins in page 5.

---

> ### Author Rebuttal · Authors · 2026-03-30
>
> We are glad that the reviewer appreciated our thorough literature review and extensive discussions on the implications of our results. We next respond to the main issues raised by the reviewer.
>
> ## **On MSE and HP analysis**
>
> Although both HP and MSE results establish convergence, the nature of these guarantees is very different. MSE results quantify the *average behaviour across many runs* of an algorithm, while HP results quantify the behaviour of *an individual run*. As mentioned briefly in the introduction, this distinction is very important in huge-scale applications like LLM training, where it is often impossible to perform more than a single training run, both resource- and time-wise. For example, the authors in [1] estimate that the GPT-3 model with 175 billion parameters was trained for 34 days, while the variant with 1 trillion parameters can be trained for approximately 84 days. This is further exacerbated by phenomena like heavy-tailed noise, frequently observed during training of deep learning models and transformers, e.g., [2]-[3], which can cause the performance of an individual run to significantly deviate from the average performance. Hence, strong guarantees with respect to an individual run are very important in such applications. As the reviewer correctly notes, MSE results can be used to provide guarantees with respect to an individual run, however, guarantees obtained in this manner can be quite loose compared to HP bounds. In particular, if we have a MSE bound of the form $\mathbb{E}[X_t^2] = 1/t$, Chebyshev's inequality implies $\mathbb{P}(X_t^2 > 1/(\delta t)) \leq \delta$, which shows an inversely proportional dependence on the confidence level $\delta \in (0,1)$. On the other hand, HP results establish guarantees of the type $\mathbb{P}(X_t^2 > \log(1/\delta)/t) \leq \delta$, having a much milder, logarithmic dependence on $1/\delta$. Equivalently, for any threshold $\epsilon > 0$, MSE + Chebyshev implies $\mathbb{P}(X_t^2 > \epsilon) \leq 1/(t\epsilon)$, while HP gives $\mathbb{P}(X_t^2 > \epsilon) \leq \exp(-t\epsilon)$, providing a much sharper bound on the tail probability. Since our goal is to show that $X_t^2 \leq \epsilon$ (e.g., if $X_t = ||x^t-x^*||$ is the optimality gap or if $X_t = \min_{k \in [t]}||\nabla f(x^k)||$ is the stationarity gap), then $\mathbb{P}(X_t^2 > \epsilon)$ represents the *probability of failure* (PoF), with MSE + Chebyshev implying that PoF decays polynomially in $t$ (i.e., as $1/t$), while HP implies that PoF decays exponentially fast in $t$, giving much stronger guarantees on the convergence of an individual run. We will include a detailed discussion on the comparison and differences between MSE and HP guarantees in the final version, further clarifying the importance of HP results.
>
> ## **Numerical experiments**
>
> In the final version, which allows for an extra page, we will move the experiments from Appendix G to the main body. Moreover, per the reviewer's request, we provide additional experiments on real data from the LIBSVM library. We consider logistic regression classification with a non-convex regularizer (leading to a non-convex problem). Formally, the local cost of each user $i \in [n]$ is given by
> $$f_i(x) = \frac{1}{m_i}\sum_{r \in [m_i]} \log\big(1+\exp(-y_{i,r}\langle h_{i,r}, x\rangle)\big) + \eta \sum_{k \in [d]} \frac{[x]_k^2}{1+[x]_k^2},$$
>
> where $(h_{i,r},y_{i,r})_{r \in [m_i]}$ are the data-label pairs, $m_i > 1$ is the size of the local dataset, $\eta > 0$ is a tunable parameter controlling the effect of the regularization, while $[x]_k$ denotes the $k$-th component of vector $x \in \mathbb{R}^d$. The full experimental setup (e.g., network size, weight matrix, step-size, etc) is similar to the synthetic experiments; we omit the details due to space limits, but would be happy to provide them if requested during discussion week. The results can be found via the following link and will be included in the final version: [link](https://drive.google.com/file/d/1_uzM5BcBElMVnhdTwUN6po0ZV9ZTyP30/view?usp=sharing).
>
> ## **Formating**
>
> We thank the reviewer for bringing the issue to our attention; it will be fixed in the final version.
>
> ## **Final remarks**
>
> We hope that our responses clarified the reviewer's concerns. In light of our responses, we kindly ask the reviewer to consider increasing the score. We would be very happy to answer any further questions during discussion week.
>
> ### **References**
>
> [1] Narayanan, D., et al. (2021). *Efficient large-scale language model training on GPU clusters using megatron-LM*. International Conference for High Performance Computing, Networking, Storage and Analysis. Association for Computing Machinery, NY, Article 58, pp. 1–15.
>
> [2] Zhang, J., et al. (2020). *Why are adaptive methods good for attention models?*. NeurlPS, 33, pp. 15383-15393.
>
> [3] Simsekli, U., et al. (2019). *A Tail-Index Analysis of Stochastic Gradient Noise in Deep Neural Networks*. 36th ICML, PMLR, 97, pp. 5827-5837.

---

> > ### Author Rebuttal · Reviewer_WSFW · 2026-04-04
> >
> > I would like to thank the authors for their time and effort in responding to my comments, especially clarifying the difference between HP and MSE analysis. However, it seems to me that the improvements the authors stated in their rebuttal for HP bounds over MSE, are due to noise being sub-Gaussian (light-tailed) according to Assumption A4-(3), and not inherently an advantage of HP analysis over MSE. Usually  MSE analysis works with general heavy-tailed noise as authors have also mentioned [2]-[3].
> >
> > Furthermore, the motivating example of this work being that HP analysis is suitable for very large models like LLMs for one-run analysis, is contrasting the experiments being done on synthetic data in the original manuscript, and only logistic regression during the rebuttal, both of which could be analyzed with MSE as average runs. For the above reasons, I will maintain my score.

---

> > > ### Author Response · Authors · 2026-04-04
> > >
> > > We would like to clarify the following, in relation to the reviewer's comments.
> > >
> > > ## On improvements in our work
> > >
> > > As stated in the original submission, we **improved the transient time** in the strongly convex case, which has nothing to do with noise conditions, but stems from a cleaner analysis. We also provided **novel technical results which are of independent interest**, like Lemmas 3.2 and 3.6. Further, in our response to reviewer z5vY, we extended the analysis of DSGD in the strongly convex case (Thm 3.7), under time-varying step-size $\alpha_t \propto t^{-\eta}$ with $\eta \in (1/2,1]$, achieving the following rate
> > > $$
> > > \frac{1}{n}\sum_{i \in [n]}(f(x_i^t) - f^*) = \mathcal{O}\bigg(\frac{1}{nt^{2\eta-1}}\bigg),
> > > $$
> > > which provides the following insight: the **optimal dependence on $n$ (i.e., linear speed-up) is attained by DSGD for strongly convex costs regardless of the choice of step-size; the choice of step-size dictates only the scale in $t$**. To the best of our knowledge, we are unaware of such result and insight in the decentralized MSE analysis.
> > >
> > > ## On numerical experiments
> > >
> > > We would like to remind the reviewer that our paper is theoretical, providing (i) guarantees for a different mode of convergence with improved transient times; (ii) several new technical results of independent interest; (iii) novel insight on the step-size effect and linear speed-up for strongly convex costs, discussed in the point above.  While we agree with the reviewer that an LLM experiment would be nice, due to time constraints of the discussion week, as well as our own computational resources, it is impossible for us to do so. We outlined LLMs as a motivating example, which the reviewer requested, but as the reviewer themselves said
> > >
> > > > Could the authors provide some insights and **ideally** provide numerical experiments
> > >
> > > numerical experiments in such settings would be ideal, but not mandatory and the insight that the reviewer requested was fully provided. Despite our work being of theoretical nature, we fully facilitated the reviewer's request for real data experiments, which further underlined the strong predictive power of our theory, **even in settings where the noise is not necessarily sub-Gaussian**. Regarding the comment that the scale of provided experiments is insufficient to justify HP results, **the fundamental difference between MSE and HP guarantees for a single run is always true, regardless of the scale of experiments**. Even if one can run logistic regression many times over in a matter of minutes, the guarantees implied by MSE for a single run will always be loose and this is only further exacerbated by larger models, which we used as motivation in explaining the difference between MSE and HP results.
> > >
> > > ## On noise assumptions
> > >
> > > There are a few important things to note. First, we used heavy-tailed noise to highlight **one of the many** possible instability phenomena in modern learning scenarios. This is not the only issue which necessitates HP analysis and it is a fact that even without heavy-tailed noise, modern large-scale models are trained for days on end and can not be quickly re-trained, so tight guarantees are needed for a single run. Next, the focus of this study is not on heavy-tailed noise, where even in MSE works it is necessary to use nonlinearities like clipping, normalization and sign, but on the light-tailed settings. The reviewer is correct that, while MSE results for vanilla SGD can be established under noise with bounded variance only, HP results for vanilla methods need sub-Gaussian noise to guarantee sufficient concentration. However, we note the following: (i) this is not specific to our analysis and it is the widely used condition for vanilla SGD/momentum/DSGD, please see the references in our literature review on HP results, both centralized and decentralized; (ii) if a bounded nonlinearity like clipping, normalization or sign is used, then it is well-known that both MSE and HP guarantees can be obtained under the *exact same conditions on the noise*, see our references on heavy-tailed noise in the centralized setting. Finally, the motivation for our work was to address a fundamental shortcoming of existing HP analysis of DSGD, i.e., assumptions on the cost stronger than in MSE results and no linear speed-up, which was present even in the light-tailed setting. While the noise assumption difference is necessary due to the type of guarantees (MSE vs HP), we showed that the cost assumption difference in existing HP works was completely the result of sub-optimal analyses. We believe that providing a tight analysis in this setting provides invaluable insights and a roadmap for future work in different settings, e.g., heavy-tailed noise.
> > >
> > > ## Final comments
> > >
> > > We hope that the response helped further clarify the merits of our work and that the reviewer will take our further comments and all the improvements provided in our work into full consideration when forming their final judgement.

---

### Official Review · Reviewer_JVfa · 2026-03-13

**Soundness:** 3
**Presentation:** 3
**Significance:** 3
**Originality:** 3
**Overall Recommendation:** 5
**Confidence:** 3

**Summary:**

In this work, the authors derive novel high-probability complexity bounds for Decentralized SGD, operating under standard light-tailed (sub-Gaussian) noise assumptions.

**Compliance With Llm Reviewing Policy:**

Affirmed.

**Final Justification:**

Overall, this is a strong theoretical paper that I recommend for acceptance.

**Key Questions For Authors:**

No Questions

**Limitations:**

No limitations

**Strengths And Weaknesses:**

### **Strengths**

1. **Presentation:** The paper is well-written, clearly structured, and easy to follow.
2. **Theoretical Contributions:** The manuscript presents interesting and solid theoretical results.

### **Weaknesses**

**1. Literature Review & Contextualization:** To enrich the literature review in the main text, I highly recommend discussing the following highly relevant works:

* **[1]:** This paper should be cited as it derives the first non-trivial high-probability complexity bounds for SGD with clipping without relying on a light-tails noise assumption.
* **[2]:** This work is highly relevant as it studies Distributed SGD with clipping in the centralized setting, achieving a linear speed-up under the $p$-Bounded Central Moment assumption.

**2. Venue Suitability (Minor):** From my perspective, the core contribution of this paper hinges heavily on its novel high-probability analysis. Because of this deep theoretical focus, the manuscript reads somewhat more like a journal submission (e.g., for JMLR) rather than a typical machine learning conference paper. However, I do not consider this a major issue or a reason for rejection, but rather a note on its framing.

**References:**

* [1] Gorbunov, E., Danilova, M. and Gasnikov, A., 2020. Stochastic optimization with heavy-tailed noise via accelerated gradient clipping. *Advances in Neural Information Processing Systems*, 33, pp.15042-15053.
* [2] Gorbunov, E., Sadiev, A., Danilova, M., Horváth, S., Gidel, G., Dvurechensky, P., Gasnikov, A. and Richtárik, P., 2024. High-Probability Convergence for Composite and Distributed Stochastic Minimization and Variational Inequalities with Heavy-Tailed Noise. In *Forty-first International Conference on Machine Learning*.

---

> ### Author Rebuttal · Authors · 2026-03-25
>
> We are glad that the reviewer found our paper well-written and clear, with strong theoretical contributions. We further thank the reviewer for bringing to our attention the two important works omitted from our literature review; both will be included in the final version. We would be glad to answer any further questions that the reviewer may have during discussion week.

---

> > ### Author Rebuttal · Reviewer_JVfa · 2026-04-01
> >
> > I thank the authors for their response and for agreeing to include the suggested references ([1] and [2]) in the final version of the paper. My concerns were primarily focused on the contextualization of the high-probability analysis within the existing literature on gradient clipping and heavy-tailed noise. Since the authors have committed to addressing these points and my original assessment of the paper's technical quality was already positive, I am maintaining my score of 5: Accept.

---

> > > ### Author Response · Authors · 2026-04-05
> > >
> > > We are glad that all of the reviewer's concerns are successfully addressed and thank the reviewer once again for their positive evaluation of our work.

---

### Decision · Program_Chairs · 2026-04-30

**Decision:**

Accept (regular)

**Comment:**

The paper establishes new high-probability convergence guarantees for standard Decentralized SGD under smoothness, (optionally) strong convexity, and sub-Gaussian stochastic noise assumptions. Notably, in contrast to prior high-probability results for DSGD, the analysis does not rely on uniform boundedness of gradients and applies directly to the standard algorithm, without requiring additional modifications such as gradient clipping. This represents a meaningful theoretical advancement.

Three of the four reviewers assigned scores of 5, highlighting the novelty, technical soundness, and importance of the results. Reviewer WSFW provided a score of 3 and raised several concerns; however, these appear relatively minor and do not undermine the main contributions of the paper.

Overall, considering the reviews, the authors’ rebuttal, and my own assessment, I recommend acceptance. The paper is technically solid and makes a valuable contribution to the literature.